# Near-Optimal Distributionally Robust Reinforcement Learning with General $L_p$ Norms

**Pierre Clavier**[*†]
Ecole Polytechnique, Inria

**Laixi Shi**[‡]
Caltech

**Erwan Le Pennec** [*]
Ecole polytechnique

**Eric Mazumdar** [‡]
Caltech

**Adam Wierman** [‡]
Caltech

**Matthieu Geist**[§]
Cohere

## Abstract

To address the challenges of sim-to-real gap and sample efficiency in reinforcement learning (RL), this work studies distributionally robust Markov decision processes (RMDPs) — optimize the worst-case performance when the deployed environment is within an uncertainty set around some nominal MDP. Despite recent efforts, the sample complexity of RMDPs has remained largely undetermined. While the statistical implications of distributional robustness in RL have been explored in some specific cases, the generalizability of the existing findings remains unclear, especially in comparison to standard RL. Assuming access to a generative model that samples from the nominal MDP, we examine the sample complexity of RMDPs using a class of generalized $L_p$ norms as the 'distance' function for the uncertainty set, under two commonly adopted $sa$-rectangular and $s$-rectangular conditions. Our results imply that RMDPs can be more sample-efficient to solve than standard MDPs using generalized $L_p$ norms in both $sa$- and $s$-rectangular cases, potentially inspiring more empirical research. We provide a near-optimal upper bound and a matching minimax lower bound for the $sa$-rectangular scenarios. For $s$-rectangular cases, we improve the state-of-the-art upper bound and also derive a lower bound using $L_\infty$ norm that verifies the tightness.

## 1 Introduction

Reinforcement learning (RL) [Sutton, 1988] is a popular paradigm in machine learning, particularly noted for its success in practical applications. The RL framework, usually modeled within the context of a Markov decision process (MDP), focuses on learning effective decision-making strategies based on interactions with a fixed environment. However, the work of Mannor et al. [2004], among others, has highlighted a vulnerability in RL strategies, revealing the sensitivity to inherent shift or estimation errors in the reward and transition probabilities. A specific example of this is when, because of a sim-to-real gap, policies learned in idealized environments fail when deployed in environments with slight changes or adversarial perturbations [Klopp et al., 2017, Mahmood et al., 2018].

To address this issue, distributionally robust RL, usually formulated as robust MDPs (RMDPs), proposed by Iyengar [2005] and Nilim and El Ghaoui [2005], have attracted considerable attention. RMDPs are formulated as max-min problems, seeking policies that are resilient to model environment

---

[*]CMAP, CNRS, Ecole Polytechnique, Institut Polytechnique de Paris, 91120 Palaiseau

[†]Inria Paris, HeKA, 75015 Paris, France

[‡]Department of Computing Mathematical Sciences, California Institute of Technology, CA 91125, USA.

[§]Cohere, Paris.

38th Conference on Neural Information Processing Systems (NeurIPS 2024).

perturbations within a specified uncertainty set. Despite the robustness benefits, solving RMDPs is NP-hard for general uncertainty sets [Nilim and El Ghaoui, 2005]. To overcome this challenge, the rectangularity condition is often adopted so that the uncertainty set can be decomposed as products of independent subsets for each state or state-action pair, denoted as $s$-rectangular or $sa$-rectangular assumptions (see Definition 4 and 5). These assumptions facilitate computation traceability of methods such as robust value iteration and robust policy iteration, preserving many structural properties of MDPs [Ho et al., 2021]. The $s$-rectangularity condition, though with less restrictive structure assumption, impose more challenges for algorithm design, while the $sa$-rectangularity condition allows for deterministic optimal policies akin to non-robust MDPs [Wiesemann et al., 2013]. Note that dealing with uncertainty in transition kernels is much more difficult than that in rewards [Kumar et al., 2022, Derman et al., 2021].

The question of sample efficiency is central in RL problems ranging from practice to theory. Although minimax sample efficiency has been achieved for standard MDPs [Azar et al., 2013b, Li et al., 2023c], this goal in general remains open in RMDPs. Specifically, there exists prior work studying the sample complexity of distributionally robust RL for a few specific divergences such as total variation ($TV$) distance, $\chi^2$ divergence, Kullback-Leibler divergence ($KL$) divergence, and Wasserstein distance (see discussions in Appendix A) [Yang et al., 2022b, Zhou et al., 2021, Panaganti and Kalathil, 2022]. While such results remain unclear for more general class, such as the general smooth $L_p$ norms (see Def. 1). To the best of our knowledge, minimax optimal sample complexity for the full range of uncertainty level has only been achieved for one case — $TV$ distance [Shi et al., 2023]. In this work, we focus on understanding the sample complexity of RMDPs with a general smooth $L_p$ that will be defined in Def. 1. This generalized result is appealing for both practice and theory. In practice, numerous applications are based on optimizations or learning approaches that involve general norms beyond those specific cases that have been studied in prior works. Additionally, optimizing $L_p$ norm weighted ambiguity sets for robust MDPs has been proposed in the context of RMDPs in Russel et al. [2019], which justifies our formulation. Theoretically, prior work has characterized the sample complexity of RMDPs for some specific norms have suggested intriguing insights about the statistical implications of distributional robustness in RL. It is interesting to further understand the statistical cost of robust RL in more general scenarios. One area of focus is the contrast between the sample efficiency of solving distributionally robust RL and solving standard RL. In particular, for the specific case of $TV$ distance, Shi et al. [2023] shows that the sample complexity for solving robust RL is at least the same as and sometimes (when the uncertainty level is relatively large) could be smaller than that of standard RL. This motivates the following open question:

*Is distributionally robust RL more sample efficient than standard RL for some general class of norms (Def.* (1)*)* ?

A second question is about the comparisons between the sample complexity of solving $s$-rectangular RMDPs and that of solving $sa$-rectangular RMDPs. Note that $s$-rectangular RMDPs involve more complex optimization problems with additional variables (uncertainty levels for each action) to optimize. This leads to a richer class of optimal policy candidates—stochastic policies in $s$-rectangular cases, in contrast to the class of deterministic policies for $sa$-rectangular cases. In addition, existing sample complexity upper bounds for solving $s$-rectangular RMDPs are larger than that for solving $sa$-rectangularity [Yang et al., 2022b] for the investigated cases. This motivates the curious question:

*Does solving $s$-rectangular RMDPs require more samples than solving $sa$-rectangular RMDPs with general smooth $L_p$ norms defined in Def. 1?*

**Main contributions.** In this paper, we address each of the two questions discussed above. In particular, we provide the first sample complexity analysis for RMDPs with general $L_p$ norms (cf. Def. 1) under both the $s$- and $sa$-rectangularity conditions. For convenience, we present detailed comparisons between the prior arts and our results in Table 1 for quick reference and discuss the contributions and their implications as below.

• Considering the first question, we illustrate our results in both $sa$- and $s$-rectangular cases in Figure 1. In the case of $sa$-rectangularity, we derive a sample complexity upper bound for RMDPs using general smooth $L_p$ norms (cf. Theorem 1) in the order of $\widetilde{O}\left(\frac{SA}{(1-\gamma)^2 \max\{1-\gamma, C_g\sigma\}\varepsilon^2}\right)$. Here, $\sigma$ is the uncertainty level/radius of the uncertainty set, and $C_g > 0$ is a positive constant related to the geometry of the norm defined in Def. 1. For classical $L_p$ norms, $C_g \geq 1$ so we can directly relax this constant to 1 to obtain the result in Table 1. In addition, we provide a matching minimax lower bound (cf. Theorem 2) that confirms the near-optimality of the upper bound for almost full range of the

| Result type | Reference | Distance | $sa$-rectangularity | | $s$-rectangularity | |
|---|---|---|---|---|---|---|
| | | | $0 < \sigma \lesssim 1-\gamma$ | $1-\gamma \lesssim \sigma < \sigma_{\max}$ | $0 < \tilde\sigma \lesssim 1-\gamma$ | $1-\gamma \lesssim \tilde\sigma < \tilde\sigma_{\max}$ |
| Upper bound | Yang et al. [2022a] | TV | $\frac{S^2A(2+\sigma)^2}{\sigma^2(1-\gamma)^4\varepsilon^2}$ | $\frac{S^2A(2+\sigma)^2}{\sigma^2(1-\gamma)^4\varepsilon^2}$ | $\frac{S^2A^2(2+\tilde\sigma)^2}{\tilde\sigma^2(1-\gamma)^4\varepsilon^2}$ | $\frac{S^2A^2(2+\tilde\sigma)^2}{\tilde\sigma^2(1-\gamma)^4\varepsilon^2}$ |
| | Panaganti and Kalathil [2022] | TV | $\frac{S^2A}{(1-\gamma)^4\varepsilon^2}$ | $\frac{S^2A}{(1-\gamma)^4\varepsilon^2}$ | $\times$ | $\times$ |
| | Shi et al. [2023] | TV | $\frac{SA}{(1-\gamma)^3\varepsilon^2}$ | $\frac{SA}{\sigma(1-\gamma)^2\varepsilon^2}$ | $\times$ | $\times$ |
| | Clavier et al. [2023] | $L_p$ | $\frac{SA}{(1-\gamma)^3\varepsilon^2}$ | $\frac{SA}{(1-\gamma)^4\varepsilon^2}$ | $\frac{SA}{(1-\gamma)^3\varepsilon^2}$ | $\frac{SA}{(1-\gamma)^4\varepsilon^2}$ |
| | **This paper** | $L_p$ | $\frac{SA}{(1-\gamma)^3\varepsilon^2}$ | $\frac{SA}{\sigma(1-\gamma)^2\varepsilon^2}$ | $\frac{SA}{(1-\gamma)^3\varepsilon^2}$ | $\frac{SA}{(1-\gamma)^2\tilde\sigma\min_s\|\pi_s\|_*\varepsilon^2}$ |
| | **This paper** | General $L_p$ [1] | $\frac{SA}{(1-\gamma)^3\varepsilon^2}$ | $\frac{SA}{\sigma(1-\gamma)^2\varepsilon^2}$ | $\frac{SA}{(1-\gamma)^3\varepsilon^2}$ | $\frac{SA}{(1-\gamma)^2\tilde\sigma C_g\min_s\|\pi_s\|_*\varepsilon^2}$ |
| Lower bound | Yang et al. [2022a] | TV | $\frac{SA}{(1-\gamma)^3\varepsilon^2}$ | $\frac{SA(1-\gamma)}{\sigma^4\varepsilon^2}$ | $\times$ | $\times$ |
| | Shi et al. [2023] | TV | $\frac{SA}{(1-\gamma)^3\varepsilon^2}$ | $\frac{SA}{\sigma(1-\gamma)^2\varepsilon^2}$ | $\times$ | $\times$ |
| | **This paper** | $L_p$ | $\frac{SA}{(1-\gamma)^3\varepsilon^2}$ | $\frac{SA}{\sigma(1-\gamma)^2\varepsilon^2}$ | $\times$ | $\times$ |
| | **This paper** | $L_\infty$ | $\frac{SA}{(1-\gamma)^3\varepsilon^2}$ | $\frac{SA}{\sigma(1-\gamma)^2\varepsilon^2}$ | $\frac{SA}{(1-\gamma)^3\varepsilon^2}$ | $\frac{SA}{\tilde\sigma(1-\gamma)^2\varepsilon^2}$ |

Table 1: Comparisons with prior results (up to log terms) regarding finding an $\varepsilon$-optimal policy for the distributionally RMDP, where $\sigma$ is the radius of the uncertainty set and $\sigma_{\max}$ defined in Theorem 1.

uncertainty level. Our results match the near-optimal sample complexity derived in Shi et al. [2023] for the specific case using TV distance, while holding for broader cases using general $L_p$ norms. The results rely on a new dual optimization form for $sa$-rectangular RMDPs and reveal the relationship between the sample complexity and this new dual form — the infinite span seminorm (controlled in Lemma 5), which may be of independent interest.

In the case of $s$-rectangularity, we provide a sample complexity upper bound for solving RMDPs with general smooth $L_p$ norms in the order of $\widetilde{O}\left(\frac{SA}{(1-\gamma)^2 \max\{1-\gamma, C_g \min_s\|\pi_s\|_*\tilde\sigma\}\varepsilon^2}\right)$ with $\|.\|_*$ the dual norm and $\tilde\sigma$ the radius of the ball in the $s$-rectangular uncertainty set. This result improves the prior art $\widetilde{O}\left(\frac{SA}{(1-\gamma)^4\varepsilon^2}\right)$ in Clavier et al. [2023] for classical $L_p$ — by at least a factor of $O\left(\frac{1}{1-\gamma}\right)$ when $\tilde\sigma \lesssim 1-\gamma$. Furthermore, we present a lower bound for a representative case with $L_\infty$ norm, which corroborates the tightness of the upper bound. To the best of our knowledge, this is the first lower bound for solving RMDPs with $s$-rectangularity.

• We highlight the technical contributions as below. For the upper bounds, regarding optimization contribution, we derive new dual optimization problem forms for both $sa-$ and $s-$ rectangular cases (Lemma 3 and 4), which is the foundation of the covering number argument in finite-sample analysis. From a statistical point of view, a new concentration lemma (See Lemma 8 for dual forms) is introduced to obtain a lower sample complexity than standard RL, controlling the infinite span semi norm of the value function, both for $sa-$ and $s-$ rectangular case are derived (See Lemma 5 and 6). For the lower bound, the technical contributions are mainly in $s$-rectangular cases, which involves entire new challenges compared to $sa$-rectangularity case: the optimal policies can be stochastic and hard to be characterized as a closed form, compared to the deterministic one in $sa$-rectangular cases. Therefore, we construct new hard instances for $s$-rectangular cases that is distinct from those used in $sa$-rectangular cases or standard RL.

• Considering the second question, as illustrated in Figure 1, our results highlight that robust RL is at least the same as and sometimes can be more sample-efficient to solve than standard RL for general smooth $L_p$ norms (cf. Def. 1). This insight is of significant practical importance and serves to provide crucial motivation for the use and study of distributionally robustness in RL. Notably, robust RL does not only reduce the vulnerability of RL policy to estimation errors and sim-to-real gaps, but also leads to better data efficiency. In terms of comparing the statistical implications of $sa$- and $s$- rectangularity, our results show that solving $s$-rectangular RMDPs is not harder than solving $sa$-rectangular RMDPs in terms of sample requirement (See Theorem 3 and Figure 2, Right).

## 2 Problem Formulation: Robust Markov Decision Processes

In this section, we formulate distributionally robust Markov decision processes (RMDPs) in the discounted infinite-horizon setting, introduce the sampling mechanism, and describe our goal.

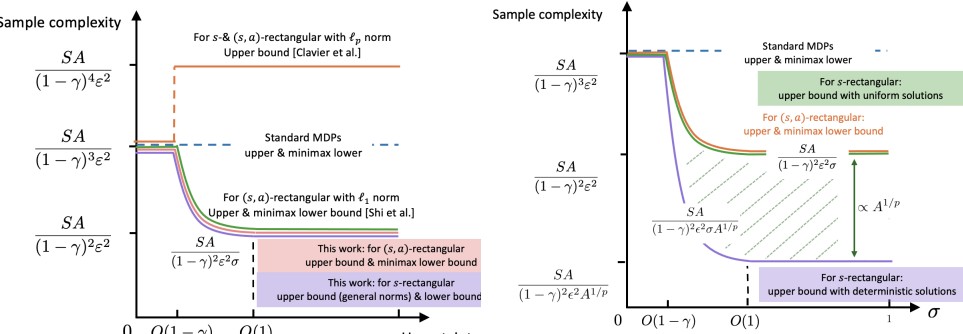

Figure 1: **Left**: Sample complexity results for RMDPs with $sa$- and $s$-rectangularity with $L_p$ with comparisons to prior arts [Shi et al., 2023] (for $L_1$ norm, or called total variation distance) and [Clavier et al., 2023] ; **Right:** The data and instance-dependent sample complexity upper bound of solving $s$-rectangular dependency RMDPs with $L_p$ norms.

**Standard Markov decision processes (MDPs).** A discounted infinite-horizon MDP is represented by $\mathcal{M} = (\mathcal{S}, \mathcal{A}, \gamma, P, r)$, where $\mathcal{S} = \{1, \cdots, S\}$ and $\mathcal{A} = \{1, \cdots, A\}$ are the finite state and action spaces, respectively, $\gamma \in [0, 1)$ is the discounted factor, $P : \mathcal{S} \times \mathcal{A} \to \Delta(\mathcal{S})$ denotes the probability transition kernel, and $r : \mathcal{S} \times \mathcal{A} \to [0, 1]$ is the immediate reward function, which is assumed to be deterministic. Moreover, we assume that the reward function is bounded in $(0, 1)$ without loss of generality of the results due to the variance reward invariance. Finally we denote $1_A$ or $1_S$ the unitary vector of respectively dimension $A$ or $S$. Moreover, $e_s$ is the standard unitary vector supported on $s$. The policy we are looking for is denoted by $\pi : \mathcal{S} \to \Delta(\mathcal{A})$, which specifies the probability of action selection over the action space in any state. Note that if the policy is deterministic in the $sa$-rectangular case, we overload the notation and refer to $\pi(s)$ as the action selected by the policy $\pi$ in state $s$. Finally, to characterize the cumulative reward, the value function $V^{\pi,P}$ for any policy $\pi$ under the transition kernel $P$ is defined by $\forall s \in \mathcal{S}$

$$V^{\pi,P}(s) := \mathbb{E}_{\pi,P}\left[\sum_{t=0}^{\infty} \gamma^t r(s_t, a_t) \,\Big|\, s_0 = s\right]. \tag{1}$$

The expectation is taken over the randomness of the trajectory $\{s_t, a_t\}_{t=0}^{\infty}$ generated by executing the policy $\pi$ under the transition kernel $P$, such that $a_t \sim \pi(\cdot \,|\, s_t)$ and $s_{t+1} \sim P(\cdot \,|\, s_t, a_t)$ for all $t \geq 0$. In the same way, the Q function $Q^{\pi,P}$ associated with any policy $\pi$ under the transition kernel $P$ is defined using expectation taken over the randomness of the trajectory under policy $\pi$ as

$$Q^{\pi,P}(s, a) := \mathbb{E}_{\pi,P}\left[\sum_{t=0}^{\infty} \gamma^t r(s_t, a_t) \,\Big|\, s_0, a_0 = s, a\right], \tag{2}$$

**Distributionally robust MDPs.** We consider distributionally robust MDPs (RMDPs) in the discounted infinite-horizon setting, denoted by $\mathcal{M}_{\text{rob}} = \{\mathcal{S}, \mathcal{A}, \gamma, \mathcal{U}^\sigma_{\|\cdot\|}(P^0), r\}$, where $\mathcal{S}, \mathcal{A}, \gamma, r$ are the same sets and parameters as in standard MDPs. The main difference compared to standard MDPs is that instead of assuming a fixed transition kernel $P$, it allows the transition kernel to be arbitrarily chosen from a prescribed uncertainty set $\mathcal{U}^\sigma_{\|\cdot\|}(P^0)$ centered around a *nominal* kernel $P^0 : \mathcal{S} \times \mathcal{A} \to \Delta(\mathcal{S})$, where the uncertainty set is specified using some called $L_p$ smooth norm denoted $\|\cdot\|$ defined in of radius $\sigma > 0$ defined in 1.

**Definition 1** (General smooth $L_p$ norms and dual norms). *A norm $\|\cdot\|$ is said to be a general smooth $L_p$, norm , $p > 1$ if*

- *for all $x \in \mathbb{R}^n$, $\|x\| := \|x\|_{p,w} = (\sum_{k=1}^n w_k(|x_k|)^p)^{1/p}$ for some $w \in \mathbb{R}^n_+$, being an arbitrary positive vector.*

- *it is twice continuously differentiable Rudin et al. [1964] with the supremum of the Hessian Matrix over the simplex $C_S = \sup_{x \in \Delta_S} \|\nabla^2 \|x\|\|_2$, where $\|\cdot\|_2$ here is the spectral norm.*

*Finally, we denote the dual norm of $\|\cdot\|$ as $\|\cdot\|_*$ s.t. $\|y\|_* := \max_x x^T y : \|x\| \leq 1$. Moreover, for any metric $\|\cdot\|$, we define $C_g$ $C_g := 1/\min_s \|e_s\|$ where $e_s \in \mathbb{R}^S$ is the standard basis vector with only $1$ at the $s$-th entry, otherwise $0$.*

Note that the quantity $C_S$ exists, as the Hessian of a $C^2$ functional is continuous and because the simplex is a compact set, so by Extreme Value Theorem Rudin et al. [1964], $C_S$ is finite. For example, considering $L_p$ norms with any $p \geq 2$, $C_S$ is bounded by $(p-1)S^{1/q}$. (See (154) ) This definition is general and includes $L_p$ norms [Rudin et al., 1964] for any $p \geq 2$ and all rescaled and weighted norms. Moreover, we could extend our results to a larger set than the one of the norms defined in Def. 1, where the further discussion can be found in Appendix B. However, it does not include divergences such as $KL$ and $\chi^2$. Not that the case of $TV$ which is not $C^2$ smooth is treated independently with different arguments in the proof but has the same sample complexity. In particular, given the nominal transition kernel $P^0$ and some uncertainty level $\sigma$, the uncertainty set—with arbitrary smooth $L_p$ norm metric $\| \| : \mathbb{R}^S \times \to \mathbb{R}^+$ in $sa$ rectangular case or from $\mathbb{R}^{S \times A}$ in the $s$-rectangular case, is specified as $\mathcal{U}^\sigma_{\|\cdot\|}(P^0) := \otimes_{s,a} \mathcal{U}^{\mathsf{sa},\sigma}_{\|\cdot\|}(P^0_{s,a})$

$$\mathcal{U}^{\mathsf{sa},\sigma}_{\|\cdot\|}(P^0_{s,a}) := \left\{ P_{s,a} \in \Delta(\mathcal{S}) : \left\| P_{s,a} - P^0_{s,a} \right\| \leq \sigma \right\}, \tag{3}$$

$$P_{s,a} := P(\cdot \,|\, s,a) \in \mathbb{R}^{1 \times S}, P^0_{s,a} := P^0(\cdot \,|\, s,a) \in \mathbb{R}^{1 \times S}, \tag{4}$$

where we denote a vector of the transition kernel $P$ or $P^0$ at state-action pair $(s,a)$. In other words, the uncertainty is imposed in a decoupled manner for each state-action pair, obeying the so-called $sa$-rectangularity [Zhou et al., 2021, Wiesemann et al., 2013]. More generally, we define $s$-rectangular MDPs as $\mathcal{U}^\sigma_{\|\cdot\|}(P) = \otimes_s \mathcal{U}^{\mathsf{s},\widetilde{\sigma}}_{\|\cdot\|}(P_s)$, for the general smooth $L_p$ norm $\|\cdot\|$. The uncertainty is imposed in a decoupled manner for each state pair, and a fixed budget given a state for all action is defined. To get a similar meaning for the radius of the ball between $sa$-rectangular and $s$-rectangular assumptions, we need to rescale the radius depending on the norm like in Yang et al. [2022b]. The $s$- uncertainty set is then defined using the rescaled radius $\tilde{\sigma}$ as

$$\mathcal{U}^{\mathsf{s},\widetilde{\sigma}}_{\|\cdot\|}(P_s) := \left\{ P'_s \in \Delta(\mathcal{S})^{\mathcal{A}} : \|P'_s - P_s\| \leq \tilde{\sigma} = \sigma \|1_A\| \right\}, \tag{5}$$

$$P_s := P(\cdot,\cdot \,|\, s) \in \mathbb{R}^{1 \times SA}, \quad P^0_s := P^0(\cdot,\cdot \,|\, s) \in \mathbb{R}^{1 \times SA}. \tag{6}$$

where $1_A \in \mathbb{R}^A$ denotes the unitary vector. For the specific case of respectively $L_1, L_p$ and $L_\infty$ norm, $\tilde{\sigma}$ is equal to $|\sigma\mathcal{A}|, \sigma|\mathcal{A}|^{1/p}$ and $\sigma$. Note that this scaling allows for a fair comparison between $sa$- and $s$-rectangular MDPs. In RMDPs, we are interested in the worst-case performance of a policy $\pi$ over all the possible transition kernels in the uncertainty set. This is measured by the *robust value function* $V^{\pi,\sigma}$ and the *robust Q-function* $Q^{\pi,\sigma}$ in $\mathcal{M}_{\mathsf{rob}}$, defined respectively as $\forall(s,a) \in \mathcal{S} \times \mathcal{A}$

$$V^{\pi,\sigma}(s) := \inf_{P \in \mathcal{U}^{\mathsf{sa},\sigma}_{\|\cdot\|}(P^0)} V^{\pi,P}(s), \quad Q^{\pi,\sigma}(s,a) := \inf_{P \in \mathcal{U}^{\mathsf{sa},\sigma}_{\|\cdot\|}(P^0)} Q^{\pi,P}(s,a). \tag{7}$$

Similarly for $s$-rectangularity, the value function is denoted $V^{\pi,\sigma}_s(s) := \inf_{P \in \mathcal{U}^{\mathsf{s},\widetilde{\sigma}}_{\|\cdot\|}(P^0)} V^{\pi,P}(s)$.

**Optimal robust policy and robust Bellman operator.** As a generalization of properties of standard MDPs in the $sa$-rectangular robust case, it is well-known that there exists at least one deterministic policy that maximizes the robust value function (resp. robust Q-function) simultaneously for all states (resp. state-action pairs) [Iyengar, 2005, Nilim and El Ghaoui, 2005] but not in the $s$-rectangular case. Therefore, we denote the *optimal robust value function* (resp. *optimal robust Q-function*) as $V^{\star,\sigma}$ (resp. $Q^{\star,\sigma}$), and the optimal robust policy as $\pi^\star$, which satisfy $\forall(s,a) \in \mathcal{S} \times \mathcal{A}$

$$V^{\star,\sigma}(s) := V^{\pi^\star,\sigma}(s) = \max_\pi V^{\pi,\sigma}(s), \quad Q^{\star,\sigma}(s,a) := Q^{\pi^\star,\sigma}(s,a) = \max_\pi Q^{\pi,\sigma}(s,a). \tag{8a}$$

A key concept in RMDPs is a generalization of Bellman's optimality principle, encapsulated in the following *robust Bellman consistency equation* (resp. *robust Bellman optimality equation*):

$$\forall(s,a) \in \mathcal{S} \times \mathcal{A}, \quad Q^{\pi,\sigma}(s,a) = r(s,a) + \gamma \inf_{\mathcal{P} \in \mathcal{U}^{\mathsf{sa},\sigma}_{\|\cdot\|}(P^0_{s,a})} \mathcal{P}V^{\pi,\sigma}, \tag{9a}$$

$$\forall(s,a) \in \mathcal{S} \times \mathcal{A}, \quad Q^{\star,\sigma}(s,a) = r(s,a) + \gamma \inf_{\mathcal{P} \in \mathcal{U}^{\mathsf{sa},\sigma}_{\|\cdot\|}(P^0_{s,a})} \mathcal{P}V^{\star,\sigma}. \tag{9b}$$

for the $sa$-rectangular case and same equation replacing $P^0_{s,a}$ by $P^0_s$ and $\sigma$ by $\tilde{\sigma}$. The robust Bellman operator [Iyengar, 2005, Nilim and El Ghaoui, 2005] is denoted by $\mathcal{T}^\sigma(\cdot) : \mathbb{R}^{SA} \to \mathbb{R}^{SA}$

$$\mathcal{T}^\sigma(Q^\pi)(s,a) := r(s,a) + \gamma \inf_{\mathcal{P} \in \mathcal{U}^{\mathsf{sa},\sigma}_{\|\cdot\|}(P^0_{s,a})} \mathcal{P}V, \quad \text{with} \quad V(s) := \max_\pi Q^\pi(s,a)\cdot \qquad (10)$$

for $sa$-rectangular MDPs. Given that $Q^{\star,\sigma}$ is the unique-fixed point of $\mathcal{T}^\sigma$ one can recover the optimal robust value function and Q-function using a procedure termed *distributionally robust value iteration* ($DRVI$). Generalizing the standard value iteration, $DRVI$ starts from some given initialization and recursively applies the robust Bellman operator until convergence. As has been shown previously, this procedure converges rapidly due to the $\gamma$-contraction property of $\mathcal{T}^\sigma$ with respect to the $L_\infty$ norm [Iyengar, 2005, Nilim and El Ghaoui, 2005].

# 3 Distributionally Robust Value Iteration

**Generative model-based sampling.** Following Zhou et al. [2021], Panaganti and Kalathil [2022], we assume access to a generative model or a simulator [Kearns and Singh, 1999], which allows us to collect $N$ independent samples for each state-action pair generated based on the *nominal* kernel $P^0$: $\forall (s,a) \in \mathcal{S} \times \mathcal{A}$, $s_{i,s,a} \overset{i.i.d}{\sim} P^0(\cdot \,|\, s,a)$, $i = 1, 2, \cdots, N$. The total sample size is, therefore, $NSA$. We consider a model-based approach tailored to RMDPs, which first constructs an empirical nominal transition kernel based on the collected samples and then applies distributionally robust value iteration (DRVI) to compute an optimal robust policy. As we decouple the statistical estimation error and the optimization error, we exhibit an algorithm that can achieve arbitrary small error $\epsilon_{opt}$ in the empirical MDP defined as an empirical nominal transition kernel $\widehat{P}^0 \in \mathbb{R}^{SA \times S}$ that can be constructed on the basis of the empirical frequency of state transitions, i.e. $\forall (s,a) \in \mathcal{S} \times \mathcal{A}$

$$\widehat{P}^0(s' \,|\, s,a) := \frac{1}{N} \sum_{i=1}^{N} \mathbb{1}\{s_{i,s,a} = s'\}, \qquad (11)$$

which leads to an empirical RMDP $\widehat{\mathcal{M}}_{\mathsf{rob}} = \{\mathcal{S}, \mathcal{A}, \gamma, \mathcal{U}^\sigma_{\|\cdot\|}(\widehat{P}^0), r\}$. Analogously, we can define the corresponding robust value function (resp. robust Q-function) of policy $\pi$ in $\widehat{\mathcal{M}}_{\mathsf{rob}}$ as $\widehat{V}^{\pi,\sigma}$ (resp. $\widehat{Q}^{\pi,\sigma}$) (cf. (8)). In addition, we denote the corresponding *optimal robust policy* as $\widehat{\pi}^\star$ and the *optimal robust value function* (resp. *optimal robust Q-function*) as $\widehat{V}^{\star,\sigma}$ (resp. $\widehat{Q}^{\star,\sigma}$) (cf. (9)), which satisfies the robust Bellman optimality equation $\forall (s,a) \in \mathcal{S} \times \mathcal{A}$:

$$\widehat{Q}^{\star,\sigma}(s,a) = r(s,a) + \gamma \inf_{\mathcal{P} \in \mathcal{U}^{\mathsf{sa},\sigma}_{\|\cdot\|}(\widehat{P}^0_{s,a})} \mathcal{P}\widehat{V}^{\star,\sigma}. \qquad (12)$$

Equipped with $\widehat{P}^0$, we can define the empirical robust Bellman operator $\widehat{\mathcal{T}}^\sigma$ as $\forall (s,a) \in \mathcal{S} \times \mathcal{A}$

$$\widehat{\mathcal{T}}^\sigma(Q^\pi)(s,a) := r(s,a) + \gamma \inf_{\mathcal{P} \in \mathcal{U}^{\mathsf{sa},\sigma}_{\|\cdot\|}(\widehat{P}^0_{s,a})} \mathcal{P}V, \qquad (13)$$

with $V(s) := \max_\pi Q^\pi(s,a)$. The aim of this work is given the collected samples, to learn the robust optimal policy for the RMDP w.r.t. some prescribed uncertainty set $\mathcal{U}^\sigma(P^0)$ around the nominal kernel using as few samples as possible. Specifically, given some target accuracy level $\varepsilon > 0$, the goal is to seek an $\varepsilon$-optimal robust policy $\widehat{\pi}$ obeying

$$\forall s \in \mathcal{S}: \quad V^{\star,\sigma}(s) - V^{\widehat{\pi},\sigma}(s) \le \varepsilon, \qquad (14)$$

$$\widehat{V}^{\widehat{\pi}^\star,\sigma} - \widehat{V}^{\widehat{\pi},\sigma} \le \varepsilon_{\mathsf{opt}}. \qquad (15)$$

This formulation allows plugging any solver of RMDPs in this bound, for instance, the distributionally robust value iteration (DRVI) algorithm detailed in Appendix G.

# 4 Theoretical guarantees

In this section, we present our main results characterizing the sample complexity of solving RMDPs with $sa$-and $s$-rectangularity. Additionally, we discuss the implications of our results for the comparisons between standard and robust RL, and for comparisons between $sa$- versus $s$-rectangularity.

## 4.1 $sa$-rectangular uncertainty set with general smooth norms

To begin, we consider the RMDPs with $sa$-rectangularity with general norms. We first provide the following sample complexity upper bound for certain oracle planning algorithms, whose proof is postponed to Appendix D.2. Technically, we derive two new dual forms for RMDPs problems using arbitrary norms in Lemmas 3 and 4 for respectively $sa$- and $s$-rectangular RMDPS. In these dual forms, a central quantity denoted $\mathrm{sp}(.)_*$, representing the dispersion of the value function, appears and is the dual span semi-norm associated with the considered general $L_p$ norm $\|.\|$ defined in 1 in the initial primal problem. The main challenge in this analysis is to derive a tight upper bound on this quantity in Lemmas (5) and (6), leading to the following sample complexity.

**Theorem 1** (Upper bound for $sa$-rectangularity). *Consider the uncertainty set $\mathcal{U}^{\mathsf{sa},\sigma}_{\|\cdot\|}(\cdot)$ associated with arbitrary $L_p$ smooth norm $\|\cdot\|$ defined in 1. We denote $\sigma_{\max} := \max_{p_1,p_2 \in \Delta(\mathcal{S})} \|p_1 - p_2\|$ as the accessible maximal uncertainty level. Consider any $\delta \in (0,1)$, discount factor $\gamma \in \left[\frac{1}{4}, 1\right)$, and uncertainty level $\sigma \in (0, \sigma_{\max}]$. Let $\widehat{\pi}$ be the output policy of some oracle planning algorithm with optimization error $\varepsilon_{\mathsf{opt}}$ introduced in (15). With introduced in 1, one has with probability at least $1 - \delta$,*

$$\forall s \in \mathcal{S}: \quad V^{\star,\sigma}(s) - V^{\widehat{\pi},\sigma}(s) \leq \varepsilon + \frac{8\varepsilon_{\mathsf{opt}}}{1-\gamma} \tag{16}$$

*for any $\varepsilon \in (0, \sqrt{1/\max\{1-\gamma, \sigma C_g\}}]$, as long as the total number of samples obeys*

$$NSA \gtrsim \frac{c_1 SA}{(1-\gamma)^2 \max\{1-\gamma, C_g\sigma\}\varepsilon^2} + \frac{c_2 SA C_S \|1_S\|_*}{(1-\gamma)^2 \epsilon} \tag{17}$$

*with $c_1, c_2, c_3$ a universal positive constant. For a sufficiently small level of accuracy $\epsilon \leq (\max\{1-\gamma, C_g\sigma\})/(C_S \|1_S\|)$, the sample complexity is*

$$NSA \gtrsim \frac{c_3 SA}{(1-\gamma)^2 \max\{1-\gamma, C_g\sigma\}\varepsilon^2}. \tag{18}$$

Note that this result is also true for $TV$ without the geometric smooth term depending on $C_S$. Considering $L_p$ norms, $C_g \geq 1$ and $C_S \leq S^{1/q}(p-1)$. In Theorem 1, we introduce the following minimax-optimal lower bound to verify the tightness of the above upper bound; a proof is provided in Appendix E.

**Theorem 2** (Lower bound for $sa$-rectangularity). *Consider the uncertainty set $\mathcal{U}^{\mathsf{sa},\sigma}_{\|\cdot\|}(\cdot)$ associated with arbitrary $L_P$ norm $\|\cdot\|$ defined in 1. We denote $\sigma_{\max} := \max_{p_1,q_1 \in \Delta(\mathcal{S})} \|p_1 - p_2\|$ as the accessible maximal uncertainty level. Consider any tuple $(S, A, \gamma, \sigma, \varepsilon)$, where $\gamma \in \left[\frac{1}{2}, 1\right)$, $\sigma \in (0, \sigma_{\max}(1 - c_0)]$ with $0 < c_0 \leq \frac{1}{8}$ being any small enough positive constant, and $\varepsilon \in \left(0, \frac{c_0}{256(1-\gamma)}\right]$. We can construct two infinite-horizon RMDPs $\mathcal{M}_0, \mathcal{M}_1$ such that giving a dataset with $N$ independent samples for each state-action pair over the nominal transition kernel (for either $\mathcal{M}_0$ or $\mathcal{M}_1$ respectively), one has*

$$\inf_{\widehat{\pi}} \max_{\mathcal{M} \in \{\mathcal{M}_0, \mathcal{M}_1\}} \left\{ \mathbb{P}_{\mathcal{M}}\left( \max_{s \in \mathcal{S}} \left[V^{\star,\sigma}(s) - V^{\widehat{\pi},\sigma}(s)\right] > \varepsilon \right) \right\} \geq \frac{1}{8},$$

*where the infimum is taken over all estimators $\widehat{\pi}$, $\mathbb{P}_0$ (resp. $\mathbb{P}_1$) are the probability when the RMDP is $\mathcal{M}_0$ (resp. $\mathcal{M}_1$), as long as, for $c_7$ is a universal positive constant,*

$$NSA \leq \frac{c_7 SA}{(1-\gamma)^2 \max\{1-\gamma, C_g\sigma\}\varepsilon^2}. \tag{19}$$

• **Near minimax-optimal sample complexity with general $L_p$ norms.** Recall that Theorem 1 shows that the sample complexity upper bound of oracle algorithms for RMDPs is in the order of $\widetilde{O}\left(\frac{SA}{(1-\gamma)^2 \max\{1-\gamma, C_g\sigma\}\varepsilon^2}\right)$. Combined with the lower bound in Theorem 2, we observe that the above sample complexity is near minimax-optimal, in almost the full range of uncertainty.

• **Solving RMDPs with general $L_p$ norms can be easier than solving standard RL.** Recall that the sample complexity of solving standard RL with a generative model [Agarwal et al., 2020, Li

et al., 2024, Azar et al., 2013a] is: $\widetilde{O}\left(\frac{SA}{(1-\gamma)^3\varepsilon^2}\right)$. Comparing this with the sample complexity in (18), it highlights that solving robust MDPs (cf. (18)) using any norm as the divergence function for the uncertainty set is not harder than (and is sometimes easier than) solving standard RL (cf. (4.1)). Specifically, when the uncertainty level is small $\sigma \lesssim 1-\gamma$, the sample complexity of solving robust MDPs matches that of standard MDPs. While when the uncertainty level is relatively larger $1-\gamma \lesssim \sigma \leq \sigma_{\max}$, the sample complexity of solving robust MDPs is smaller than that of standard MDPs by a factor or $\frac{\sigma}{1-\gamma}$, which goes to $\frac{1}{1-\gamma}$ when $\sigma = O(1)$.

• **Comparisons with prior arts.** In Figure 1, we illustrate the comparisons with two state-of-the-arts [Clavier et al., 2023, Shi et al., 2023] which use some divergence functions belonging to the class of general norms considered in this work. In particular, Shi et al. [2023] achieved the state-of-the-art minimax-optimal sample complexity $\widetilde{O}\left(\frac{SA}{(1-\gamma)^2\max\{1-\gamma,\sigma\}\varepsilon^2}\right)$ for specific $L_1$ norm (or called total variation distance). In this work, we attain near minimax-optimal sample complexity for any general norm (including $L_1$) which matches the one in Shi et al. [2023] when narrowing down to $L_1$ norm. Note that in $TV$ case, $C_g = 1$. This reveals that the finding of robust MDPs can be easier than standard MDPs [Shi et al., 2023] in terms of sample requirement does not only hold for $L_1$ norm, but for any general norm. In addition, compared to Clavier et al. [2023] which focuses on $L_p$ norms for any $1 \leq p \leq \infty$: when $1-\gamma \lesssim \sigma \leq \sigma_{\max}$, we improve the sample complexity $\widetilde{O}(\frac{SA}{(1-\gamma)^4\varepsilon^2})$ to $\widetilde{O}(\frac{SA}{(1-\gamma)^2\sigma\varepsilon^2})$ by at least a factor of $\frac{1}{1-\gamma}$; otherwise, we match the results in Clavier et al. [2023].

• **Burn-in Condition, $C_g$ factor and $TV$ case :** In Th. 1 and 3 we need a sufficiently small level of accuracy $\epsilon \leq (\max\{1-\gamma, C_g\sigma\})/(C_s\|1_S\|)$, to obtain the sample complexity. This type of condition is usual in MDPS analysis Shi et al. [2022] and is equivalent to burn in term. Moreover, the quantity $C_S$ exists (see 1) and for example, considering $L_p$ norms, $C_S$ is bounded by $S^{1/q}$. (See (154)) and the product $C_S\|1_S\|$ is upper bounded by $S$ for $L_2$ norm. Moreover, note that our theorem for the smooth norm is also true for $TV$ which is not $C^2$ and has the same complexity as (Shi et al. [2023]. In this case, the burn-in condition is not needed. (See Lemma D.3.3). Finally, the factor $C_g = 1/\min_s\|e_s\|$ is norm dependent and depends on how big the vector $e_{s_0}$ is in the considered norm. Note for classical $L_p$ this quantity is bigger than 1, which reduces the sample complexity.

### 4.2 $s$-rectangular uncertainty set with general norms

To continue, we move on to the case when the uncertainty set is constructed under $s$-rectangularity smooth norm. The following theorem presents the sample complexity upper bound for learning an $\epsilon$-optimal policy for RMDPs with $s$-rectangularity. A proof is shown in Appendix D.2.

**Theorem 3** (Upper bound for $s$-rectangularity). *Consider the uncertainty set $\mathcal{U}_{\|\cdot\|}^{s,\widetilde{\sigma}}(\cdot)$ with $s$-rectangularity. Consider any discount factor $\gamma \in \left[\frac{1}{4}, 1\right)$, the rescaled uncertainty level $\tilde{\sigma} = \sigma\|1_A\|$, and denote $\tilde{\sigma}_{\max} := \|1_A\|\max_{p_1,p_2\in\Delta(\mathcal{S})}\|p_1-p_2\|$ and $\delta \in (0,1)$. Let $\widehat{\pi}$ be the output policy of an arbitrary optimization algorithm with error $\varepsilon_{\mathsf{opt}}$. , with probability at least $1-\delta$, one has for any $\varepsilon \in (0, \sqrt{1/\max\{1-\gamma, C_g\min_s\|\pi_s\|_*\sigma\}}], \forall s \in \mathcal{S}: \quad V^{\star,\widetilde{\sigma}}(s) - V^{\widehat{\pi},\widetilde{\sigma}}(s) \leq \varepsilon + \frac{8\varepsilon_{\mathsf{opt}}}{1-\gamma}$ as long as the total number of samples obeys*

$$NSA \gtrsim \frac{c_4 SA}{(1-\gamma)^2\varepsilon^2}\min\left\{\frac{1}{\max\{1-\gamma, C_g\sigma\}}, \frac{1}{\sigma C_g\min_{s\in\mathcal{S}}\{\|\pi_s^*\|_*\|1_A\|, \|\hat{\pi}_s\|_*\|1_A\|\}}\right\} + \frac{c_5 SAC_S\|1_S\|_*}{(1-\gamma)^2\epsilon}$$

(20)

*For a sufficiently small accuracy, $\epsilon \leq (\max\{1-\gamma, C_g\tilde{\sigma}\})/(C_s\|1_S\|)$ the sample complexity is*

$$NSA \gtrsim \frac{c_6 SA}{(1-\gamma)^2\varepsilon^2}\min\left\{\frac{1}{\max\{1-\gamma, C_g\sigma\}}, \frac{1}{\sigma C_g\min_{s\in\mathcal{S}}\{\|\pi_s^*\|_*\|1_A\|, \|\hat{\pi}_s\|_*\|1_A\|\}}\right\}$$ (21)

where $\hat{\pi}_s \in \Delta_A$ denote the policy of the empirical RMPDs at state $s$, $\pi_s^* \in \Delta_A$ the optimal policy given $s$ of the true RMPDs, $\|.\|_*$ the dual norm and $c_4, c_5, c_6$ are universal constant. Note that this result is also true for $TV$ without the term depending on smoothness $C_S$. In addition, we provide the lower bounds for a representative divergence function — $L_\infty$ norm in the following. Note that for classical $L_p$, $C_S = S^{1/q}(p-1)$ and $C_g$ can be lower bounded by 1. A proof is provided in Appendix F.

**Theorem 4** (Lower bound for $s$-rectangularity). *Consider the uncertainty set $\mathcal{U}_{L_\infty}^{\mathsf{s},\widetilde{\sigma}}(\cdot)$ associated with the $L_\infty$ norm. Consider any tuple $(S, A, \gamma, \sigma, \varepsilon)$ and $0 < c_0 \leq \frac{1}{8}$ being any small enough positive constant, where $\gamma \in \left[\frac{1}{2}, 1\right)$, and $\varepsilon \in \left(0, \frac{c_0}{256(1-\gamma)}\right]$. Correspondingly, we denote the accessible maximal uncertainty level for $\mathcal{U}_{L_\infty}^{\mathsf{s},\widetilde{\sigma}}(\cdot)$ as $\sigma_{\max}^\infty := \max_{p_1, p_1 \in \Delta(\mathcal{S})^A} \|p_1 - p_2\|_\infty = 1$. Then we can construct a collection of infinite-horizon RMDPs $\mathcal{M}_{L_\infty}$ defined by the uncertainty set with $\mathcal{U}_{L_\infty}^{\mathsf{s},\widetilde{\sigma}}(\cdot)$ so that for any $\sigma \in (0, \sigma_{\max}^\infty(1-c_0)]$, and any dataset with in total $N_{\mathsf{all}}$ independent samples for all state-action pairs over the nominal transition kernel (for any RMDP inside $\mathcal{M}_{L_\infty}$), one has*

$$\inf_{\widehat{\pi}} \max_{\mathcal{M} \in \mathcal{M}_{L_\infty}} \left\{ \mathbb{P}_{\mathcal{M}}\left( \max_{s \in \mathcal{S}} \left[ V^{\star,\sigma}(s) - V^{\widehat{\pi},\sigma}(s) \right] > \varepsilon \right) \right\} \geq \frac{1}{8}, \tag{22}$$

*provided that for $c_8$ is a universal positive constant,*

$$N_{\mathsf{all}} \leq \frac{c_8 SA}{(1-\gamma)^2 \max\{1-\gamma, \tilde{\sigma}\}\varepsilon^2}, \tag{23}$$

*with $\mathbb{P}_{\mathcal{M}}$ the probability when the RMDP is $\mathcal{M}$, and the infimum is taken over all estimators $\widehat{\pi}$.*

Now we can present some implications of Theorem 3 and Theorem 4.

• **Robust MDPs with $s$-rectangularity are at least as easy as $sa$-rectangularity.** Theorem 3 shows that the sample complexity of solving RMDPs with $s$-rectangularity does not exceed the order of $\widetilde{O}\left(\frac{SA}{(1-\gamma)^2 \max\{1-\gamma, C_g \sigma\}\varepsilon^2}\right)$. This matches the sample complexity for $sa$-rectangularity (cf. (18)) and indicates that although $s$-rectangular RMDPs are of a more complicated formulation, solving $s$-rectangular RMDPs is at least as easy as solving $sa$-rectangular RMDPs in terms of the sample complexity. In addition to the worst-case sample complexity upper bound, Theorem 3 also provides a data and instance-dependent sample complexity upper bound for $s$-rectangular RMDPs (cf. in (20)).Taking the divergence function $\|\cdot\| = L_p$ for instance, the data and instance-dependent sample complexity upper bound is

$$\begin{cases} \widetilde{O}\left(\frac{SA}{(1-\gamma)^2\varepsilon^2} \frac{1}{\max\{1-\gamma, \sigma\}}\right) & \text{if } \widehat{\pi}_s(a \mid s) = \pi_s^*(a \mid s) = \frac{1}{A}, \quad \forall (s, a) \in \mathcal{S} \times \mathcal{A} \\ \widetilde{O}\left(\frac{SA}{(1-\gamma)^2\varepsilon^2} \frac{1}{\max\{1-\gamma, \sigma A^{1/p}\}}\right) & \text{if } \|\widehat{\pi}_s(\cdot \mid s)\|_0 = \|\pi_s^*(\cdot \mid s)\|_0 = 1, \quad \forall s \in \mathcal{S}. \end{cases}$$

where $\|.\|_0$ corresponds to the total number of nonzero elements in a vector.The intuition beyond this theorem is that when the policy becomes proportional to uniform, the uncertainty budget of the $s$-rectangular MDPs is equally spread into all actions, and we retrieve the $sa$-rectangular case. When the policy becomes deterministic, all the uncertainty budget concentrates on one action. In this case, most of the actions are not robust except one, and the problem is simpler than classical MDP for this only specific action. An illustration of this result can be found in Fig. 2.

• **Comparisons with prior arts.** In Figure 1, we illustrate the comparisons with Clavier et al. [2023] which use $L_p$ norms functions belonging to the class of general norms considered in this work. We do not compare in this section to Yang et al. [2022a] as it is not anymore state-of-the-art with regard to the work of Clavier et al. [2023]. In particular, the latest achieves in the $s$-rectangular case at sample complexity of $\widetilde{O}\left(\frac{SA}{(1-\gamma)^3\varepsilon^2}\right)$ in the regime where $\tilde{\sigma} \lesssim 1 - \gamma$. In this regime, our result is the same but more general but in the regime where $\tilde{\sigma} \gtrsim 1 - \gamma$, they achieve sample complexity of $\widetilde{O}\left(\frac{SA}{(1-\gamma)^4\varepsilon^2}\right)$ which is bigger than our result $\widetilde{O}\left(\frac{SA}{(1-\gamma)^2 \max\{1-\gamma, \tilde{\sigma}\}\varepsilon^2}\right)$ by a factor at least $\frac{1}{1-\gamma}$.

## 5   Conclusion

This work refined sample complexity bounds to learn robust Markov decision processes when the uncertainty set is characterized by an general $L_p$ metric, assuming the presence of a generative model. Our findings not only strengthen the current knowledge by improving both the upper and lower bounds, but also highlight that learning $s$-rectangular MDPs is less challenging in terms of sample complexity compared to classical $sa$-rectangular MDPs. This work is the first to provide results with a minimax bound, as prior results concerning $s$-rectangular cases were not minimax optimal. Additionally, we

have established the minimax sample complexity for RMDPs using a general $L_p$ norm, demonstrating that it is never larger than that required for learning standard MDPs. Our research identifies potential avenues for future work, such as exploring the characterization of tight sample complexity for RMDPs under a broader family of uncertainty sets, such as those defined by $f$-divergence. It would be highly desirable for a more unified theoretical foundation, as the distance between probability measures is more natural to define using divergence. Moreover, it would be interesting to focus on the finite-horizon Setting and linear setting, as our current analytical framework opens the door for potential extensions to address finite-horizon RMDPs. Such an extension would contribute to a more comprehensive understanding of tabular cases. Finally, the case of linear MDPs would be interesting to explore.

## 6 Acknowledgements

Fondation Mathématique Jacques Hadamard supported this work during Pierre Clavier's visiting the Californian Institute of Technology. Pierre Clavier has been supported by a grant from Region Ile-de-France; DIM Math Innov. The work of L. Shi is supported in part by the Resnick Institute and Computing, Data, and Society Postdoctoral Fellowship at California Institute of Technology. The work of E. Mazumdar is supported in part from NSF-2240110. The work of A. Wierman is supported in part from CNS-2146814, CPS-2136197, CNS-2106403, and NGSDI-2105648.

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

## A    Other related works

Here we provide additional discussion of related work that could not be fit into the main paper due to space considerations. We limit our discussions to the tabular setting with finite state and action spaces provable RL algorithms.

**Classical reinforcement learning with finite-sample guarantees.**    A recent surge in attention for RL has leveraged the methodologies derived from high-dimensional probability and statistics to analyze RL algorithms in non-asymptotic scenarios. Substantial efforts have been devoted to conducting non-asymptotic sample analyses of standard RL in many settings. Illustrative instances encompass investigations employing Probably Approximately Correct (PAC) bonds in the context of *generative model* settings [Kearns and Singh, 1999, Beck and Srikant, 2012, Li et al., 2022a, Chen et al., 2020, Azar et al., 2013b, Sidford et al., 2018, Agarwal et al., 2020, Li et al., 2023a,b, Wainwright, 2019] and the *online setting* via both in PAC-base or regret-based analyses [Jin et al., 2018, Bai et al., 2019, Li et al., 2021, Zhang et al., 2020b, Dong et al., 2019, Jin et al., 2020, Li et al., 2023c, Jafarnia-Jahromi et al., 2020, Yang et al., 2021] and finally *offline setting* [Rashidinejad et al., 2021, Xie et al., 2021, Yin et al., 2021, Shi et al., 2022, Li et al., 2022b, Jin et al., 2021, Yan et al., 2022].

**Robustness in reinforcement learning.**    Reinforcement learning has had notable achievements but has also exhibited significant limitations, particularly when the learned policy is susceptible to deviations in the deployed environment due to perturbations, model discrepancies, or structural modifications. To address these challenges, the idea of robustness in RL algorithms has been studied. Robustness could concern uncertainty or perturbations across different Markov Decision Processes (MDPs) components, encompassing reward, state, action, and the transition kernel. Moos et al. [2022] gives a recent overview of the different work in this field.

The distributionally robust MDP (RMDP) framework has been proposed [Iyengar, 2005] to enhance the robustness of RL has been proposed. In addition to this work, various other research efforts, including, but not limited to, Zhang et al. [2020a, 2021], Han et al. [2022], Clavier et al. [2022], Qiaoben et al. [2021], explore robustness regarding state uncertainty. In these scenarios, the agent's policy is determined on the basis of perturbed observations generated from the state, introducing restricted noise, or undergoing adversarial attacks. Finally, robustness considerations extend to uncertainty in the action domain. Works such as Tessler et al. [2019], Tan et al. [2020] consider the robustness of actions, acknowledging potential distortions introduced by an adversarial agent.

Given the focus of our work, we provide a more detailed background on progress related to distributionally robust RL. The idea of distributionally robust optimization has been explored within the context of supervised learning [Rahimian and Mehrotra, 2019, Gao, 2020, Duchi and Namkoong, 2018, Blanchet and Murthy, 2019] and has also been extended to distributionally robust dynamic programming and Distributionally Robust Markov Decision Processes (DRMDPs) such as in [Iyengar, 2005, Xu and Mannor, 2012, Wolff et al., 2012, Kaufman and Schaefer, 2013, Ho et al., 2018, Smirnova et al., 2019, Ho et al., 2021, Goyal and Grand-Clement, 2022, Derman and Mannor, 2020, Tamar et al., 2014, Badrinath and Kalathil, 2021]. Despite the considerable attention received, both empirically and theoretically, most previous theoretical analyses in the context of RMDPs adopt an asymptotic perspective [Roy et al., 2017] or focus on planning with exact knowledge of the uncertainty set [Iyengar, 2005, Xu and Mannor, 2012, Tamar et al., 2014]. Many works have focused on the finite-sample performance of verifiable robust Reinforcement Learning (RL) algorithms. These investigations encompass various data generation mechanisms and uncertainty set formulations over the transition kernel. Closely

related to our work, various forms of uncertainty sets have been explored, showcasing the versatility of approaches. Divergence such as Kullback-Leibler (KL) divergence is another prevalent choice, extensively studied by Yang et al. [2022a], Panaganti and Kalathil [2022], Zhou et al. [2021], Shi and Chi [2022], Xu et al. [2023], Wang et al. [2023], Blanchet et al. [2023], who investigated the sample complexity of both model-based and model-free algorithms in simulator or offline settings. Xu et al. [2023] considered various uncertainty sets, including those associated with the Wasserstein distance. The introduction of an R-contamination uncertainty set Wang and Zou [2021], has been proposed to tackle a robust Q-learning algorithm for the online setting, with guarantees analogous to standard RL. Finally, the finite-horizon scenario has been studied by Xu et al. [2023], Dong et al. [2022] with finite-sample complexity bounds for (RMDPs) using TV and $\chi^2$ divergence. More broadly, other related topics have been explored, such as the iteration complexity of policy-based methods [Li et al., 2022c, Kumar et al., 2023], and regularization-based robust RL [Yang et al., 2023]. Finally, Badrinath and Kalathil [2021] examined a general $sa$-rectangular form of the uncertainty set, proposing a model-free algorithm for the online setting with linear function approximation to address large state spaces.

# B  Further discussions of Theorem 1 and Theorem 3

- *What norms are included in the Definition 1?*  In our upper bound result Theorems 3 and 1, we upper bound the sample complexity for $C^2$ norms and TV. The set of $C^2$ smooth norm is very large as it includes all, $L_p$ norm, weighted, rescaled $L_p$ norms for $p \geq 2$. Weighted norms can be useful in practice, to get more weights on dangerous specific states in Robust MDPs formulation such as in Russel et al. [2019]. Moreover, note that our result can generalize to metric or pseudo metric (which are not homogeneous ie $\|\lambda\| = |\lambda| \, \|x\| \, \forall x \in \mathbb{R}^n, \lambda \in \mathbb{R}$) with norms of the form $x \mapsto \phi^{-1}(\sum_{k=1}^n \phi(|x_k|))$ with $\phi$ a convex incising function such as the norm is still positive, definite positive. Choosing $\phi(x) = x^p$ leads to the $L_p$ norms.

- *Assumptions on $\gamma$ in Theorems 1 and 3, and Assumptions on $\gamma$ for lower bound.*  When $\gamma$ is small (e.g., $\gamma \in (0, \frac{1}{2}]$ leads to the effective horizon length is at most 2), the sequential structure almost disappears and is much less of interest for RL community. So people Li et al. [2023b] Yan et al. [2023] usually focus on reasonable range $\gamma \in (c, 1)$ for some small positive constant $c$, such as $\gamma \in [\frac{1}{2}, 1)$. However, the theorems can be directly extended to a broader range of $\gamma \in (c, 1)$ along with $c$ as small as desired so that almost cover the full range $(0, 1)$.

- *Why final results on s depend on $\hat{\pi}$?*

  Theorem 3 is $\hat{\pi}$ data dependent which is randomness-dependent measure. However, taking the minimum of this quantity leads to the same bound as is $sa$-rectangular, so to illustrate that it is possible to get tighter bounds for $s$-rectangular with instance-dependent RMDPs, we decide to write also randomness-dependent quantity, while the less tight upper bound is written also in the theorem, taking the first term in the min operator in (21).

- *Why our results are still true for $TV$?*  Theorems 1 and 3 are stated for $C^2$ smooth norms, however, our result is still true for $TV$ which is not $C^2$ as in this specific case, the dual of the optimization problem becomes a $1-$dimensional problem. In this case in the main concentration lemma 8, the additional term involving smoothness term denoted $C_S$ is not present and the bound is simpler as is not required this additional term.

- *Why burn-in or sufficiently small $\epsilon$ condition is not too restrictive?*  The burn-in term in Th. 1 and 3 is proportional to $1/\epsilon$ where the "sample complexity" term is proportional to $1/\epsilon^2$. The smooth term depending on $C_S$ or burn-in is then not too large for sufficiently small $\epsilon$ compared to the other term, which will give final sample complexity.

- *Why this is not extendable to $f$-divergence currently?* The f-divergence as a distinct family of divergence is beyond the scope of this paper. Current proof for arbitrary norms cannot be directly extended since the key phenomenon of shrinking range of the robust value function has not been verified for $f$-divergence yet, while it is promising as an interesting future direction.

# C Preliminaries

These quantities appear in the dual formulation of the robust optimization problem and more preciously the dual span semi norm $\mathrm{sp}(.)_*$ note that for $L_2$, we retrieve the classical mean with the definition of $\omega$) With slight abuse of notation, we denote 0 (resp. 1) as the all-zero (resp. all-one) vector. We then introduce the notation $[T] := \{1, \cdots, T\}$ for any positive integer $T > 0$. Then, for all $1 \le i \le n$, for two vectors $x = [x_i]_{1 \le i \le n}$ and $y = [y_i]_{1 \le i \le n}$, the notation $x \le y$ (resp. $x \ge y$) means $x_i \le y_i$ (resp. $x_i \ge y_i$) . Finally, for any vector $x$, the notation is overloaded by letting $x^{\circ 2} = [x(s,a)^2]_{(s,a) \in \mathcal{S} \times \mathcal{A}}$ (resp. $x^{\circ 2} = [x(s)^2]_{s \in \mathcal{S}}$), Finally, we drop the subscript $\|.\|$ to write $\mathcal{U}^{\sigma}_{\|.\|}(\cdot) = \mathcal{U}^{\sigma}(\cdot)$ for both $sa$- and $s$- rectangular assumptions such that we write uncertainty set in the for $sa$-rectangular case $\mathcal{U}^{\mathsf{sa},\sigma}(.)$ or $\mathcal{U}^{\mathsf{s},\tilde{\sigma}}(.)$ in the $s$-rectangular assumptions.

**Matrix and Vector Notations.** We define the following notation.

- $r \in \mathbb{R}^{SA}$ the reward function, such that $r_{(s,a)} = r(s,a)$ for all $(s,a) \in \mathcal{S} \times \mathcal{A}$.

- $P^0 \in \mathbb{R}^{SA \times S}$ the nominal transition kernel matrix using $P^0_{s,a}$ as the $(s,a)$-th row.

- $\widehat{P}^0 \in \mathbb{R}^{SA \times S}$ the estimated nomimal transition kernel matrix with $\widehat{P}^0_{s,a}$ as the $(s,a)$-th row.

- $\Pi^{\pi} \in \{0,1\}^{S \times SA}$ the projection matrix associated with a policy $\pi$

$$
\Pi^{\pi} = \begin{pmatrix} 1^{\top}_{\pi(1)} & 0^{\top} & \cdots & 0^{\top} \\ 0^{\top} & 1^{\top}_{\pi(2)} & \cdots & 0^{\top} \\ \vdots & \vdots & \ddots & \vdots \\ 0^{\top} & 0^{\top} & \cdots & 1^{\top}_{\pi(S)} \end{pmatrix}, \tag{24}
$$

where $1^{\top}_{\pi(1)}, 1^{\top}_{\pi(2)}, \dots, 1^{\top}_{\pi(S)} \in \mathbb{R}^A$ are simplex vector such as

$$
1^{\top}_{\pi(1)} = (\pi(a_1|s_1), \pi(a_2|s_1), ..., \pi(a_A|s_1)).
$$

- The two matrices $P^V \in \mathbb{R}^{SA \times S}$, $\widehat{P}^V \in \mathbb{R}^{SA \times S}$ represent the probability transition kernel in the uncertainty set that leads to the worst-case value for any vector $V \in \mathbb{R}^S$. Moreover, the quantities $P^V_{s,a}$ (resp. $\widehat{P}^V_{s,a}$) stands for the $(s,a)$-th row of the transition matrix $P^V$ (resp. $\widehat{P}^V$). In $sa$-rectangular case , the $(s,a)$-th rows of these transition matrices are defined as

$$
P^V_{s,a} = \mathrm{argmin}_{\mathcal{P} \in \mathcal{U}^{\mathsf{sa},\sigma}(P^0_{s,a})} \mathcal{P}V, \qquad \text{and} \qquad \widehat{P}^V_{s,a} = \mathrm{argmin}_{\mathcal{P} \in \mathcal{U}^{\mathsf{sa},\sigma}(\widehat{P}^0_{s,a})} \mathcal{P}V. \tag{25a}
$$

Moreover, the shorthand notation defined below is used

$$
P^{\pi,V}_{s,a} := P^{V^{\pi,\sigma}}_{s,a} = \mathrm{argmin}_{\mathcal{P} \in \mathcal{U}^{\mathsf{sa},\sigma}(P^0_{s,a})} \mathcal{P}V^{\pi,\sigma}, \tag{25b}
$$

$$
P^{\pi,\widehat{V}}_{s,a} := P^{\widehat{V}^{\pi,\sigma}}_{s,a} = \mathrm{argmin}_{\mathcal{P} \in \mathcal{U}^{\mathsf{sa},\sigma}(P^0_{s,a})} \mathcal{P}\widehat{V}^{\pi,\sigma}, \tag{25c}
$$

$$
\widehat{P}^{\pi,V}_{s,a} := \widehat{P}^{V^{\pi,\sigma}}_{s,a} = \mathrm{argmin}_{P \in \mathcal{U}^{\mathsf{sa},\sigma}(\widehat{P}^0_{s,a})} PV^{\pi,\sigma}, \tag{25d}
$$

$$
\widehat{P}^{\pi,\widehat{V}}_{s,a} := \widehat{P}^{\widehat{V}^{\pi,\sigma}}_{s,a} = \mathrm{argmin}_{P \in \mathcal{U}^{\mathsf{sa},\sigma}(\widehat{P}^0_{s,a})} P\widehat{V}^{\pi,\sigma}. \tag{25e}
$$

In the following, we define the corresponding probability transition matrices which are denoted by $P^{\pi,V} \in \mathbb{R}^{SA \times S}$, $P^{\pi,\widehat{V}} \in \mathbb{R}^{SA \times S}$, $\widehat{P}^{\pi,V} \in \mathbb{R}^{SA \times S}$ and $\widehat{P}^{\pi,\widehat{V}} \in \mathbb{R}^{SA \times S}$.

- Using the projection over $\pi$, the matrices $P^{\pi} \in \mathbb{R}^{S \times S}$, $\widehat{P}^{\pi} \in \mathbb{R}^{S \times S}$, $\underline{P}^{\pi,V} \in \mathbb{R}^{S \times S}$, $\underline{P}^{\pi,\widehat{V}} \in \mathbb{R}^{S \times S}$, $\underline{\widehat{P}}^{\pi,V} \in \mathbb{R}^{S \times S}$ and $\underline{\widehat{P}}^{\pi,\widehat{V}} \in \mathbb{R}^{S \times S}$ represent probability transition matrices w.r.t. policy $\pi$.

$$
P^{\pi} := \Pi^{\pi} P^0, \qquad \widehat{P}^{\pi} := \Pi^{\pi} \widehat{P}^0, \qquad \underline{P}^{\pi,V} := \Pi^{\pi} P^{\pi,V}, \qquad \underline{P}^{\pi,\widehat{V}} := \Pi^{\pi} P^{\pi,\widehat{V}},
$$

$$
\underline{\widehat{P}}^{\pi,V} := \Pi^{\pi} \widehat{P}^{\pi,V}, \qquad \text{and} \qquad \underline{\widehat{P}}^{\pi,\widehat{V}} := \Pi^{\pi} \widehat{P}^{\pi,\widehat{V}}. \tag{26}
$$

For $s$-rectangular, we will use the same notation for these transition matrices. Finally, we denote $P^{\pi}_s$ as the $s$-th row of the transition matrix $P^{\pi}$.

- $r_\pi \in \mathbb{R}^S$ is the reward function restricted to the actions chosen by $\pi$, $r_\pi = \Pi^\pi r$.
- $\mathrm{Var}_P(V) \in \mathbb{R}^{SA}$ is the variance for a given transition kernel $P \in \mathbb{R}^{SA \times S}$ and vector $V \in \mathbb{R}^S$, we denote the $(s,a)$-th row of $\mathrm{Var}_P(V)$ as

$$\mathsf{Var}_P(s,a) := \mathrm{Var}_{P_{s,a}}(V). \tag{27}$$

## C.1 Additional definitions and basic facts

For any norm smooth $\|.\|$ introduced in 1, we define the span semi norm as

**Definition 2** (Span semi norm). *Given any norm $\|\cdot\|$, we define the span semi norm as:* $\mathrm{sp}(x) = \min_{\omega \in \mathbb{R}} \|v - \omega\mathbf{1}\|$ *and the generalized mean as* $\omega(x) := \arg\min_{\omega \in \mathbb{R}} \|x - \omega\mathbf{1}\|$.

Let vector $P \in \mathbb{R}^{1 \times S}$ and vector $V \in \mathbb{R}^S$, we define the variance

$$\mathrm{Var}_P(V) := P(V \circ V) - (PV) \circ (PV). \tag{28}$$

The following lemma bounds the Lipschitz constant of the variance function.

**Lemma 1.** *(Shi et al. [2023], Lemma 2) Assuming $0 \le V_1, V_2 \le \frac{1}{1-\gamma}$ which obey $\|V_1 - V_2\|_\infty \le x$, then for $P \in \Delta(S)$, one has*

$$|\mathrm{Var}_P(V_1) - \mathrm{Var}_P(V_2)| \le \frac{2x}{(1-\gamma)}. \tag{29}$$

**Lemma 2.** *[Panaganti and Kalathil, 2022, Lemma 6] Consider any $\delta \in (0,1)$. For any fixed policy $\pi$ and fixed value vector $V \in \mathbb{R}^S$, one has with probability at least $1 - \delta$,*

$$\left| \sqrt{\mathrm{Var}_{\widehat{P}^\pi}(V)} - \sqrt{\mathrm{Var}_{P^\pi}(V)} \right| \le \sqrt{\frac{2\|V\|_\infty^2 \log(\frac{2SA}{\delta})}{N}}\mathbf{1}.$$

## C.2 Empirical robust MDP $\widehat{\mathcal{M}}_{\mathrm{rob}}$ Bellman equations

We define the robust MDP $\widehat{\mathcal{M}}_{\mathrm{rob}} = \{\mathcal{S}, \mathcal{A}, \gamma, \mathcal{U}^\sigma(\widehat{P}^0), r\}$ based on the estimated nominal distribution $\widehat{P}^0$ in (11). Then, we denote the associated robust value function (resp. robust Q-function) are $\widehat{V}^{\pi,\sigma}$ (resp. $\widehat{Q}^{\pi,\sigma}$) qnd we can notice that that $\widehat{Q}^{\star,\sigma}$ is the unique-fixed point of $\widehat{\mathcal{T}}^\sigma(\cdot)$ (see Lemma C.3), the empirical robust Bellman operator constructed using $\widehat{P}^0$. Finally, similarly to (9), for $\widehat{\mathcal{M}}_{\mathrm{rob}}$, the Bellman's optimality principle gives the following *robust Bellman consistency equation* (resp. *robust Bellman optimality equation*) for $sa$-rectangular assumptions:

$$\widehat{Q}^{\pi,\sigma}(s,a) = r(s,a) + \gamma \inf_{\mathcal{P} \in \mathcal{U}^{\mathsf{sa},\sigma}(\widehat{P}^0_{s,a})} \mathcal{P}\widehat{V}^{\pi,\sigma}, \tag{30a}$$

$$\widehat{Q}^{\star,\sigma}(s,a) = r(s,a) + \gamma \inf_{\mathcal{P} \in \mathcal{U}^{\mathsf{sa},\sigma}(\widehat{P}^0_{s,a})} \mathcal{P}\widehat{V}^{\star,\sigma}. \tag{30b}$$

Using matrix notation, we can write the robust Bellman consistency equations as

$$Q^{\pi,\sigma} = r + \gamma \inf_{\mathcal{P} \in \mathcal{U}^{\mathsf{sa},\sigma}(P^0)} \mathcal{P}V^{\pi,\sigma} \quad \text{and} \quad \widehat{Q}^{\pi,\sigma} = r + \gamma \inf_{\mathcal{P} \in \mathcal{U}^{\mathsf{sa},\sigma}(\widehat{P}^0)} \mathcal{P}\widehat{V}^{\pi,\sigma}, \tag{31}$$

which imply

$$V^{\pi,\sigma} = r_\pi + \gamma\Pi^\pi \inf_{\mathcal{P} \in \mathcal{U}^{\mathsf{sa},\sigma}(P^0)} \mathcal{P}V^{\pi,\sigma} \overset{\text{(i)}}{=} r_\pi + \gamma\underline{P}^{\pi,V}V^{\pi,\sigma},$$

$$\widehat{V}^{\pi,\sigma} = r_\pi + \gamma\Pi^\pi \inf_{\mathcal{P} \in \mathcal{U}^{\mathsf{sa},\sigma}(\widehat{P}^0)} \mathcal{P}\widehat{V}^{\pi,\sigma} \overset{\text{(ii)}}{=} r_\pi + \gamma\underline{\widehat{P}}^{\pi,\widehat{V}}\widehat{V}^{\pi,\sigma}, \tag{32}$$

where (i) and (ii) hold by the definitions in (24), (25) and (26). For $s$-rectangular, we can define the same notation, removing $a$ subscript:

$$V^{\pi,\sigma} = r_\pi + \gamma\Pi^\pi \inf_{\mathcal{P} \in \mathcal{U}^{\mathsf{s},\widetilde{\sigma}}(P^0)} \mathcal{P}V^{\pi,\sigma} \overset{\text{(i)}}{=} r_\pi + \gamma\underline{P}^{\pi,V}V^{\pi,\sigma},$$

$$\widehat{V}^{\pi,\sigma} = r_\pi + \gamma\Pi^\pi \inf_{\mathcal{P} \in \mathcal{U}^{\mathsf{s},\widetilde{\sigma}}(\widehat{P}^0)} \mathcal{P}\widehat{V}^{\pi,\sigma} \overset{\text{(ii)}}{=} r_\pi + \gamma\underline{\widehat{P}}^{\pi,\widehat{V}}\widehat{V}^{\pi,\sigma},. \tag{33}$$

## C.3  Properties of the robust Bellman operator and dual representation

The robust Bellman operator (cf. (10)) shares the $\gamma$-contraction property of the standard Bellman operator as:

**[Iyengar, 2005, Theorem 3.2]**  Given $\gamma \in [0, 1)$, the robust Bellman operator $\mathcal{T}^\sigma(\cdot)$ (cf. (10)) is a $\gamma$-contraction w.r.t. $\|\cdot\|_\infty$. More formally, for any $Q_1, Q_2 \in \mathbb{R}^{SA}$ s.t. $Q_1(s,a), Q_2(s,a) \in \left[0, \frac{1}{1-\gamma}\right]$ for all $(s,a) \in \mathcal{S} \times \mathcal{A}$, one has

$$\|\mathcal{T}^\sigma(Q_1) - \mathcal{T}^\sigma(Q_2)\|_\infty \leq \gamma \|Q_1 - Q_2\|_\infty. \tag{34}$$

It can be also shown that, $Q^{\star,\sigma}$ is the unique fixed point of $\mathcal{T}^\sigma(\cdot)$ obeying $0 \leq Q^{\star,\sigma}(s,a) \leq \frac{1}{1-\gamma}$ for all $(s,a) \in \mathcal{S} \times \mathcal{A}$.

One of the main contributions is to derive the dual form of optimization problem using arbitrary norms. These lemma take ideas from Iyengar [2005] and are adapted to arbitrary norms and not only $TV$ distance.

**Dual equivalence of the robust Bellman operator.**  Fortunately, the robust Bellman operator can be evaluated efficiently by resorting to its dual formulation, and this idea is central in all proofs for RMPDs. Dual formulation of RMDPs have been introduced in [Iyengar, 2005] but the proof was done uniquely for the $TV$ and the $\chi^2$ case. Before continuing, for any $V \in \mathbb{R}^S$, we denote $[V]_\alpha$ as its clipped version by some non-negative vector $\alpha$, namely,

$$[V]_\alpha(s) := \begin{cases} \alpha, & \text{if } V(s) > \alpha(s), \\ V(s), & \text{otherwise.} \end{cases} \tag{35}$$

Defining the gradient of $P \mapsto \|P\|$ as $\nabla \|P\|$, $\lambda > 0$, a positive scalar and $\omega$ is the generalized mean defined as the argmin in the definition of the span semi norm in Def.2, we derive two optimization lemmas.

**Lemma 3** (Strong duality using norm $\|\cdot\|$ in the $sa$-rectangular case.)**.** *Consider any probability vector $P \in \Delta(\mathcal{S})$ and any fixed uncertainty level $\sigma$, we abbreviate the notation of the uncertainty set $\mathcal{U}_{\|.\|}^{\mathsf{sa},\sigma}(P)$ (cf. (3)) as $\mathcal{U}^{\mathsf{sa},\sigma}(P)$. For any vector $V \in \mathbb{R}^S$ obeying $V \geq 0$, recalling the definition of $[V]_\alpha$ in (35), one has*

$$\inf_{\mathcal{P} \in \mathcal{U}^{\mathsf{sa},\sigma}(P)} \mathcal{P}V = \max_{\mu_P^{\lambda,\omega} \in \mathcal{M}_P^{\lambda,\omega}} \left\{ P(V - \mu_P^{\lambda,\omega}) - \sigma\left(\mathrm{sp}((V - \mu_P^{\lambda,\omega}))_*\right) \right\}. \tag{36}$$

$$= \max_{\alpha_P^{\lambda,\omega} \in \mathrm{A}_P^{\lambda,\omega}} \left\{ P[V]_{\alpha_P^{\lambda,\omega}} - \sigma\left(\mathrm{sp}([V]_{\alpha_P^{\lambda,\omega}})_*\right) \right\} \tag{37}$$

*where $\mathrm{sp}()_*$ is defined in Def..2. Here, the two auxiliary variational family $\mathrm{A}_P^{\lambda,\omega}, \mathcal{M}_P^{\lambda,\omega}$ are defined as below:*

$$\mathrm{A}_P^{\lambda,\omega} = \{\alpha_P^{\lambda,\omega} : \alpha_P^{\lambda,\omega}(s) = \omega + \lambda |\nabla \|P\|\,|(s) : \lambda > 0, w > 0, P \in \Delta(S), \alpha_P^{\lambda,\omega} \in \left[0, \frac{1}{1-\gamma}\right]^S \} \tag{38}$$

$$\mathcal{M}_P^{\lambda,\omega} = \{\mu_P^{\lambda,\omega} = V - \alpha_P^{\lambda,\omega}, \lambda, \omega \in \mathbb{R}^+, P \in \Delta(S), \mu \in \mathbb{R}_+^S, \mu_P^{\lambda,\omega} = \left[0, \frac{1}{1-\gamma}\right]^S \}. \tag{39}$$

$$\tag{40}$$

For $L_1$ or $TV$, case , the vector $\alpha_P^{\lambda,\omega}$ reduces to a 1 dimensional scalar such as $\alpha \in [0, 1/(1-\gamma)]$.

*Proof.*

$$\inf_{\mathcal{P} \in \mathcal{U}^{\mathsf{sa},\sigma}(P)} \mathcal{P}V = \inf_{\{\mathcal{P}: \mathcal{P} \in \Delta_s, \|\mathcal{P} - P\| \leq \sigma\}} \sum_{s'} \mathcal{P}(s')V(s')$$

$$= PV + \inf_{\{y: \|y\| \leq \sigma, 1y = 0, y \geq -P\}} \sum_{s'} y(s')V(s')$$

where we use the change of variable $y(s') = \mathcal{P}(s') - P(s')$ for all $s' \in \mathcal{S}$. Then the Lagrangian function of the above optimization problem can be written as follows:

$$\inf_{\mathcal{P}\in\mathcal{U}_{s,a}^\sigma(P)} \mathcal{P}V = PV + \sup_{\mu\geq 0, \nu\in\mathbb{R}} \inf_{\{y:\|y\|\leq\sigma\}} -\sum_{s'}\mu(s)P(s') + \sum_{s'}(y(s')(V(s')-\mu(s')-\nu)) \quad (41)$$

$$\overset{(a)}{=} PV + \sup_{\mu\geq 0, \nu\in\mathbb{R}} -\sum_{s'}\mu(s')P(s') - \sigma\|(V(s')-\mu(s')-\nu\mathbf{1})\|_* \quad (42)$$

$$\overset{(b)}{=} \sup_{\mu\geq 0} P(V-\mu) - \sigma\mathrm{sp}(V-\mu)_* \quad (43)$$

where $\mu \in \mathbb{R}_+^\mathcal{S}$, $\nu \in \mathbb{R}$ are Lagrangian variables, (a) is true using the equality case of Cauchy-Swartz inequality for dual norm Yang [1991], and (b) is due to is the definition of the span semi-norm (see (C)). The value that maximizes the inner maximization problem in (42) in $\omega(V,\mu)$ is the generalized-mean by definition denoted with abbreviate notation $\omega$. If the norm is differentiable, then we have that the equality (a) comes from the generalized Holder's inequality for arbitrary norms Yang [1991], namely, defining $z = (V - \mu - \omega)$, it satisfies

$$z = \|z\|_* \nabla \|y\| \quad (44)$$

The quantity $\nu$ is replaced by the generalized mean for equality in (b) while (44) comes from Yang [1991]. Using complementary slackness Karush [2013]stackness let $\mathcal{B} = \{s \in \mathcal{S} : \mu(s) > 0\}$

$$\forall s \in \mathcal{B}: \quad y^*(s) = -P(s), \quad (45)$$

which leads to the following equality by plugging the previous (45) in (44) and defining $z^* = V - \mu^* - \omega$:

$$\forall s \in \mathcal{B}, \quad z^*(s) = \|z^*\|_* \nabla \|P\|(s) \quad (46)$$

or

$$\forall s \in \mathcal{B}, \quad V(s) - \mu^*(s) = \omega + \lambda\nabla\|P\|(s) \hat{=} \alpha_P^{\lambda,\omega} \quad (47)$$

by letting $\lambda = \|z^*\|_* \in \mathbb{R}^+$. Note that here the hypothesis of 1 are use and especially separability is needed to ensure that for $s \in \mathcal{B}$, $\nabla\|y\| = \nabla\|P\|$ only depend on $P(s)$ and not on other coordinates, which is true form generalized $L_p$ norms. We can remark that $v - \mu^*$ is $P$ dependent, but if $P$ is known, the best $\mu^*$ is only determined by one 2 dimensional parameters $\lambda = \|v - \mu^* - \nu\|_*$ and $\omega \in \mathbb{R}^+$. Moreover, when $P$ is fixed, the scalar $\omega$ is a constant is fully determined by $P$, $v$ and $\mu^*$. This is why the quantity defined $\alpha_P^\lambda$ varies through 2 parameter $\lambda$ and $\omega$. Given this observation, we can rewrite the optimization problem as :

$$\sup_{\mu\geq 0} P(V-\mu) - \sigma\mathrm{sp}(V-\mu)_* = \sup_{\mu_P^{\lambda,\omega}\in\mathcal{M}_P^{\lambda,\omega}} P(V-\mu_P^{\lambda,\omega}) - \sigma\mathrm{sp}((V-\mu_P^{\lambda,\omega}))_* \quad (48)$$

$$= \sup_{\alpha_P^{\lambda,\omega}\in A_P^{\lambda,\omega}} P[V]_{\alpha_P^{\lambda,\omega}} - \sigma\mathrm{sp}([V]_{\alpha_P^{\lambda,\omega}})_* \quad (49)$$

where we defined the maximization problem on $\mu$ not in $\mathbb{R}^S$ but at the optimal in the variational family denote $\mathcal{M}_P^{\lambda,\omega} = \{v - \alpha_P^{\lambda,\omega}, (\lambda,\omega) \in \mathbb{R}_+^2, P \in \Delta(S)\}$. We can rewrite the optimization problem in terms of $\alpha_P$ with $[V]_{\alpha_P^{\lambda,\omega}}$ defined in 35. Contrary to the $TV$ case, $\alpha$ is not a scalar but $\alpha_P^{\lambda,\omega}$ belongs to a variational family only determined by two parameter. Note that this lemma is still true writing subgradient and not gradient of $P$. As we assume $C^2$-regularity on norms, the subgradient space of the norm reduce to the singleton of the gradient in our case. $C^2$ smoothness will be needed in concentration part while it is possible to be more general in optimization lemmas. Note that for $TV$ or $L_1$, this lemma holds, but the vector $\alpha_P^{\lambda,\omega}$ reduces to a positive scalar denoted $\alpha$ which is equal to $\|v - \mu^*\|_\infty$ according to Iyengar [2005].

$\square$

**Lemma 4** (Strong duality for the distance induced by the norm $\|\|$ in the $s$-rectangular case.)**.**
*Consider any probability vector $P^\pi := \Pi^\pi P \in \Delta_s$ for $P \in \Delta(S)^\mathcal{A}$, any fixed uncertainty level $\tilde{\sigma}$*

and the uncertainty set $\mathcal{U}^{s,\tilde{\sigma}}_{\|\cdot\|}(P)$, we abbreviate the subscript to use $\mathcal{U}^{s,\tilde{\sigma}}(P) := \mathcal{U}^{s,\tilde{\sigma}}_{\|\cdot\|}(P)$. Then for any vector $V \in \mathbb{R}^S$ obeying $V \geq 0$, recalling the definition of $[V]_\alpha$ in (35), one has

$$\inf_{\mathcal{P} \in \mathcal{U}^{s,\tilde{\sigma}}(P)} \mathcal{P}^\pi V = \sum_a \pi(a|s)\Big( \Big( \max_{\alpha^{\lambda,\omega}_{P_{sa}} \in A^{\lambda,\omega}_{P_{sa}}} P_{sa}[V]_{\alpha^{\lambda,\omega}_{P_{sa}}} - \tilde{\sigma} \|\pi_s\|_* \, \mathrm{sp}([V]_{\alpha^{\lambda,\omega}_{P_{sa}}})_* \Big). \tag{50}$$

with the definition of $\mathrm{sp}()_*$ in C and where the variational family $A^{\lambda,\omega}_P$ is defined as :

$$A^{\lambda,\omega}_P = \{\alpha \in \left[0, 1/(1-\gamma)\right]^S, \alpha = \omega + \lambda|\nabla \|P\|| := \alpha^{\lambda,\omega}_P\} \tag{51}$$

$$\tag{52}$$

with $\omega$ is the generalized mean defined as the argmin in the definition of the span semi norm in 2 and $\lambda, \omega$ a positive scalar. Moreover, for $L_1$ or $TV$, case, the vector $\alpha^{\lambda,\omega}_P$ reduces to a 1 dimensional scalar such as $\alpha \in [0, 1/(1-\gamma)]$.

In the proof of the previous lemma, we decompose this problem $s$-rectangular radius $\tilde{\sigma}$ into $sa$-rectangular sub-problem with respectively radius $\sigma_{sa}$.

*Proof.*

$$\inf_{\mathcal{P}^\pi \in \mathcal{U}^{s,\tilde{\sigma}}(P^\pi)} \mathcal{P}^\pi V = \inf_{\{\sigma_{sa}:\|\sigma_{sa}\| \leq \tilde{\sigma}\}} \inf_{\mathcal{P}' \in \mathcal{U}^{sa,\sigma}(P_{sa})} \sum_a \pi(a|s)\mathcal{P}'V$$

$$\overset{(a)}{=} \sum_a \pi(a|s)P_{sa}V + \min_{\{\sigma_{sa}:\|\sigma_{sa}\| \leq \tilde{\sigma}\}} \sum_a \pi(a|s) \min_{\{y:\|y\| \leq \sigma_{sa}, 1y=0, y \geq -P_{sa}\}} \sum_{s'} y(s')V$$

where we use the change of variable $y(s') = \mathcal{P}_{sa}(s') - P_{sa}(s')$ in (a). Then we case use the previous lemma for $sa$ rectangular assumption, Lemma 3. Then,

$$\min_{\{\sigma_{sa}:\|\sigma_{sa}\| \leq \tilde{\sigma}\}} \sum_a \pi(a|s) \min_{\{y,\|y\| \leq \sigma_{s,a}, 1y=0, y \geq -P_{sa}\}} \sum_{s'} y(s')V$$

$$= \min_{\{\sigma_{sa}:\|\sigma_{sa}\| \leq \tilde{\sigma}\}} \sum_a \pi(a|s) \max_{\mu \geq 0} \Big( -P_{sa}\mu - \sigma_{sa}\mathrm{sp}(V-\mu)_* \Big)$$

$$= \Big( \sum_a \pi(a|s) \max_{\mu \geq 0} \Big\{ (-P_{sa}\mu) - \max_{\{\sigma_{sa}:\|\sigma_{sa}\| \leq \tilde{\sigma}\}} \sum_a \pi(a|s)\sigma\mathrm{sp}(V-\mu)_* \Big\} \Big)$$

$$= \sum_a \pi(a|s) \max_{\mu \geq 0} \Big\{ (-P_{sa}\mu) - \tilde{\sigma} \|\pi_s\|_* \, \mathrm{sp}(V-\mu)_* \Big\}.$$

We can exchange the min and the max as we get concave-convex problems in $\sigma$ and $\mu$ in the second line according to minimax theorem [v. Neumann, 1928] and using Cauchy Swartz inequality which is attained in the last equality. Finally, we obtain:

$$\inf_{\mathcal{P} \in \mathcal{U}^{s,\tilde{\sigma}}(P)} \mathcal{P}^\pi V = \sum_a \pi(a|s)\Big( \max_{\mu \geq 0} P_{sa}(V-\mu) - \tilde{\sigma} \|\pi_s\|_* \, \mathrm{sp}(V-\mu)_* \Big)$$

$$\overset{(a)}{=} \sum_a \pi(a|s)\Big( \max_{\alpha^{\lambda,\omega}_{P_{sa}} \in A^{\lambda,\omega}_{P_{sa}}} P_{sa}[V]_{\alpha^{\lambda,\omega}_{P_{sa}}} - \tilde{\sigma} \|\pi_s\|_* \, \mathrm{sp}([V]_{\alpha^{\lambda,\omega}_{P_{sa}}})_* \Big)$$

where in (a) we use the previous lemma for $sa-$ rectangular case. Note that as we are using $sa$-rectangular case, for $TV$ or $L_1$, this lemma holds, but the vector $\alpha^\lambda_P$ reduces to a positive scalar denoted $\alpha$ which is equal to $\|v - \mu^*\|_\infty$. (See also Iyengar [2005]).

$\square$

# D Proof of the upper bound : Theorem 1 and 3

## D.1 Technical lemmas

We begin with a key lemma concerning the dynamic range of the robust value function $V^{\pi,\sigma}$ (cf. (7)), which produces tighter control when $\sigma$ is large; the proof is deferred to Appendix D.3.1. This lemma allows tighter control compared to Clavier et al. [2023].

**Lemma 5.** *In $sa-$rectangular case (see (3), for any nominal transition kernel $P \in \mathbb{R}^{SA \times S}$, any fixed uncertainty level $\sigma$, and any policy $\pi$, its corresponding robust value function $V^{\pi,\sigma}$ (cf. (7)) satisfies*

$$\mathrm{sp}(V^{\pi,\sigma})_\infty \leq \frac{1}{\gamma \max\{1 - \gamma, C_g \sigma\}} \tag{53}$$

where $C_g = 1/(\min_s \|e_s\|)$ is a geometric constant depending on the geometry of the norm. For example, for $L_p$, norms $p \geq 1$, $C_g \geq 1$ which reduce the sample complexity. In $s$-rectangular case, we obtain a slightly different lemma because of the dependency on $\pi$.

**Lemma 6.** *The infinite span semi norm can be controlled as follows for every $s$ in $s$-rectanuglar case (See (5)):*

$$\mathrm{sp}(V^{\pi,\sigma})_\infty \leq \frac{1}{\gamma \max\{1 - \gamma, \|\pi_s\|_* C_g \tilde{\sigma}\}} \leq \frac{1}{\gamma \max\{1 - \gamma, \min_s \|\pi_s\|_* C_g \tilde{\sigma}\}} \tag{54}$$

where $C_g = \frac{1}{\min_s \|e_s\|}$ is a geometric constant depending on the geometry of the norm. These lemmas are required to get tight bounds for the sample complexity. The main difference between $sa$- and $s$-rectangular case is that we have an extra dependency on $\|\pi_s\|_*$, which represents how stochastic the policy can be in $s$ rectangular MDPs.

**Lemma 7.** *Consider an MDP with transition kernel matrix $P$ and reward function $0 \leq r \leq 1$. For any policy $\pi$ and its associated state transition matrix $P_\pi := \Pi^\pi P$ and value function $0 \leq V^{\pi,P} \leq \frac{1}{1-\gamma}$ (cf. (1)), one has for $sa$- and $s$- rectangular assumptions.*

$$(I - \gamma P_\pi)^{-1} \sqrt{\mathrm{Var}_{P_\pi}(V^{\pi,P})} \leq \sqrt{\frac{8}{\gamma^2(1-\gamma)^2} \mathrm{sp}(V^{\pi,P})_\infty} 1.$$

*See D.3.7 for the proof*

## D.2 Proof of Theorem 1 and Theorem 3

The first decomposition of the proof of Theorem 1 and Theorem 3 Agarwal et al. [2020] while the argument needs essential adjustments in order to adapt to the robustness setting. One has by assumptions using any planner in empirical RMDPs :

$$\left\| \widehat{V}^{\star,\sigma} - \widehat{V}^{\widehat{\pi},\sigma} \right\|_\infty \leq \varepsilon_{\mathsf{opt}}, \tag{55}$$

using previous inequality, performance gap $\left\| V^{\star,\sigma} - V^{\widehat{\pi},\sigma} \right\|_\infty$, can be upper bounded using 3 steps.

**First step: subdivide the performance gap in 3 terms.** We recall the definition of the optimal robust policy $\pi^\star$ with regard to $\mathcal{M}_{\mathsf{rob}}$ and the optimal robust policy $\widehat{\pi}^\star$, the optimal robust value function $\widehat{V}^{\star,\sigma}$ (resp. robust value function $\widehat{Q}^{\pi,\sigma}$) w.r.t. $\widehat{\mathcal{M}}_{\mathsf{rob}}$. Then, the performance gap $V^{\star,\sigma} - V^{\widehat{\pi},\sigma}$ can be decomposed in one optimization term and two statistical error terms

$$
\begin{aligned}
V^{\star,\sigma} - V^{\widehat{\pi},\sigma} &= \left( V^{\pi^\star,\sigma} - \widehat{V}^{\pi^\star,\sigma} \right) + \left( \widehat{V}^{\pi^\star,\sigma} - \widehat{V}^{\widehat{\pi}^\star,\sigma} \right) + \left( \widehat{V}^{\widehat{\pi}^\star,\sigma} - \widehat{V}^{\widehat{\pi},\sigma} \right) + \left( \widehat{V}^{\widehat{\pi},\sigma} - V^{\widehat{\pi},\sigma} \right) \\
&\overset{(i)}{\leq} \left( V^{\pi^\star,\sigma} - \widehat{V}^{\pi^\star,\sigma} \right) + \left( \widehat{V}^{\widehat{\pi}^\star,\sigma} - \widehat{V}^{\widehat{\pi},\sigma} \right) + \left( \widehat{V}^{\widehat{\pi},\sigma} - V^{\widehat{\pi},\sigma} \right) \\
&\overset{(ii)}{\leq} \left( V^{\pi^\star,\sigma} - \widehat{V}^{\pi^\star,\sigma} \right) + \varepsilon_{\mathsf{opt}} + \left( \widehat{V}^{\widehat{\pi},\sigma} - V^{\widehat{\pi},\sigma} \right)
\end{aligned} \tag{56}
$$

where (i) holds by $\widehat{V}^{\pi^\star,\sigma} - \widehat{V}^{\widehat{\pi}^\star,\sigma} \le 0$ since $\widehat{\pi}^\star$ is the robust optimal policy for $\widehat{\mathcal{M}}_{\text{rob}}$, and (ii) comes from (55) and definition of optimization error. The proof aims to control the last remaining terms in (56) using concentration theory and sufficiently big number of step $N$. To do so, we will consider a more general term $\widehat{V}^{\pi,\sigma} - V^{\pi,\sigma}$ for any policy $\pi$ even if control of these two terms slightly differ at the end. Using (32), it holds that for both $sa$- and $s$-rectangular assumptions:

$$
\begin{aligned}
\widehat{V}^{\pi,\sigma} - V^{\pi,\sigma} &= r_\pi + \gamma \underline{\widehat{P}}^{\pi,\widehat{V}} \widehat{V}^{\pi,\sigma} - \left( r_\pi + \gamma \underline{P}^{\pi,V} V^{\pi,\sigma} \right) \\
&= \left( \gamma \underline{\widehat{P}}^{\pi,\widehat{V}} \widehat{V}^{\pi,\sigma} - \gamma \underline{P}^{\pi,\widehat{V}} \widehat{V}^{\pi,\sigma} \right) + \left( \gamma \underline{P}^{\pi,\widehat{V}} \widehat{V}^{\pi,\sigma} - \gamma \underline{P}^{\pi,V} V^{\pi,\sigma} \right) \\
&\overset{(i)}{\le} \gamma \left( \underline{P}^{\pi,V} \widehat{V}^{\pi,\sigma} - \underline{P}^{\pi,V} V^{\pi,\sigma} \right) + \left( \gamma \underline{\widehat{P}}^{\pi,\widehat{V}} \widehat{V}^{\pi,\sigma} - \gamma \underline{P}^{\pi,\widehat{V}} \widehat{V}^{\pi,\sigma} \right),
\end{aligned}
$$

where (i) holds because $\underline{P}^{\pi,\widehat{V}} \widehat{V}^{\pi,\sigma} \le \underline{P}^{\pi,V} \widehat{V}^{\pi,\sigma}$ because of the optimality of $\underline{P}^{\pi,\widehat{V}}$ (see. (25)). Factorizing terms leads to the following equation

$$
\widehat{V}^{\pi,\sigma} - V^{\pi,\sigma} \le \gamma \left( I - \gamma \underline{P}^{\pi,V} \right)^{-1} \left( \underline{\widehat{P}}^{\pi,\widehat{V}} \widehat{V}^{\pi,\sigma} - \underline{P}^{\pi,\widehat{V}} \widehat{V}^{\pi,\sigma} \right). \tag{57}
$$

In the same manner, we can also obtain a lower bound of this quantity:

$$
\begin{aligned}
\widehat{V}^{\pi,\sigma} - V^{\pi,\sigma} &= r_\pi + \gamma \underline{\widehat{P}}^{\pi,\widehat{V}} \widehat{V}^{\pi,\sigma} - \left( r_\pi + \gamma \underline{P}^{\pi,V} V^{\pi,\sigma} \right) \\
&= \left( \gamma \underline{\widehat{P}}^{\pi,\widehat{V}} \widehat{V}^{\pi,\sigma} - \gamma \underline{P}^{\pi,\widehat{V}} \widehat{V}^{\pi,\sigma} \right) + \left( \gamma \underline{P}^{\pi,\widehat{V}} \widehat{V}^{\pi,\sigma} - \gamma \underline{P}^{\pi,V} V^{\pi,\sigma} \right) \\
&\ge \gamma \left( \underline{P}^{\pi,\widehat{V}} \widehat{V}^{\pi,\sigma} - \underline{P}^{\pi,\widehat{V}} V^{\pi,\sigma} \right) + \left( \gamma \underline{\widehat{P}}^{\pi,\widehat{V}} \widehat{V}^{\pi,\sigma} - \gamma \underline{P}^{\pi,\widehat{V}} \widehat{V}^{\pi,\sigma} \right) \\
&\ge \gamma \left( I - \gamma \underline{P}^{\pi,\widehat{V}} \right)^{-1} \left( \underline{\widehat{P}}^{\pi,\widehat{V}} \widehat{V}^{\pi,\sigma} - \underline{P}^{\pi,\widehat{V}} \widehat{V}^{\pi,\sigma} \right). \tag{58}
\end{aligned}
$$

Using both (57) and (58), we obtain infinite norm control:

$$
\begin{aligned}
\left\| \widehat{V}^{\pi,\sigma} - V^{\pi,\sigma} \right\|_\infty \le \gamma \max \Big\{ & \left\| \left( I - \gamma \underline{P}^{\pi,V} \right)^{-1} \left( \underline{\widehat{P}}^{\pi,\widehat{V}} \widehat{V}^{\pi,\sigma} - \underline{P}^{\pi,\widehat{V}} \widehat{V}^{\pi,\sigma} \right) \right\|_\infty, \\
& \left\| \left( I - \gamma \underline{P}^{\pi,\widehat{V}} \right)^{-1} \left( \underline{\widehat{P}}^{\pi,\widehat{V}} \widehat{V}^{\pi,\sigma} - \underline{P}^{\pi,\widehat{V}} \widehat{V}^{\pi,\sigma} \right) \right\|_\infty \Big\}. \tag{59}
\end{aligned}
$$

By decomposing the error in a symmetric way, he have

$$
\begin{aligned}
\left\| \widehat{V}^{\pi,\sigma} - V^{\pi,\sigma} \right\|_\infty \le \gamma \max \Big\{ & \left\| \left( I - \gamma \underline{\widehat{P}}^{\pi,V} \right)^{-1} \left( \underline{\widehat{P}}^{\pi,V} V^{\pi,\sigma} - \underline{P}^{\pi,V} V^{\pi,\sigma} \right) \right\|_\infty, \\
& \left\| \left( I - \gamma \underline{\widehat{P}}^{\pi,\widehat{V}} \right)^{-1} \left( \underline{\widehat{P}}^{\pi,V} V^{\pi,\sigma} - \underline{P}^{\pi,V} V^{\pi,\sigma} \right) \right\|_\infty \Big\}. \tag{60}
\end{aligned}
$$

Armed with these inequalities, we can use concentration inequalities to upper bound the two remaining terms $\left\| \widehat{V}^{\pi^\star,\sigma} - V^{\pi^\star,\sigma} \right\|_\infty$ and $\left\| \widehat{V}^{\widehat{\pi},\sigma} - V^{\widehat{\pi},\sigma} \right\|_\infty$ in (56). Taking $\pi = \widehat{\pi}$, applying (59) leads to

$$
\begin{aligned}
\left\| \widehat{V}^{\widehat{\pi},\sigma} - V^{\widehat{\pi},\sigma} \right\|_\infty \le \gamma \max \Big\{ & \left\| \left( I - \gamma \underline{P}^{\widehat{\pi},\widehat{V}} \right)^{-1} \left( \underline{\widehat{P}}^{\widehat{\pi},\widehat{V}} \widehat{V}^{\widehat{\pi},\sigma} - \underline{P}^{\widehat{\pi},\widehat{V}} \widehat{V}^{\widehat{\pi},\sigma} \right) \right\|_\infty, \\
& \left\| \left( I - \gamma \underline{P}^{\widehat{\pi},V} \right)^{-1} \left( \underline{\widehat{P}}^{\widehat{\pi},\widehat{V}} \widehat{V}^{\widehat{\pi},\sigma} - \underline{P}^{\widehat{\pi},\widehat{V}} \widehat{V}^{\widehat{\pi},\sigma} \right) \right\|_\infty \Big\}. \tag{61}
\end{aligned}
$$

Finally, $\pi = \pi^\star$, applying (60) gives us

$$
\begin{aligned}
\left\| \widehat{V}^{\pi^\star,\sigma} - V^{\pi^\star,\sigma} \right\|_\infty \le \gamma \max \Big\{ & \left\| \left( I - \gamma \underline{\widehat{P}}^{\pi^\star,V} \right)^{-1} \left( \underline{\widehat{P}}^{\pi^\star,V} V^{\pi^\star,\sigma} - \underline{P}^{\pi^\star,V} V^{\pi^\star,\sigma} \right) \right\|_\infty, \\
& \left\| \left( I - \gamma \underline{\widehat{P}}^{\pi^\star,\widehat{V}} \right)^{-1} \left( \underline{\widehat{P}}^{\pi^\star,V} V^{\pi^\star,\sigma} - \underline{P}^{\pi^\star,V} V^{\pi^\star,\sigma} \right) \right\|_\infty \Big\}. \tag{62}
\end{aligned}
$$

Note that to control $\left\| \widehat{V}^{\pi^\star,\sigma} - V^{\pi^\star,\sigma} \right\|_\infty$, we use decomposition not depending on $\widehat{\pi}$ for value function as $V^{\pi^\star,\sigma}$ is deterministic and fixed, allowing use of classical concentration analysis tools. This decomposition is the same for both $sa$-rectangular and $s$-rectangular case.

**Second step: bound first term and second term in** (62) **to control** $\|\widehat{V}^{\pi^\star,\sigma} - V^{\pi^\star,\sigma}\|_\infty$   To control the two terms in (62), we use lemma 8 based Bernstein's concentration argument and whose proof is in Appendix D.3.3.

**Lemma 8.** *For both $sa-$ and $s$-rectangular setting, consider any $\delta \in (0,1)$, with probability $1 - \delta$, it holds:*

$$\left| \widehat{\underline{P}}^{\pi^\star,V} V^{\pi^\star,\sigma} - \underline{P}^{\pi^\star,V} V^{\pi^\star,\sigma} \right| \leq 2\sqrt{\frac{L}{N}} \sqrt{\mathrm{Var}_{P^{\pi^\star}}(V^{\star,\sigma})} + \frac{3LC_S \|1\|_*}{N(1-\gamma)} 1 \qquad (63)$$

*with $L = 2\log(18 \|1\|_* SAN/\delta)$ and where $\mathrm{Var}_{P^{\pi^\star}}(V^{\star,\sigma})$ is defined in (27). Moreover, for the specific case of TV, this lemma is true without the smoothness term $\frac{3LC_S\|1\|_*}{N(1-\gamma)}$.*

Armed with the above lemma, now we control the **first term** on the right-hand side of (62) as follows:

$$\left(I - \gamma \widehat{\underline{P}}^{\pi^\star,V}\right)^{-1} \left(\widehat{\underline{P}}^{\pi^\star,V} V^{\pi^\star,\sigma} - \underline{P}^{\pi^\star,V} V^{\pi^\star,\sigma}\right)$$

$$\overset{(a)}{\leq} \left(I - \gamma \widehat{\underline{P}}^{\pi^\star,V}\right)^{-1} \left\| \widehat{\underline{P}}^{\pi^\star,V} V^{\pi^\star,\sigma} - \underline{P}^{\pi^\star,V} V^{\pi^\star,\sigma} \right\|_\infty$$

$$\overset{(b)}{\leq} \left(I - \gamma \widehat{\underline{P}}^{\pi^\star,V}\right)^{-1} \left( 2\sqrt{\frac{L}{N}} \sqrt{\mathrm{Var}_{P^{\pi^\star}}(V^{\star,\sigma})} + \frac{3LC_S \|1\|_*}{N(1-\gamma)} \right)$$

$$\leq \left(I - \gamma \widehat{\underline{P}}^{\pi^\star,V}\right)^{-1} \frac{3LC_S \|1\|_*}{N(1-\gamma)} 1 + \underbrace{2\sqrt{\frac{L}{N}} \left(I - \gamma \widehat{\underline{P}}^{\pi^\star,V}\right)^{-1} \sqrt{\mathrm{Var}_{\widehat{\underline{P}}^{\pi^\star,V}}(V^{\star,\sigma})}}_{=:\mathcal{R}_1}$$

$$+ \underbrace{2\sqrt{\frac{L}{N}} \left(I - \gamma \widehat{\underline{P}}^{\pi^\star,V}\right)^{-1} \sqrt{\left| \mathrm{Var}_{\widehat{P}^{\pi^\star}}(V^{\star,\sigma}) - \mathrm{Var}_{\widehat{\underline{P}}^{\pi^\star,V}}(V^{\star,\sigma}) \right|}}_{=:\mathcal{R}_2}$$

$$+ \underbrace{2\sqrt{\frac{L}{N}} \left(I - \gamma \widehat{\underline{P}}^{\pi^\star,V}\right)^{-1} \left( \sqrt{\mathrm{Var}_{P^{\pi^\star}}(V^{\star,\sigma})} - \sqrt{\mathrm{Var}_{\widehat{P}^{\pi^\star}}(V^{\star,\sigma})} \right)}_{=:\mathcal{R}_3}, \qquad (64)$$

where (a) holds as the matrix $\left(I - \gamma \widehat{\underline{P}}^{\pi^\star,V}\right)^{-1}$ is positive definite, (b) holds due to Lemma 8, and the last point holds from the following decomposition for variance and triangular inequality

$$\sqrt{\mathrm{Var}_{P^{\pi^\star}}(V^{\star,\sigma})} = \left( \sqrt{\mathrm{Var}_{P^{\pi^\star}}(V^{\star,\sigma})} - \sqrt{\mathrm{Var}_{\widehat{P}^{\pi^\star}}(V^{\star,\sigma})} \right) + \sqrt{\mathrm{Var}_{\widehat{P}^{\pi^\star}}(V^{\star,\sigma})}$$

$$\leq \left( \sqrt{\mathrm{Var}_{P^{\pi^\star}}(V^{\star,\sigma})} - \sqrt{\mathrm{Var}_{\widehat{P}^{\pi^\star}}(V^{\star,\sigma})} \right)$$

$$+ \sqrt{\left| \mathrm{Var}_{\widehat{P}^{\pi^\star}}(V^{\star,\sigma}) - \mathrm{Var}_{\widehat{\underline{P}}^{\pi^\star,V}}(V^{\star,\sigma}) \right|} + \sqrt{\mathrm{Var}_{\widehat{\underline{P}}^{\pi^\star,V}}(V^{\star,\sigma})}.$$

Finally, the fact that $\widehat{\underline{P}}^{\pi^\star,V}$ is a stochastic matrix, so

$$\left(I - \gamma \widehat{\underline{P}}^{\pi^\star,V}\right)^{-1} 1 = \left(I + \sum_{t=1}^{\infty} \gamma^t \left(\widehat{\underline{P}}^{\pi^\star,V}\right)^t\right) 1 \leq \frac{1}{1-\gamma} 1. \qquad (65)$$

Armed with these inequalities, the three terms $\mathcal{R}_1, \mathcal{R}_2, \mathcal{R}_3$ in (64) can be controlled separately.

- Consider $\mathcal{R}_1$. We first introduce the following lemma, whose proof is postponed to Appendix D.3.4.

**Lemma 9.** *Consider any $\delta \in (0,1)$. With probability at least $1 - \delta$, one has*

$$\left(I - \gamma \widehat{\underline{P}}^{\pi^\star, V}\right)^{-1} \sqrt{\mathrm{Var}_{\widehat{\underline{P}}^{\pi^\star, V}}(V^{\star,\sigma})} \leq 4 \sqrt{\frac{\left(1 + \left(\sqrt{\frac{L}{(1-\gamma)^2 N}} + \frac{C_S \|1\|_* L}{N(1-\gamma)}\right)\right)}{\gamma^3 (1-\gamma)^2 \max\{1-\gamma, C_g \sigma\}}} 1$$

$$\leq 4 \sqrt{\frac{\left(1 + \left(\sqrt{\frac{L}{(1-\gamma)^2 N}} + \frac{C_S \|1\|_* L}{N(1-\gamma)}\right)\right)}{\gamma^3 (1-\gamma)^3}} 1$$

*with $L = 2 \log(\frac{18 \|1\|_* SAN}{\delta})$ in the sa-rectangular case. In the s-rectangular case, it holds:*

$$\left(I - \gamma \widehat{\underline{P}}^{\pi^\star, V}\right)^{-1} \sqrt{\mathrm{Var}_{\widehat{\underline{P}}^{\pi^\star, V}}(V^{\star,\sigma})} \leq 4 \sqrt{\frac{\left(1 + \left(\sqrt{\frac{L}{(1-\gamma)^2 N}} + \frac{C_S \|1\|_* L}{N(1-\gamma)}\right)\right)}{\gamma^3 (1-\gamma)^2 \max\{1-\gamma, C_g \tilde{\sigma} \min_s \|\pi_s\|_*\}}} 1$$

$$\leq 4 \sqrt{\frac{\left(1 + \left(\sqrt{\frac{L}{(1-\gamma)^2 N}} + \frac{C_S \|1\|_* L}{N(1-\gamma)}\right)\right)}{\gamma^3 (1-\gamma)^3}} 1$$

Using Lemma 9 and inserting back to (64) gives in $sa$-rectangular case

$$\mathcal{R}_1 = 2 \sqrt{\frac{L}{N}} \left(I - \gamma \widehat{\underline{P}}^{\pi^\star, V}\right)^{-1} \sqrt{\mathrm{Var}_{\widehat{\underline{P}}^{\pi^\star, V}}(V^{\star,\sigma})}$$

$$\leq 8 \sqrt{\frac{L}{\gamma^3 (1-\gamma)^2 \max\{1-\gamma, C_g \sigma\} N}} \left(1 + \sqrt{\frac{L}{(1-\gamma)^2 N}} + \frac{C_S \|1\|_* L}{N(1-\gamma)}\right) 1. \quad (66)$$

- Consider $\mathcal{R}_2$. First, denote $V' := V^{\star,\sigma} - \eta 1 \; \eta \in \mathbb{R}$, by Lemma 5, we have for any $\pi$,

$$0 \leq \min_\eta \|V - \eta 1\|_\infty \leq \frac{1}{\gamma \max\{1-\gamma, C_g \sigma\}} \quad (67)$$

for $sa$-rectangular case or in $s$-rectangular we obtain

$$0 \leq \min_\eta \|V - \eta 1\|_\infty \leq \frac{1}{\gamma \max\{1-\gamma, \tilde{\sigma} C_g \|\pi_s\|_*\}} \quad (68)$$

by the definition of the span semi norm. Moreover, we can use Holder with $L_1$ and $L_\infty$ we have for both $sa$ and $s$-rectangular case to as it holds that:

$$\left|\mathrm{Var}_{\widetilde{P}_{s,a}}(V^{\star,\sigma}) - \mathrm{Var}_{P_{s,a}}(V^{\star,\sigma})\right| = \left|\mathrm{Var}_{\widetilde{P}_{s,a}}(V') - \mathrm{Var}_{P_{s,a}}(V')\right|$$

$$\leq \left\|\widetilde{P}_{s,a} - P_{s,a}\right\|_1 \|V'\|_\infty^2 \overset{a}{\leq} \frac{\sigma_1}{(\gamma^2 (\max(1-\gamma), C_g \sigma)^2}$$

$$\leq \frac{1}{\gamma^2 \max\{(1-\gamma), \sigma C_g\}}. \quad (69)$$

In the first inequality, we use $\|V'\|_\infty^2 = \|V'^2\|_\infty$ and and we use Lemma 5 in (a) where $C_g \sigma = \sigma_1$.

With the same arguments for $s$-rectangular, we obtain for $V' := V^{\star,\sigma} - \eta 1, \eta \in \mathbb{R}$,

$$\left|\Pi^{\pi^*}\left(\mathrm{Var}_{\widetilde{P}_s}(V^{\star,\sigma}) - \mathrm{Var}_{P_s}(V^{\star,\sigma})\right)\right| = \left|\Pi^{\pi^*}\left(\mathrm{Var}_{\widetilde{P}_s}(V') - \mathrm{Var}_{P_s}(V')\right)\right|$$

$$\leq \left|\sum_a \pi^*(a|s) \sum_{s'} (\widetilde{P}_s(s', a) - P_s(s', a)) V'(s')^2\right| \quad (70)$$

$$\leq \|V'\|_\infty^2 \sum_a \sum_{s'} \pi^*(a|s)(\widetilde{P}_s(s', a) - P_s(s', a)) \overset{a}{\leq} \|V'\|_\infty^2 \tilde{\sigma} \|\pi_s^*\|_* C_g^s 1 \quad (71)$$

$$\overset{b}{\leq} \frac{\tilde{\sigma} C_g^s \|\pi_s^*\|_* \|V'\|_\infty}{\gamma \|\pi_s^*\|_* \tilde{\sigma} C_g^s} 1 \leq \frac{\|V'\|}{\gamma} 1. \quad (72)$$

where (a) comes Eq 126, (b) comes lemma 6 or more precisely eq (139). Then, taking the sup over $s$ in the previous equations, it holds

$$\left|\Pi^{\pi^\star}\left(\operatorname{Var}_{\widetilde{P}_s}(V^{\star,\sigma}) - \operatorname{Var}_{P_s}(V^{\star,\sigma})\right)\right| \le \frac{\inf_{\eta\in\mathbb{R}^+}\|V - \eta 1'\|}{\gamma}1 \tag{73}$$

$$\le \frac{1}{\gamma^2\tilde{\sigma}\min_s\|\pi_s^*\|_* C_g}1. \tag{74}$$

Applying the previous inequality, it holds in $sa$-rectangular case:

$$\mathcal{R}_2 = 2\sqrt{\frac{L}{N}}\left(I - \gamma\widehat{\underline{P}}^{\pi^\star,V}\right)^{-1}\sqrt{\left|\operatorname{Var}_{\widehat{P}^{\pi^\star}}(V^{\star,\sigma}) - \operatorname{Var}_{\widehat{\underline{P}}^{\pi^\star,V}}(V^{\star,\sigma})\right|}$$

$$= 2\sqrt{\frac{L}{N}}\left(I - \gamma\widehat{\underline{P}}^{\pi^\star,V}\right)^{-1}\sqrt{\left|\Pi^{\pi^\star}\left(\operatorname{Var}_{\widehat{P}^0}(V^{\star,\sigma}) - \operatorname{Var}_{\widehat{P}^{\pi^\star},V}(V^{\star,\sigma})\right)\right|}$$

$$\le 2\sqrt{\frac{L}{N}}\left(I - \gamma\widehat{\underline{P}}^{\pi^\star,V}\right)^{-1}\sqrt{\left\|\operatorname{Var}_{\widehat{P}^0}(V^{\star,\sigma}) - \operatorname{Var}_{\widehat{P}^{\pi^\star},V}(V^{\star,\sigma})\right\|_\infty}1$$

$$\le 2\sqrt{\frac{L}{N}}\left(I - \gamma\widehat{\underline{P}}^{\pi^\star,V}\right)^{-1}\sqrt{\frac{1}{\gamma^2\max\{1-\gamma, C_g\sigma\}}}1 \tag{75}$$

$$\le 4\sqrt{\frac{L}{\gamma^2(1-\gamma)^2\max\{1-\gamma, C_g\sigma\}N}}1, \tag{76}$$

where the last inequality uses $\left(I - \gamma\widehat{\underline{P}}^{\pi^\star,V}\right)^{-1}1 \le \frac{1}{1-\gamma}1$ (cf. (65)) for $sa$-rectangular. In the $s$-rectangular case, we obtain a different result as

$$\mathcal{R}_2 = 2\sqrt{\frac{L}{N}}\left(I - \gamma\widehat{\underline{P}}^{\pi^\star,V}\right)^{-1}\sqrt{\left|\operatorname{Var}_{\widehat{P}^{\pi^\star}}(V^{\star,\sigma}) - \operatorname{Var}_{\widehat{\underline{P}}^{\pi^\star,V}}(V^{\star,\sigma})\right|}$$

$$= 2\sqrt{\frac{L}{N}}\left(I - \gamma\widehat{\underline{P}}^{\pi^\star,V}\right)^{-1}\sqrt{\left|\Pi^{\pi^\star}\left(\operatorname{Var}_{\widehat{P}^0}(V^{\star,\sigma}) - \operatorname{Var}_{\widehat{P}^{\pi^\star},V}(V^{\star,\sigma})\right)\right|}$$

$$\le 2\sqrt{\frac{L}{N}}\left(I - \gamma\widehat{\underline{P}}^{\pi^\star,V}\right)^{-1}\sqrt{\frac{1}{\gamma^2\max\{1-\gamma, \min_s\|\pi_s^*\|_\infty C_g\tilde{\sigma}\}}}1 \tag{77}$$

$$\le 2\sqrt{\frac{L}{\gamma^2(1-\gamma)^2\max\{1-\gamma, \min_s\|\pi_s^*\|_\infty\tilde{\sigma}C_g\}N}}1, \tag{78}$$

- Consider $\mathcal{R}_3$. The following lemma plays an important role.
  Applying Lemma 2 and using $\pi = \pi^\star$ and $V = V^{\star,\sigma}$, it holds

$$\sqrt{\operatorname{Var}_{P^{\pi^\star}}(V^{\star,\sigma})} - \sqrt{\operatorname{Var}_{\widehat{P}^{\pi^\star}}(V^{\star,\sigma})} \le \sqrt{\frac{2\|V^{\star,\sigma}\|_\infty^2\log(\frac{2SA}{\delta})}{N}}1,$$

which can be inserted in (64) to gives

$$\mathcal{R}_3 = 2\sqrt{\frac{L}{N}}\left(I - \gamma\widehat{\underline{P}}^{\pi^\star,V}\right)^{-1}\left(\sqrt{\operatorname{Var}_{P^{\pi^\star}}(V^{\star,\sigma})} - \sqrt{\operatorname{Var}_{\widehat{P}^{\pi^\star}}(V^{\star,\sigma})}\right)$$

$$\le \frac{4}{(1-\gamma)}\frac{\log(\frac{SAN}{\delta})\|[V^{\star,\sigma}\|_\infty}{N}1 \le \frac{4L}{(1-\gamma)^2N}1, \tag{79}$$

where the last line uses $\left(I - \gamma\widehat{\underline{P}}^{\pi^\star,V}\right)^{-1}1 \le \frac{1}{1-\gamma}1$ (cf. (65)).

Finally, inserting the results of $\mathcal{R}_1$ in (66), $\mathcal{R}_2$ in (76), $\mathcal{R}_3$ in (79), and (65) back into (64) gives

$$\left(I - \gamma \underline{\widehat{P}}^{\pi^\star, V}\right)^{-1} \left(\underline{\widehat{P}}^{\pi^\star, V} V^{\pi^\star, \sigma} - \underline{P}^{\pi^\star, V} V^{\pi^\star, \sigma}\right) \tag{80}$$

$$\leq 8 \sqrt{\frac{L}{\gamma^3 (1-\gamma)^2 \max\{1-\gamma, C_g \sigma\} N}} \left(1 + \sqrt{\frac{L}{(1-\gamma)^2 N}} + \frac{C_S \|1\|_* L}{N(1-\gamma)}\right) 1 + \frac{3 L C_S \|1\|_*}{N(1-\gamma)^2} 1$$

$$+ 2 \sqrt{\frac{2L}{\gamma^2 (1-\gamma)^2 \max\{1-\gamma, C_g \sigma\} N}} 1 + \frac{4L}{(1-\gamma)^2 N} 1$$

$$\leq 10 \sqrt{\frac{2L}{\gamma^3 (1-\gamma)^2 \max\{1-\gamma, C_g \sigma\} N}} \left(1 + \sqrt{\frac{L}{(1-\gamma)^2 N}} + \frac{C_S \|1\|_* L}{N(1-\gamma)}\right) 1 + \frac{4L}{(1-\gamma)^2 N} 1$$

$$\tag{81}$$

$$+ \frac{3 L C_S \|1\|_*}{N(1-\gamma)^2} 1$$

$$\leq 160 \sqrt{\frac{L(1 + \frac{C_S \|1\|_*}{N(1-\gamma)})}{(1-\gamma)^2 \max\{1-\gamma, C_g \sigma\} N}} 1 + \frac{7 L C_S \|1\|_*}{N(1-\gamma)^2} 1, \tag{82}$$

where the last inequality holds by the fact $\gamma \geq \frac{1}{4}$ and letting $N \geq \frac{L}{(1-\gamma)^2}$. We have the same result for $s$-rectangular, replacing, $\max\{1-\gamma, C_g \sigma\}$ by $\max\{1-\gamma, \min_s \|\pi_s^*\|_* \tilde{\sigma} C_g\}$.

Now we are ready to control **second term in** (62) to control $\|\widehat{V}^{\pi^\star, \sigma} - V^{\pi^\star, \sigma}\|_\infty$. To proceed, applying Lemma 8 on the second term of the right-hand side of (62) leads to

$$\left(I - \gamma \underline{\widehat{P}}^{\pi^\star, \widehat{V}}\right)^{-1} \left(\underline{\widehat{P}}^{\pi^\star, V} V^{\pi^\star, \sigma} - \underline{P}^{\pi^\star, V} V^{\pi^\star, \sigma}\right)$$

$$\leq \left(I - \gamma \underline{\widehat{P}}^{\pi^\star, \widehat{V}}\right)^{-1} \left(2 \sqrt{\frac{L}{N}} \sqrt{\mathrm{Var}_{P^{\pi^\star}}(V^{\star, \sigma})} + \frac{3 L C_S \|1\|_*}{N(1-\gamma)}\right)$$

$$\leq \left(I - \gamma \underline{\widehat{P}}^{\pi^\star, \widehat{V}}\right)^{-1} \frac{L' C_S \|1\|_*}{N(1-\gamma)} + \underbrace{2 \sqrt{\frac{L}{N}} \left(I - \gamma \underline{\widehat{P}}^{\pi^\star, \widehat{V}}\right)^{-1} \sqrt{\mathrm{Var}_{\underline{\widehat{P}}^{\pi^\star, \widehat{v}}}(\widehat{V}^{\pi^\star, \sigma})}}_{=: \mathcal{R}_4}$$

$$\underbrace{2 \sqrt{\frac{L}{N}} \left(I - \gamma \underline{\widehat{P}}^{\pi^\star, \widehat{V}}\right)^{-1} \left(\sqrt{\mathrm{Var}_{\underline{\widehat{P}}^{\pi^\star, \widehat{v}}}(V^{\pi^\star, \sigma} - \widehat{V}^{\pi^\star, \sigma})}\right)}_{=: \mathcal{R}_5}$$

$$+ \underbrace{2 \sqrt{\frac{L}{N}} \left(I - \gamma \underline{\widehat{P}}^{\pi^\star, \widehat{V}}\right)^{-1} \left(\sqrt{\left|\mathrm{Var}_{\widehat{P}^{\pi^\star}}(V^{\star, \sigma}) - \mathrm{Var}_{\underline{\widehat{P}}^{\pi^\star, \widehat{v}}}(V^{\star, \sigma})\right|}\right)}_{=: \mathcal{R}_6}$$

$$+ \underbrace{2 \sqrt{\frac{L}{N}} \left(I - \gamma \underline{\widehat{P}}^{\pi^\star, \widehat{V}}\right)^{-1} \left(\sqrt{\mathrm{Var}_{P^{\pi^\star}}(V^{\star, \sigma})} - \sqrt{\mathrm{Var}_{\widehat{P}^{\pi^\star}}(V^{\star, \sigma})}\right)}_{=: \mathcal{R}_7}. \tag{83}$$

We now bound the above four terms $\mathcal{R}_4, \mathcal{R}_5, \mathcal{R}_6, \mathcal{R}_7$ separately.

- Using Lemma 7 with $P = \widehat{P}^{\pi^\star, \widehat{V}}$, $\pi = \pi^\star$ and $V = \widehat{V}^{\pi^\star, \sigma}$ which follow $\widehat{V}^{\pi^\star, \sigma} = r_{\pi^\star} + \gamma \underline{\widehat{P}}^{\pi^\star, \widehat{V}} \widehat{V}^{\pi^\star, \sigma}$, and in view of (65), the term $\mathcal{R}_4$ in (83) can be controlled as follows:

$$\mathcal{R}_4 = 2\sqrt{\frac{L}{N}} \left(I - \gamma \underline{\widehat{P}}^{\pi^\star, \widehat{V}}\right)^{-1} \sqrt{\mathrm{Var}_{\underline{\widehat{P}}^{\pi^\star, \hat{v}}}(\widehat{V}^{\pi^\star, \sigma})}$$

$$\leq 2\sqrt{\frac{L}{N}} \sqrt{\frac{8\min\{\mathrm{sp}(\widehat{V}^{\pi^\star, \sigma})_*, 1/(1-\gamma))}{\gamma^2 (1-\gamma)^2}} 1$$

$$\leq 8\sqrt{\frac{L}{\gamma^3 (1-\gamma)^2 \max\{1-\gamma, C_g \sigma\} N}} 1, \tag{84}$$

where the last inequality is due to Lemma 5 for $sa$-rectangular case and with the same quantity replacing $\max\{1 - \gamma, \sigma\}$ by $\max\{1 - \gamma, \min_s \|\pi_s^*\|_* \tilde{\sigma}\}$ in the $s-$ rectangular case.

- For bounding $\mathcal{R}_5$, we can simply use (65)) to get

$$\mathcal{R}_5 = 2\sqrt{\frac{L}{N}} \left(I - \gamma \underline{\widehat{P}}^{\pi^\star, \widehat{V}}\right)^{-1} \sqrt{\mathrm{Var}_{\underline{\widehat{P}}^{\pi^\star, \hat{v}}}(V^{\pi^\star, \sigma} - \widehat{V}^{\pi^\star, \sigma})}$$

$$\leq 2\sqrt{\frac{L}{(1-\gamma)^2 N}} \left\|V^{\star, \sigma} - \widehat{V}^{\pi^\star, \sigma}\right\|_\infty 1. \tag{85}$$

$$\mathcal{R}_5 \leq 2\sqrt{\frac{L}{(1-\gamma)^2 N}} \left\|V^{\star, \sigma} - \widehat{V}^{\pi^\star, \sigma}\right\|_\infty 1. \tag{86}$$

- The term $\mathcal{R}_6$ can upper bounded as (76) as follows:

$$\mathcal{R}_6 \leq 2\sqrt{\frac{2L}{\gamma^2 (1-\gamma)^2 \max\{1-\gamma, C_g \sigma\} N}} 1. \tag{87}$$

for $sa$-rectangular case and with the same quantity replacing $\max\{1 - \gamma, C_g \sigma\}$ by $\max\{1 - \gamma, \min_s \|\pi_s^*\|_* \tilde{\sigma} C_g\}$ in the $s-$ rectangular case.

- Finally, $\mathcal{R}_7$ can be controlled the same as (79) shown below:

$$\mathcal{R}_7 \leq \frac{4L}{(1-\gamma)^2 N} 1. \tag{88}$$

Combining the results in (84), (86), (87), and (88) and inserting back to (83) leads to for $N \geq \frac{L}{(1-\gamma)^2}$

$$\left(I - \gamma \underline{\widehat{P}}^{\pi^\star, \widehat{V}}\right)^{-1} \left(\underline{\widehat{P}}^{\pi^\star, V} V^{\pi^\star, \sigma} - \underline{P}^{\pi^\star, V} V^{\pi^\star, \sigma}\right) \leq 8\sqrt{\frac{L(1 + \frac{C_S \|1\|_*}{N(1-\gamma)})}{\gamma^3 (1-\gamma)^2 \max\{1-\gamma, C_g \sigma\} N}} 1$$

$$+ 2\sqrt{\frac{L}{(1-\gamma)^2 N}} \left\|V^{\star, \sigma} - \widehat{V}^{\pi^\star, \sigma}\right\|_\infty 1 + 2\sqrt{\frac{2L}{\gamma^2 (1-\gamma)^2 \max\{1-\gamma, C_g \sigma\} N}} 1 + \frac{7LC_S \|1\|_*}{N(1-\gamma)^2}$$

$$\leq 80\sqrt{\frac{L(1 + \frac{C_S \|1\|_*}{N(1-\gamma)})}{(1-\gamma)^2 \max\{1-\gamma, C_g \sigma\} N}} 1 + 2\sqrt{\frac{L}{(1-\gamma)^2 N}} \left\|V^{\star, \sigma} - \widehat{V}^{\pi^\star, \sigma}\right\|_\infty 1 + \frac{7LC_S \|1\|_*}{N(1-\gamma)^2} 1, \tag{89}$$

where the last inequality follows from the assumption $\gamma \geq \frac{1}{4}$. Finally, inserting (82) and (89) back to (62) yields

$$
\left\| \widehat{V}^{\pi^\star, \sigma} - V^{\pi^\star, \sigma} \right\|_\infty \leq \max \left\{ 160 \sqrt{ \frac{L(1 + \frac{C_S \|1\|_*}{N(1-\gamma)})}{(1-\gamma)^2 \max\{1-\gamma, C_g \sigma\} N} } + \frac{7 L C_S \|1\|_*}{N(1-\gamma)^2}, \right.
$$

$$
\left. 80 \sqrt{ \frac{L(1 + \frac{C_S \|1\|_*}{N(1-\gamma)})}{(1-\gamma)^2 \max\{1-\gamma, C_g \sigma\} N} } + 2 \sqrt{ \frac{L}{(1-\gamma)^2 N} } \left\| V^{\star, \sigma} - \widehat{V}^{\pi^\star, \sigma} \right\|_\infty + \frac{7 L C_S \|1\|_*}{N(1-\gamma)^2} \right\}
$$

$$
\leq 160 \sqrt{ \frac{L(1 + \frac{C_S \|1\|_*}{N(1-\gamma)})}{(1-\gamma)^2 \max\{1-\gamma, C_g \sigma\} N} } + \frac{14 L C_S \|1\|_*}{N(1-\gamma)^2}, \tag{90}
$$

where the last inequality holds by taking $N \geq \frac{16 \log(\frac{SAN}{\delta})}{(1-\gamma)^2}$ rearranging terms. In $s$-rectangular case, we obtain the same result, replacing $\max\{1-\gamma, C_g \sigma\}$ by $\max\{1-\gamma, \min_s \|\pi_s^*\|_* C_g \tilde{\sigma}\}$.

**Third step: controlling $\|\widehat{V}^{\widehat{\pi}, \sigma} - V^{\widehat{\pi}, \sigma}\|_\infty$ or bounding the first and second term in** (61). Unlike the earlier term, one has to face a more complicated statistical dependency between $\widehat{\pi}$ and the empirical RMDP. To begin with, we introduce the following lemma which controls the main term on the right-hand side of (61), which is proved in Appendix D.3.5.

**Lemma 10.** *Consider any $\delta \in (0, 1)$. Taking $N \geq L''$ with probability at least $1 - \delta$, one has for sa- or s-rectangular case :*

$$
\left| \widehat{\underline{P}}^{\widehat{\pi}, \widehat{V}} \widehat{V}^{\widehat{\pi}, \sigma} - \underline{P}^{\widehat{\pi}, \widehat{V}} \widehat{V}^{\widehat{\pi}, \sigma} \right| \leq 2 \sqrt{ \frac{L'}{N} } \sqrt{ \mathrm{Var}_{P_{s,a}^0}(\widehat{V}^{\star, \sigma}) } 1 + 2 \varepsilon_{\mathsf{opt}} 1 + \frac{15 L'' C_S \|1\|_*}{N(1-\gamma)}
$$

$$
\leq 2 \sqrt{ \frac{L''}{(1-\gamma)^2 N} } 1 + 2 \varepsilon_{\mathsf{opt}} 1 + \frac{14 L'' C_S \|1\|_*}{N(1-\gamma)} 1. \tag{91}
$$

*with $L'' = 2 \log(\frac{54 \|1\|_* SAN^2}{(1-\gamma)\delta})$. Moreover, for TV this lemma holds but without the geometric term $\frac{14 L'' C_S \|1\|_*}{N(1-\gamma)} 1$. Taking the sup over $s$ gives the final result.*

With Lemma 10 in hand, we have to control **first term** in (61)

$$
\left( I - \gamma \underline{P}^{\widehat{\pi}, \widehat{V}} \right)^{-1} \left( \widehat{\underline{P}}^{\widehat{\pi}, \widehat{V}} \widehat{V}^{\widehat{\pi}, \sigma} - \underline{P}^{\widehat{\pi}, \widehat{V}} \widehat{V}^{\widehat{\pi}, \sigma} \right)
$$

$$
\overset{(i)}{\leq} \left( I - \gamma \underline{P}^{\widehat{\pi}, \widehat{V}} \right)^{-1} \left| \widehat{\underline{P}}^{\widehat{\pi}, \widehat{V}} \widehat{V}^{\widehat{\pi}, \sigma} - \underline{P}^{\widehat{\pi}, \widehat{V}} \widehat{V}^{\widehat{\pi}, \sigma} \right|
$$

$$
\leq 2 \sqrt{ \frac{L'}{N} } \left( I - \gamma \underline{P}^{\widehat{\pi}, \widehat{V}} \right)^{-1} \sqrt{ \mathrm{Var}_{P^{\widehat{\pi}}}(\widehat{V}^{\star, \sigma}) } + \left( I - \gamma \underline{P}^{\widehat{\pi}, V^{\widehat{\pi}}} \right)^{-1} \left( 2 \varepsilon_{\mathsf{opt}} \right) 1 \tag{92}
$$

$$
+ \left( I - \gamma \underline{P}^{\widehat{\pi}, V^{\widehat{\pi}}} \right)^{-1} \frac{14 L'' C_S \|1\|_*}{N(1-\gamma)} 1
$$

$$
\overset{(ii)}{\leq} \left( \frac{2 \varepsilon_{\mathsf{opt}}}{1-\gamma} \right) 1 + \underbrace{ 2 \sqrt{ \frac{L'}{N} } \left( I - \gamma \underline{P}^{\widehat{\pi}, \widehat{V}} \right)^{-1} \sqrt{ \mathrm{Var}_{\underline{P}^{\widehat{\pi}, \widehat{v}}}(\widehat{V}^{\widehat{\pi}, \sigma}) } }_{=: \mathcal{S}_1}
$$

$$
+ \underbrace{ 2 \sqrt{ \frac{L'}{N} } \left( I - \gamma \underline{P}^{\widehat{\pi}, \widehat{V}} \right)^{-1} \sqrt{ \left| \mathrm{Var}_{\underline{P}^{\widehat{\pi}, \widehat{v}}}(\widehat{V}^{\star, \sigma}) - \mathrm{Var}_{\underline{P}^{\widehat{\pi}, \widehat{v}}}(\widehat{V}^{\widehat{\pi}, \sigma}) \right| } }_{=: \mathcal{S}_2}
$$

$$
+ \underbrace{ 2 \sqrt{ \frac{L'}{N} } \left( I - \gamma \underline{P}^{\widehat{\pi}, \widehat{V}} \right)^{-1} \sqrt{ \left| \mathrm{Var}_{P^{\widehat{\pi}}}(\widehat{V}^{\star, \sigma}) - \mathrm{Var}_{\underline{P}^{\widehat{\pi}, \widehat{v}}}(\widehat{V}^{\star, \sigma}) \right| } }_{=: \mathcal{S}_3} + \frac{14 L'' C_S \|1\|_*}{N(1-\gamma)^2} 1, \tag{93}
$$

where (i) and (ii) hold by the fact that each row of $(1-\gamma)\left(I-\gamma\underline{P}^{\widehat{\pi},\widehat{V}}\right)^{-1}$ is a probability vector that falls into $\Delta(\mathcal{S})$. The remainder of the proof will focus on controlling the three terms in (93) separately.

- For $\mathcal{S}_1$, we introduce the following lemma, whose proof is postponed to D.3.6.

  **Lemma 11.** *Consider any* $\delta \in (0,1)$. *Taking* $N \geq \frac{L''}{(1-\gamma)^2}$ *one has with probability at least* $1-\delta$, *for sa$-$ rectangular*

$$\left(I-\gamma\underline{P}^{\widehat{\pi},\widehat{V}}\right)^{-1}\sqrt{\mathrm{Var}_{\underline{P}^{\widehat{\pi},\widehat{v}}}(\widehat{V}^{\widehat{\pi},\sigma})} \qquad \leq 6\sqrt{\frac{\left(1+\varepsilon_{\mathsf{opt}}+\frac{L''C_S\|1\|_*}{N(1-\gamma)}\right)}{\gamma^3(1-\gamma)^2\max\{1-\gamma,\sigma\}}}\mathbf{1}$$

$$\leq 6\sqrt{\frac{\left(1+\varepsilon_{\mathsf{opt}}+\frac{L''C_S\|1\|_*}{N(1-\gamma)}\right)}{(1-\gamma)^3\gamma^3}}\mathbf{1}.$$

  *and for s-rectangular*

$$\left(I-\gamma\underline{P}^{\widehat{\pi},\widehat{V}}\right)^{-1}\sqrt{\mathrm{Var}_{\underline{P}^{\widehat{\pi},\widehat{v}}}(\widehat{V}^{\widehat{\pi},\sigma})} \leq 6\sqrt{\frac{L''\left(1+\varepsilon_{\mathsf{opt}}+\frac{C_S\|1\|_*}{N(1-\gamma)}\right)}{\gamma^3(1-\gamma)^2\max\{1-\gamma,C_g\tilde{\sigma}\min_s\|\widehat{\pi}_s\|_\infty\}}}\mathbf{1}$$

$$\leq 6\sqrt{\frac{L''\left(1+\varepsilon_{\mathsf{opt}}+\frac{C_S\|1\|_*}{N(1-\gamma)}\right)}{(1-\gamma)^3\gamma^2}}\mathbf{1}.$$

  Applying Lemma 11 and (65) to (93) leads to

$$\mathcal{S}_1 = 2\sqrt{\frac{L'}{N}}\left(I-\gamma\underline{P}^{\widehat{\pi},\widehat{V}}\right)^{-1}\sqrt{\mathrm{Var}_{\underline{P}^{\widehat{\pi},\widehat{v}}}(\widehat{V}^{\widehat{\pi},\sigma})}$$

$$\leq 12\sqrt{\frac{L''}{\gamma^3(1-\gamma)^2\max\{1-\gamma,C_g\sigma\}N}}\mathbf{1}. \tag{94}$$

  for $sa$-rectangular and the same quantity replacing $\max\{1-\gamma,C_g\sigma\}$ by $\max\{1-\gamma,C_g\tilde{\sigma}\min_s\|\widehat{\pi}_s\|_*\}$ for $s-$ rectangular case.

- Applying Lemma 1 with $\|\widehat{V}^{\star,\sigma}-\widehat{V}^{\widehat{\pi},\sigma}\|_\infty \leq \varepsilon_{\mathsf{opt}}$ and (65), $\mathcal{S}_2$ can be controlled as

$$\mathcal{S}_2 = 2\sqrt{\frac{L''}{N}}\left(I-\gamma\underline{P}^{\widehat{\pi},\widehat{V}}\right)^{-1}\sqrt{\left|\mathrm{Var}_{\underline{P}^{\widehat{\pi},\widehat{v}}}(\widehat{V}^{\star,\sigma})-\mathrm{Var}_{\underline{P}^{\widehat{\pi},\widehat{v}}}(\widehat{V}^{\widehat{\pi},\sigma})\right|}$$

$$\leq 4\sqrt{\frac{L''}{N}}\left(I-\gamma\underline{P}^{\widehat{\pi},\widehat{V}}\right)^{-1}\sqrt{\varepsilon_{\mathsf{opt}}\frac{1}{1-\gamma}^2} \leq 8\sqrt{\frac{\varepsilon_{\mathsf{opt}}L''}{(1-\gamma)^4N}}\mathbf{1}. \tag{95}$$

- $\mathcal{S}_3$ can be controlled similar to $\mathcal{R}_2$ in (76) as follows:

$$\mathcal{S}_3 = 2\sqrt{\frac{L''}{N}}\left(I-\gamma\underline{P}^{\widehat{\pi},\widehat{V}}\right)^{-1}\sqrt{\left|\mathrm{Var}_{P^{\widehat{\pi}}}(\widehat{V}^{\star,\sigma})-\mathrm{Var}_{\underline{P}^{\widehat{\pi},\widehat{v}}}(\widehat{V}^{\star,\sigma})\right|}$$

$$\leq 4\sqrt{\frac{L''}{N}}\left(I-\gamma\underline{P}^{\widehat{\pi},\widehat{V}}\right)^{-1}\sqrt{\frac{1}{\gamma^2\max\{1-\gamma,C_g\sigma\}}}\mathbf{1} \tag{96}$$

$$\leq 8\sqrt{\frac{L''}{\gamma^2(1-\gamma)^2\max\{1-\gamma,C_g\sigma\}N}}\mathbf{1} \tag{97}$$

  for $sa$-rectangular and replacing $\max\{1-\gamma,\sigma\}$ by $\max\{1-\gamma,\tilde{\sigma}\min_s\|\widehat{\pi}_s\|_*\}$ for $s-$ rectangular case.

Finally, summing up the results in (94), (95), and (97) and inserting them back to (93) yields: taking $N \geq \frac{L''}{(1-\gamma)^2}$, with probability at least $1 - \delta$,

$$\left(I - \gamma \underline{P}^{\widehat{\pi},\widehat{V}}\right)^{-1} \left(\widehat{\underline{P}}^{\widehat{\pi},\widehat{V}} \widehat{V}^{\widehat{\pi},\sigma} - \underline{P}^{\widehat{\pi},\widehat{V}} \widehat{V}^{\widehat{\pi},\sigma}\right) \leq \left(\frac{2\varepsilon_{\mathsf{opt}}}{1 - \gamma}\right) 1 + \frac{14 L'' C_S \|1\|_*}{N(1-\gamma)^2} 1$$

$$+ 12 \sqrt{\frac{L''\left(1 + \varepsilon_{\mathsf{opt}} + \frac{C_S \|1\|_*}{N(1-\gamma)}\right)}{\gamma^3 (1-\gamma)^2 \max\{1 - \gamma, C_g \sigma\} N}} 1 + 8 \sqrt{\frac{\varepsilon_{\mathsf{opt}} L'}{(1-\gamma)^4 N}} 1 + \tag{98}$$

$$8 \sqrt{\frac{L'}{\gamma^2 (1-\gamma)^2 \max\{1 - \gamma, C_g \sigma\} N}} 1$$

$$\leq 16 \sqrt{\frac{L''\left(1 + \varepsilon_{\mathsf{opt}} + \frac{C_S \|1\|_*}{N(1-\gamma)}\right)}{\gamma^3 (1-\gamma)^2 \max\{1 - \gamma, \sigma\} N}} 1 + \left(\frac{2\varepsilon_{\mathsf{opt}} \gamma}{(1 - \gamma)} + 8 \sqrt{\frac{\varepsilon_{\mathsf{opt}} \gamma L'}{(1-\gamma)^4 N}} 1 + \frac{15 L'' C_S \|1\|_*}{N(1-\gamma)^2} 1\right)$$

$$\tag{99}$$

$$\tag{100}$$

for $sa$-rectangular and the same quantity replacing $\max\{1 - \gamma, \sigma\}$ by $\max\{1 - \gamma, \widetilde{\sigma} \min_s \|\widehat{\pi}_s\|_*\}$ for $s-$ rectangular case. In this step, it is harder to decouple terms as $\widehat{V}^{\widehat{\pi}}$ depends on data both in $\widehat{\pi}$ and $\widehat{V}$.

**Step 5: controlling $\|\widehat{V}^{\widehat{\pi},\sigma} - V^{\widehat{\pi},\sigma}\|_\infty$: bounding the second term in** (61).    Towards this, applying Lemma 10 leads to in $sa$-rectangular case:

$$\left(I - \gamma \underline{P}^{\widehat{\pi},V}\right)^{-1} \left(\widehat{\underline{P}}^{\widehat{\pi},\widehat{V}} \widehat{V}^{\widehat{\pi},\sigma} - \underline{P}^{\widehat{\pi},\widehat{V}} \widehat{V}^{\widehat{\pi},\sigma}\right) \leq \left(I - \gamma \underline{P}^{\widehat{\pi},V}\right)^{-1} \left|\widehat{\underline{P}}^{\widehat{\pi},\widehat{V}} \widehat{V}^{\widehat{\pi},\sigma} - \underline{P}^{\widehat{\pi},\widehat{V}} \widehat{V}^{\widehat{\pi},\sigma}\right|$$

$$\leq 2 \sqrt{\frac{L''}{N}} \left(I - \gamma \underline{P}^{\widehat{\pi},V}\right)^{-1} \sqrt{\mathrm{Var}_{P^{\widehat{\pi}}}(\widehat{V}^{\star,\sigma})} + \left(I - \gamma \underline{P}^{\widehat{\pi},V}\right)^{-1} \left(2\varepsilon_{\mathsf{opt}}\right) 1 \tag{101}$$

$$+ \left(I - \gamma \underline{P}^{\widehat{\pi},V}\right)^{-1} \frac{L'' 14 C_S \|1\|_*}{N(1-\gamma)} 1$$

$$\leq \left(\frac{2\varepsilon_{\mathsf{opt}}}{(1 - \gamma)}\right) 1 + \underbrace{2 \sqrt{\frac{L''}{N}} \left(I - \gamma \underline{P}^{\widehat{\pi},V}\right)^{-1} \sqrt{\mathrm{Var}_{\underline{P}^{\widehat{\pi},V}}(V^{\widehat{\pi},\sigma})}}_{=:S_4} + \left(I - \gamma \underline{P}^{\widehat{\pi},V}\right)^{-1} \frac{14 L'' C_S \|1\|_*}{N(1-\gamma)} 1$$

$$+ \underbrace{2 \sqrt{\frac{L'}{N}} \left(I - \gamma \underline{P}^{\widehat{\pi},V}\right)^{-1} \sqrt{\mathrm{Var}_{\underline{P}^{\widehat{\pi},V}}(\widehat{V}^{\widehat{\pi},\sigma} - V^{\widehat{\pi},\sigma})}}_{=:S_5}$$

$$+ \underbrace{2 \sqrt{\frac{L''}{N}} \left(I - \gamma \underline{P}^{\widehat{\pi},\widehat{V}}\right)^{-1} \sqrt{\left|\mathrm{Var}_{\underline{P}^{\widehat{\pi},V}}(\widehat{V}^{\star,\sigma}) - \mathrm{Var}_{\underline{P}^{\widehat{\pi},V}}([\widehat{V}^{\widehat{\pi},\sigma})\right|}}_{=:S_6}$$

$$+ \underbrace{2 \sqrt{\frac{L''}{N}} \left(I - \gamma \underline{P}^{\widehat{\pi},\widehat{V}}\right)^{-1} \sqrt{\left|\mathrm{Var}_{P^{\widehat{\pi}}}(\widehat{V}^{\star,\sigma}) - \mathrm{Var}_{\underline{P}^{\widehat{\pi},V}}([\widehat{V}^{\star,\sigma})\right|}}_{=:S_7}. \tag{102}$$

We shall bound each of the terms separately.

- Applying Lemma 7 with $P = \underline{P}^{\widehat{\pi},V}$, $\pi = \widehat{\pi}$, and taking $V = V^{\widehat{\pi},\sigma}$ which obeys $V^{\widehat{\pi},\sigma} = r_{\widehat{\pi}} + \gamma \underline{P}^{\widehat{\pi},V} V^{\widehat{\pi},\sigma}$, the term $S_4$ can be controlled similar to (84) as follows:

$$S_4 \leq 8 \sqrt{\frac{L''\left(1 + \varepsilon_{\mathsf{opt}} + \frac{C_S \|1\|_*}{N(1-\gamma)}\right)}{\gamma^3 (1-\gamma)^2 \max\{1 - \gamma, C_g \sigma\} N}} 1. \tag{103}$$

  for $sa$-rectangular and the same quantity replacing $\max\{1 - \gamma, C_g \sigma\}$ by $\max\{1 - \gamma, \min_s \|\widehat{\pi}_s\|_* \widetilde{\sigma} C_g\}$ for $s-$ rectangular case.

- For $\mathcal{S}_5$, it is observed that

$$\mathcal{S}_5 = 2\sqrt{\frac{L''}{N}}\left(I - \gamma\underline{P}^{\widehat{\pi},V}\right)^{-1}\sqrt{\mathrm{Var}_{\underline{P}^{\widehat{\pi},V}}\left(\widehat{V}^{\widehat{\pi},\sigma} - V^{\widehat{\pi},\sigma}\right)}$$

$$\leq 2\sqrt{\frac{L''}{(1-\gamma)^2 N}}\left\|V^{\widehat{\pi},\sigma} - \widehat{V}^{\widehat{\pi},\sigma}\right\|_\infty 1. \tag{104}$$

- Next, observing that $\mathcal{S}_6$ and $\mathcal{S}_7$ are almost the same as the terms $\mathcal{S}_2$ (controlled in (95)) and $\mathcal{S}_3$ (controlled in (97)) in (93), it is easily verified that they can be controlled as follows

$$\mathcal{S}_6 \leq 4\sqrt{\frac{\varepsilon_{\mathsf{opt}} L''}{(1-\gamma)^4 N}}1, \qquad \mathcal{S}_7 \leq 4\sqrt{\frac{L''}{\gamma^2(1-\gamma)^2\max\{1-\gamma, C_g\sigma\}N}}1. \tag{105}$$

for $sa$-rectangular and the same quantity replacing $\max\{1-\gamma, \sigma\}$ by $\max\{1-\gamma, \min_s\|\widehat{\pi}_s\|_*\widetilde{\sigma}\}$ for $s-$ rectangular case. Then inserting the results in (103), (104), and (105) back to (102) leads to

$$\left(I - \gamma\underline{P}^{\widehat{\pi},V}\right)^{-1}\left(\widehat{\underline{P}}^{\widehat{\pi},\widehat{V}}\widehat{V}^{\widehat{\pi},\sigma} - \underline{P}^{\widehat{\pi},\widehat{V}}\widehat{V}^{\widehat{\pi},\sigma}\right) \tag{106}$$

$$\leq \left(\frac{2\varepsilon_{\mathsf{opt}}}{(1-\gamma)}\right)1 + 8\sqrt{\frac{L''\left(1 + \varepsilon_{\mathsf{opt}} + \frac{C_S\|1\|_*}{N(1-\gamma)}\right)}{\gamma^3(1-\gamma)^2\max\{1-\gamma, \sigma\}N}}1 + \frac{14L''C_S\|1\|_*}{N(1-\gamma)^2}1$$

$$+ 2\sqrt{\frac{L''}{(1-\gamma)^2 N}}\left\|V^{\widehat{\pi},\sigma} - \widehat{V}^{\widehat{\pi},\sigma}\right\|_\infty 1 + 4\sqrt{\frac{L''\varepsilon_{\mathsf{opt}}}{(1-\gamma)^4 N}}1 + 4\sqrt{\frac{L''}{\gamma^2(1-\gamma)^2\max\{1-\gamma, C_g\sigma\}N}}1$$

$$\leq 12\sqrt{\frac{L''\left(1 + \varepsilon_{\mathsf{opt}} + \frac{C_S\|1\|_*}{N(1-\gamma)}\right)}{\gamma^3(1-\gamma)^2\max\{1-\gamma, \sigma\}N}} + 4\sqrt{\frac{L''}{(1-\gamma)^2 N}}\left\|V^{\widehat{\pi},\sigma} - \widehat{V}^{\widehat{\pi},\sigma}\right\|_\infty 1 \tag{107}$$

$$+ \frac{3\varepsilon_{\mathsf{opt}}}{(1-\gamma)}1 + \frac{14L''C_S\|1\|_*}{N(1-\gamma)^2}1. \tag{108}$$

$$\tag{109}$$

Taking $N \geq \frac{16L''}{1-\gamma}$, we obtain $\frac{2\varepsilon_{\mathsf{opt}}}{(1-\gamma)} + 4\varepsilon_{\mathsf{opt}}\sqrt{\frac{L''}{(1-\gamma)^4 N}}1 \leq \frac{3\varepsilon_{\mathsf{opt}}}{(1-\gamma)}$ with probability at least $1 - \delta$, inserting (99) and (107) back to (61)

$$\left\|\widehat{V}^{\widehat{\pi},\sigma} - V^{\widehat{\pi},\sigma}\right\|_\infty \leq \max\left\{16\sqrt{\frac{L''\left(1 + \varepsilon_{\mathsf{opt}} + \frac{C_S\|1\|_*}{N(1-\gamma)}\right)}{\gamma^3(1-\gamma)^2\max\{1-\gamma, \sigma\}N}} + \left(\frac{2\varepsilon_{\mathsf{opt}}\gamma}{(1-\gamma)} + \frac{14L''C_S\|1\|_*}{N(1-\gamma)^2}\right),\right.$$

$$12\sqrt{\frac{L''\left(1 + \varepsilon_{\mathsf{opt}} + \frac{C_S\|1\|_*}{N(1-\gamma)}\right)}{\gamma^3(1-\gamma)^2\max\{1-\gamma, \sigma\}N}} + 4\sqrt{\frac{L''}{(1-\gamma)^2 N}}\left\|V^{\widehat{\pi},\sigma} - \widehat{V}^{\widehat{\pi},\sigma}\right\|_\infty \tag{110}$$

$$\left.+ \frac{3\varepsilon_{\mathsf{opt}}}{(1-\gamma)} + \frac{14L''C_S\|1\|_*}{N(1-\gamma)^2}\right\}$$

$$\leq 48\sqrt{\frac{L''\left(1 + \varepsilon_{\mathsf{opt}} + \frac{C_S\|1\|_*}{N(1-\gamma)}\right)}{\gamma^3(1-\gamma)^2\max\{1-\gamma, C_g\sigma\}N}} + \frac{6\varepsilon_{\mathsf{opt}}}{(1-\gamma)} + \frac{28L''C_S\|1\|_*}{N(1-\gamma)^2} \tag{111}$$

for $sa$-rectangular and the same quantity, replacing $\max\{1-\gamma, C_g\sigma\}$ by $\max\{1-\gamma, \widetilde{\sigma}\min_s\|\widehat{\pi}_s\|_*\}$ for $s-$ rectangular case. The proof is similar for $TV$ without the geometric term depending on $C_S$.

**Step 6: summing all the previous inequalities results.** Using all the previous results in (90) and (111) and inserting back to (56) complete the proof as follows: taking $N \geq \frac{16L''}{(1-\gamma)^2}, \gamma > 1/4,$ , with

probability at least $1 - \delta$, for $sa$-rectangular

$$
\begin{aligned}
\left\|V^{\star,\sigma} - V^{\widehat{\pi},\sigma}\right\|_\infty &\leq \left\|V^{\pi^\star,\sigma} - \widehat{V}^{\pi^\star,\sigma}\right\|_\infty + \varepsilon_{\mathsf{opt}} + \left\|\widehat{V}^{\widehat{\pi},\sigma} - V^{\widehat{\pi},\sigma}\right\|_\infty \\
&\leq \varepsilon_{\mathsf{opt}} + 48\sqrt{\frac{L''\left(1 + \varepsilon_{\mathsf{opt}} + \frac{C_S\|1\|_*}{N(1-\gamma)}\right)}{\gamma^3(1-\gamma)^2 \max\{1-\gamma, C_g\sigma\}N}} + \frac{6\varepsilon_{\mathsf{opt}}}{(1-\gamma)} + \frac{28L''C_S\|1\|_*}{N(1-\gamma)^2} \\
&\quad + 160\sqrt{\frac{L(1 + \frac{C_S\|1\|_*}{N(1-\gamma)})}{(1-\gamma)^2 \max\{1-\gamma, C_g\sigma\}N}} + \frac{14LC_S\|1\|_*}{N(1-\gamma)^2} \\
&\leq \frac{8\varepsilon_{\mathsf{opt}}}{1-\gamma} + \frac{42L''C_S\|1\|_*}{N(1-\gamma)^2} + 1508\sqrt{\frac{L''(1 + \frac{C_S\|1\|_*}{N(1-\gamma)})}{(1-\gamma)^2 \max\{1-\gamma, C_g\sigma\}N}}
\end{aligned}
$$
(112)

where the last inequality holds by $\gamma \geq \frac{1}{4}$ and $N \geq \frac{16L''}{(1-\gamma)^2}$ for $sa$-rectangular and the same quantity replacing $\max\{1-\gamma, \sigma\}$ by $\max\{1-\gamma, \tilde{\sigma}\min_s\{\|\pi_s^*\|_*\}\}$ for $s-$ rectangular case. The proof is similar for $TV$ without the geometric term depending on $C_S$.

### D.3 Proof of the auxiliary lemmas

#### D.3.1 Proof of Lemma 5

Similarly to Shi et al. [2023], denoting $s_0$ the argmax of $V^{\pi,\sigma}$ such that $V^{\pi,\sigma}(s_0) = \min_{s \in \mathcal{S}} V^{\pi,\sigma}(s)$ using recursive Bellman's equation

$$
\max_{s \in \mathcal{S}} V^{\pi,\sigma}(s) = \max_{s \in \mathcal{S}} \mathbb{E}_{a \sim \pi(\cdot|s)}\left[r(s,a) + \gamma \inf_{\mathcal{P} \in \mathcal{U}^\sigma(P_{s,a})} \mathcal{P}V^{\pi,\sigma}\right] \tag{113}
$$

$$
\leq \max_{(s,a) \in \mathcal{S} \times \mathcal{A}} \left(1 + \gamma \inf_{\mathcal{P} \in \mathcal{U}^\sigma(P_{s,a})} \mathcal{P}V^{\pi,\sigma}\right) \tag{114}
$$

where the second line holds since the reward function $r(s,a) \in [0,1]$ for all $(s,a) \in \mathcal{S} \times \mathcal{A}$.

Then we construct for any $(s,a) \in \mathcal{S} \times \mathcal{A}$, $\widetilde{P}_{s,a} \in \mathbb{R}^S$ by reducing the values of some elements of $P_{s,a}$ such that $P_{s,a} \geq \widetilde{P}_{s,a} \geq 0$ and $\sum_{s'}\left(P_{s,a}(s') - \widetilde{P}_{s,a}(s')\right) = \sigma C_g^{s,a}$. with $C_g^{s,a} = \frac{1}{\|e_{s_0}\|}$ It lead to $\widetilde{P}_{s,a} + \sigma C_g^{s,a}e_{s_0}^\top \in \mathcal{U}_{\|\cdot\|}^\sigma(P_{s,a})$, where $e_{s_0}$ is the standard basis vector supported on $s_0$, since

$$
\frac{1}{2}\left\|\widetilde{P}_{s,a} + \sigma C_g^{s,a}e_{s_0}^\top - P_{s,a}\right\| \leq \frac{1}{2}\left\|\widetilde{P}_{s,a} - P_{s,a}\right\| + \frac{C_g^{s,a}\sigma\|e_{s_0}\|}{2} = \sigma/2 + \sigma/2 = \sigma \tag{115}
$$

Consequently,

$$
\inf_{\mathcal{P} \in \mathcal{U}_{\|\cdot\|}^\sigma(P_{s,a})} \mathcal{P}V^{\pi,\sigma} \leq \left(\widetilde{P}_{s,a} + \sigma C_g^{s,a}e_{s_0}^\top\right)V^{\pi,\sigma} \leq \left\|\widetilde{P}_{s,a}\right\|_1 \|V^{\pi,\sigma}\|_\infty + \sigma V^{\pi,\sigma}(s_0)C_g^{s,a} \tag{116}
$$

$$
\leq (1 - C_g^{s,a}\sigma)\max_{s \in \mathcal{S}} V^{\pi,\sigma}(s) + \sigma C_g^{s,a}\min_{s \in \mathcal{S}} V^{\pi,\sigma}(s) \tag{117}
$$

where the second inequality holds by

$$
\left\|\widetilde{P}_{s,a}\right\|_1 = \sum_{s'}\widetilde{P}_{s,a}(s') = -\sum_{s'}\left(P_{s,a}(s') - \widetilde{P}_{s,a}(s')\right) + \sum_{s'}P_{s,a}(s') = 1 - \sigma C_g^{s,a} \tag{118}
$$

Plugging this back to the previous relation gives

$$\max_{s \in \mathcal{S}} V^{\pi,\sigma}(s) \leq 1 + \gamma(1 - C_g^{s,a}\sigma)\max_{s \in \mathcal{S}} V^{\pi,\sigma}(s) + \gamma C_g^{s,a}\sigma \min_{s \in \mathcal{S}} V^{\pi,\sigma}(s) \tag{119}$$

which, by rearranging terms, yields

$$\max_{s \in \mathcal{S}} V^{\pi,\sigma}(s) \leq \frac{1 + \gamma C_g^{s,a}\sigma \min_{s \in \mathcal{S}} V^{\pi,\sigma}(s)}{1 - \gamma(1 - C_g^{s,a}\sigma)} \tag{120}$$

$$\leq \frac{1}{(1-\gamma) + \gamma C_g^{s,a}\sigma} + \min_{s \in \mathcal{S}} V^{\pi,\sigma}(s) \leq \frac{1}{\gamma \max\{1-\gamma, C_g^{s,a}\sigma\}} + \min_{s \in \mathcal{S}} V^{\pi,\sigma}(s) \tag{121}$$

So rearranging terms it holds :

$$\mathrm{sp}(V^{\pi,\sigma})_\infty \leq \frac{1}{\gamma \max\{1-\gamma, C_g^{s,a}\sigma\}} \tag{122}$$

or taking the sup over $s$:

$$\mathrm{sp}(V^{\pi,\sigma})_\infty \leq \frac{1}{\gamma \max\{1-\gamma, C_g\sigma\}} \tag{123}$$

As we pick the supreme over $s$, the quantity, $C_g^{s,a}$ is replaced by $C_g = 1/(\min_s \|e_s\|)$ to obtain a control for every $s$.

### D.3.2 Proof of Lemma 6

Similarly to 5 denoting $s_0$ the argmax of $V^{\pi,\sigma}$ such that $V^{\pi,\sigma}(s_0) = \min_{s \in \mathcal{S}} V^{\pi,\sigma}(s)$ using recursive Bellman's equation

$$\max_{s \in \mathcal{S}} V^{\pi,\sigma}(s) = \max_{s \in \mathcal{S}} \mathbb{E}_{a \sim \pi(\cdot|s)}\left[ r(s,a) + \gamma \inf_{\mathcal{P} \in \mathcal{U}^{\tilde{\sigma}}(P_s)} \mathcal{P}V^{\pi,\tilde{\sigma}} \right] \tag{124}$$

$$\leq \max_{s \in \mathcal{S}}\left( 1 + \gamma \inf_{\mathcal{P}^\pi \in \mathcal{U}^{\tilde{\sigma}}(P_s^\pi)} \mathcal{P}^\pi V^{\pi,\tilde{\sigma}} \right) \tag{125}$$

where the second line holds since the reward function $r(s,a) \in [0,1]$ for all $(s,a) \in \mathcal{S} \times \mathcal{A}$. Then we construct for any $s \in \mathcal{S}$ $\widetilde{P}_s \in \mathbb{R}^{S \times A}$ by reducing the values of some elements of $P_s$ such that $P_s \geq \widetilde{P}_s \geq 0$ and

$$\forall a \in A, \quad \sum_{s'}\left( P_s(s',a) - \widetilde{P}_s(s',a) \right) = \sigma_{s,a}C_g^s$$

where $C_g^s$ is defined as $1/\|e_s\|$. Writting $\|\sigma_{s,a}\| \leq \tilde{\sigma}$ we construction $\sigma_{s,a}$ such that

$$\sum_a \pi(a|s) \sum_{s'}\left( P_s(s',a) - \widetilde{P}_s(s',a) \right) = \|\pi_s\|_* \tilde{\sigma}C_g^s. \tag{126}$$

Not that this construction is possible as it is simply Cauchy Swartz equality case. It leads to $\widetilde{P}_s + \sigma e_{s_0,a}^\top \in \mathcal{U}^{\tilde{\sigma}}(P_s)$, where $e_{s_0,a} \in \mathbb{R}^{S \times A}$ is the standard basis vector supported on $s_0$ which is equal to 1 at $s_0$ for every $a$ and otherwise.

$$\frac{1}{2}\left\| \widetilde{P}_s + \sigma_{s,a}C_g^s e_{s_0,a}^\top - P_s \right\| \leq \frac{1}{2}\left\| \widetilde{P}_s - P_s \right\| + \frac{\tilde{\sigma}\|e_{s_0}\|C_g^s}{2} = \tilde{\sigma}/2 + \tilde{\sigma}/2 \tag{127}$$

as $C_g^s \left\| \sigma_{s,a} e_{s_0,a} \right\|$ is equal to $C_g^s \tilde{\sigma} \left\| e_{s_0} \right\|$ Consequently,

$$\inf_{\mathcal{P}^\pi \in \mathcal{U}^\sigma(P_s)} \mathcal{P}^\pi V^{\pi,\tilde{\sigma}} \le \Pi^\pi \left( \widetilde{P}_s^\pi + \sigma C_g^s e_{s_0}^\top \right) V^{\pi,\tilde{\sigma}} \tag{128}$$

$$= \sum_a \sum_{s'} \widetilde{P}_s(s',a) \pi(a|s) V^{\pi,\tilde{\sigma}}(s') + \sigma e_{s_0,a} C_g^s V^{\pi,\tilde{\sigma}}(s_0) \pi(a|s) \tag{129}$$

$$\le \sum_a \sup_{s'}[V^{\pi,\tilde{\sigma}}(s')] (\sum_{s'} \widetilde{P}_s(s',a))) \pi(a|s) + V^{\pi,\tilde{\sigma}}(s_0) \pi(a|s) \sigma_{s,a} C_g^s \tag{130}$$

$$\overset{(a)}{=} \max_{s \in \mathcal{S}} V^{\pi,\sigma}(s) \sum_a (1 - \sigma C_g^s) \pi(a|s) + \sum_a V^{\pi,\tilde{\sigma}}(s_0) \pi(a|s) \sigma_{s,a} C_g^s \tag{131}$$

$$\overset{(b)}{=} \max_{s \in \mathcal{S}} V^{\pi,\sigma}(s)(1 - \tilde{\sigma} C_g^s) \left\| \pi_s \right\|_* + \left\| \pi_s \right\|_* \tilde{\sigma} C_g^s \min_{s \in \mathcal{S}} V^{\pi,\tilde{\sigma}}(s) \tag{132}$$

$$\le (1 - C_g^s \tilde{\sigma}) \max_{s \in \mathcal{S}} V^{\pi,\sigma}(s) + \sigma C_g^s \min_{s \in \mathcal{S}} V^{\pi,\tilde{\sigma}}(s) \tag{133}$$

where $\left\| \pi \right\|_\infty$ is the norm of the vector $\pi(.|s)$ and where (a) holds because

$$\sum_{s'} \widetilde{P}_s(s') = - \sum_{s'} \left( P_s(s') - \widetilde{P}_s(s') \right) + \sum_{s'} P_s(s') = 1 - \sigma_{s,a} C_g^s \tag{134}$$

Finally (b) is due to (126) and using Holder's inequality in the second term. Plugging this back to the previous relation gives

$$\max_{s \in \mathcal{S}} V^{\pi,\tilde{\sigma}}(s) \le 1 + \gamma(1 - \tilde{\sigma} C_g^s \left\| \pi_s \right\|_*) \max_{s \in \mathcal{S}} V^{\pi,\sigma}(s) + \gamma \left\| \pi_s \right\|_* \tilde{\sigma} C_g^s \min_{s \in \mathcal{S}} V^{\pi,\tilde{\sigma}}(s) \tag{135}$$

which, by rearranging terms, yields

$$\max_{s \in \mathcal{S}} V^{\pi,\tilde{\sigma}}(s) \le \frac{1 + \gamma \tilde{\sigma} \left\| \pi_s \right\|_* C_g^s \min_{s \in \mathcal{S}} V^{\pi,\tilde{\sigma}}(s)}{1 - \gamma(1 - C_g^s \tilde{\sigma} \left\| \pi_s \right\|_*)} \tag{136}$$

$$\le \frac{1}{(1 - \gamma) + \left\| \pi_s \right\|_* \gamma C_g^s \tilde{\sigma}} + \min_{s \in \mathcal{S}} V^{\pi,\tilde{\sigma}}(s) \tag{137}$$

$$\le \frac{1}{(1 - \gamma) + \gamma \left\| \pi_s \right\|_* C_g^s \tilde{\sigma}} + \min_{s \in \mathcal{S}} V^{\pi,\tilde{\sigma}}(s) \tag{138}$$

$$\le \frac{1}{\gamma \max\{1 - \gamma, C_g^s \left\| \pi_s \right\|_* \tilde{\sigma}\}} + \min_{s \in \mathcal{S}} V^{\pi,\tilde{\sigma}}(s). \tag{139}$$

So rearranging and taking the sumpremum over all $s$term it holds :

$$\mathrm{sp}(V^{\pi,\tilde{\sigma}})_\infty \le \frac{1}{\gamma \max\{1 - \gamma, \min_s \left\| \pi_s \right\|_* C_g \tilde{\sigma}\}}. \tag{140}$$

As we pick the supreme over $s$ ovf this quantity, $C_g^s$ is replaced by $C_g = 1/\min_s \left\| e_s \right\|$.

### D.3.3 Proof of Lemma 8

*Proof.* Concentration of the robust values function. with probability $1 - \delta$, it holds:

$$\left| P_{s,a}^{\pi,V} V - \widehat{P}_{s,a}^{\pi,V} V \right| \le 2 \sqrt{\frac{L}{N}} \sqrt{\mathrm{Var}_{P_{s,a}^0}(V)} + \frac{3LC_S \left\| 1 \right\|_*}{N(1 - \gamma)}$$

with $L = 2\log(18\,\|1\|_* \, SAN/\delta)$ and First we can use optimization duality such as in (50):

$$\left| P^{\pi,V}_{s,a} V - \widehat{P}^{\pi,V}_{s,a} V \right| \tag{141}$$

$$= \left| \max_{\mu^{\lambda,\omega}_{P^0_{s,a}} \in \mathcal{M}^{\lambda,\omega}_{P^0_{s,a}}} \left\{ P^0_{s,a}(V - \mu) - \sigma\left( \mathrm{sp}((V - \mu))_* \right) \right\} \right.$$

$$\left. - \max_{\mu^{\lambda,\omega}_{\widehat{P}^0_{s,a}} \in \mathcal{M}^{\lambda,\omega}_{\widehat{P}^0_{s,a}}} \left\{ \widehat{P}^0_{s,a}(V - \mu^{\lambda,\omega}_{\widehat{P}^0_{s,a}}) - \sigma\left( \mathrm{sp}((V - \mu^{\lambda,\omega}_{\widehat{P}^0_{s,a}}))_* \right) \right\} \right|$$

$$\leq \max \left\{ \left| \max_{\mu^{\lambda,\omega}_{P^0_{s,a}} \in \mathcal{M}^{\lambda,\omega}_{P^0_{s,a}}} \left\{ P^0_{s,a}(V - \mu^{\lambda,\omega}_{P^0_{s,a}}) - \sigma\left( \mathrm{sp}((V - \mu^{\lambda,\omega}_{P^0_{s,a}}))_* \right) \right\} \right. \right.$$

$$\left. - \max_{\mu^{\lambda,\omega}_{P^0_{s,a}} \in \mathcal{M}^{\lambda,\omega}_{P^0_{s,a}}} \left\{ \widehat{P}^0_{s,a}(V - \mu^{\lambda,\omega}_{P^0_{s,a}}) - \sigma\left( \mathrm{sp}((V - \mu^{\lambda,\omega}_{P^0_{s,a}}))_* \right) \right\} \right|; \tag{142}$$

$$\left| \max_{\mu^{\lambda,\omega}_{\widehat{P}^0_{s,a}} \in \mathcal{M}^{\lambda,\omega}_{\widehat{P}^0_{s,a}}} \left\{ \widehat{P}^0_{s,a}(V - \mu^{\lambda,\omega}_{\widehat{P}^0_{s,a}}) - \sigma\left( \mathrm{sp}((V - \mu^{\lambda,\omega}_{\widehat{P}^0_{s,a}}))_* \right) \right\} \right. \tag{143}$$

$$\left. - \max_{\mu^{\lambda,\omega}_{\widehat{P}^0_{s,a}} \in \mathcal{M}^{\lambda,\omega}_{\widehat{P}^0_{s,a}}} \left\{ P^0_{s,a}(V - \mu^{\lambda,\omega}_{\widehat{P}^0_{s,a}}) - \sigma\left( \mathrm{sp}((V - \mu^{\lambda,\omega}_{\widehat{P}^0_{s,a}}))_* \right) \right\} \right| \right\}$$

$$\leq \max \left\{ \underbrace{\left| \max_{\mu \in \mu^{\lambda,\omega}_{P^0_{s,a}}} \left( P^0_{s,a} - \widehat{P}^0_{s,a} \right) (V - \mu^{\lambda,\omega}_{P^0_{s,a}}) \right|}_{=:g_{s,a}(\alpha^{\lambda,\omega}_P, V)}, \underbrace{\left| \max_{\mu^{\lambda,\omega}_{\widehat{P}^0_{s,a}} \in \mathcal{M}^{\lambda,\omega}_{\widehat{P}^0_{s,a}}} \left( P^0_{s,a} - \widehat{P}^0_{s,a} \right) (V - \mu^{\lambda,\omega}_{\widehat{P}^0_{s,a}}) \right|}_{=:g_{s,a}(\alpha^{\lambda,\omega}_{\widehat{P}}, V)} \right\} \tag{144}$$

where in the first equality we use Lemma 3. The final inequality is a consequence of the 1-Lipschitzness of the max operator. First, we control $g_{s,a}(\alpha^{\lambda,\omega}_P, V)$. To do so, we use for a fixed $\alpha^{\lambda,\omega}_P$ and any vector $V$ that is independent with $\widehat{P}^0$, the Bernstein's inequality, one has with probability at least $1 - \delta$ with $sa$-rectangular notations,

$$g_{s,a}(\alpha^{\lambda,\omega}_P, V) = \left| \left( P^0_{s,a} - \widehat{P}^0_{s,a} \right) [V]_{\alpha^{\lambda,\omega}_P} \right| \leq \sqrt{\frac{2\log(\frac{2}{\delta})}{N}} \sqrt{\mathrm{Var}_{P^0_{s,a}}(V)} + \frac{2\log(\frac{2}{\delta})}{3N(1-\gamma)}. \tag{145}$$

Once pointwise concentration derived, we will use uniform concentration to yield this lemma. First, union bound, is obtained noticing that $g_{s,a}(\alpha^{\lambda,\omega}_P, V)$ is 1-Lipschitz w.r.t. $\lambda$ and $\omega$ as it is linear in $\lambda$ and $\omega$. Moreover, $\lambda^* = \|V - \mu^* - \omega\|_*$ obeying $\lambda^* \leq \frac{\|1\|_*}{1-\gamma}$. The quantity $\omega \in [0, 1/(1-\gamma)]$ as it is always smaller that $V$ by definition. We construct then a 2-dimensional a $\varepsilon_1$-net $N_{\varepsilon_1}$ over $\lambda^* \in [0, \frac{\|1\|_*}{1-\gamma}]$ and $\omega \in [0, 1/(1-\gamma)]$ whose size satisfies $|N_{\varepsilon_1}| \leq \left( \frac{3\|1\|_*}{\varepsilon_1(1-\gamma)} \right)^2$ [Vershynin, 2018]. Using union bound and (145), it holds with probability at least $1 - \frac{\delta}{SA}$ that for all $\lambda \in N_{\varepsilon_1}$,

$$g_{s,a}(\alpha^\lambda_P, V) \leq \sqrt{\frac{2\log(\frac{2SA|N_{\varepsilon_1}|}{\delta})}{N}} \sqrt{\mathrm{Var}_{P^0_{s,a}}(V)} + \frac{2\log(\frac{2SA|N_{\varepsilon_1}|}{\delta})}{3N(1-\gamma)}. \tag{146}$$

Using the previous equation and also (144), it results in using notation $2\log(\frac{18SAN\|1\|_*}{\delta}) = L$,

$$g_{s,a}(\alpha_P^\lambda, V) \overset{(a)}{\leq} \sup_{\alpha_P^\lambda \in N_{\varepsilon_1}} \left| \left(P_{s,a}^0 - \widehat{P}_{s,a}^0\right) [V]_{\alpha_P^\lambda} \right| + \varepsilon_1$$

$$\overset{(b)}{\leq} \sqrt{\frac{2\log(\frac{SA|N_{\varepsilon_1}|}{\delta})}{N}} \sqrt{\mathrm{Var}_{P_{s,a}^0}(V)} + \frac{2\log(\frac{2SA|N_{\varepsilon_1}|}{\delta})}{3N(1-\gamma)} + \varepsilon_1 \tag{147}$$

$$\overset{(c)}{\leq} \sqrt{\frac{2\log(\frac{2SA|N_{\varepsilon_1}|}{\delta})}{N}} \sqrt{\mathrm{Var}_{P_{s,a}^0}(V)} + \frac{\log(\frac{2SA|N_{\varepsilon_1}|}{\delta})}{N(1-\gamma)}$$

$$\overset{(d)}{\leq} \sqrt{\frac{L}{N}} \sqrt{\mathrm{Var}_{P_{s,a}^0}(V)} + \frac{L}{N(1-\gamma)} \tag{148}$$

$$\leq \sqrt{\frac{L}{N}} \|V\|_\infty + \frac{L}{N(1-\gamma)}$$

$$\leq 2\sqrt{\frac{L}{(1-\gamma)^2 N}} \tag{149}$$

where (a) is because the optimal $\alpha$ falls into the $\varepsilon_1$-ball centered around some point inside $N_{\varepsilon_1}$ and $g_{s,a}(\alpha_P^\lambda, V)$ is 1-Lipschitz with regard to $\lambda$ and $\omega$, (b) is due to Eq. (146), (c) arises from taking $\varepsilon_1 = \frac{\log(\frac{2SA|N_{\varepsilon_1}|}{\delta})}{3N(1-\gamma)}$, (d) is verified by $|N_{\varepsilon_1}| \leq \left(\frac{3\|1\|_*}{\varepsilon_1(1-\gamma)}\right)^2 \leq 9N\|1\|$ and that variance of a ceiling function of a vector is smaller than the variance of non-ceiling vector , and the last inequality comes from the fact $\|V^{\star,\sigma}\|_\infty \leq \frac{1}{1-\gamma}$ and taking $N \geq 2\log(\frac{18SAN\|1\|_*}{\delta}) = L$.

Contrary to the previous term, the second term $g_{s,a}(\alpha_{\widehat{P}}^\lambda, V)$ is more difficult as we need concentration. Still, the data has an extra dependency through the parameter $\alpha_{\widehat{P}}^\lambda$. We need to decouple this problem using absorbing MDPs. Then it leads to

$$g_{s,a}(\alpha_{\widehat{P}}^{\lambda,\omega}, V) \tag{150}$$

$$= |\max_{\mu_{\widehat{P}_{s,a}^0}^{\lambda,\omega} \in \mathcal{M}_{\widehat{P}_{s,a}^0}^{\lambda,\omega}} \left(P_{s,a}^0 - \widehat{P}_{s,a}^0\right) (V - \mu_{\widehat{P}_{s,a}^0}^{\lambda,\omega})| \tag{151}$$

$$= |\max_{\mu \in \mathcal{M}_{\widehat{P}_{s,a}^0}^{\lambda,\omega}} \left(P_{s,a}^0 - \widehat{P}_{s,a}^0\right) (V - \mu_{P_{s,a}^0}^{\lambda,\omega}) + \left(P_{s,a}^0 - \widehat{P}_{s,a}^0\right) (\mu_{P_{s,a}^0}^{\lambda,\omega} - \mu_{\widehat{P}_{s,a}^0}^{\lambda,\omega})| \tag{152}$$

$$\leq |\max_{\mu_{P_{s,a}^0}^{\lambda,\omega} \in \mathcal{M}_{P_{s,a}^0}^{\lambda,\omega}} \left(P_{s,a}^0 - \widehat{P}_{s,a}^0\right) (V - \mu_{P_{s,a}^0}^{\lambda,\omega}) + \max_{\mu_{\widehat{P}_{s,a}^0}^{\lambda,\omega} \in \mathcal{M}_{\widehat{P}_{s,a}^0}^{\lambda,\omega}} \left(P_{s,a}^0 - \widehat{P}_{s,a}^0\right) (\mu_{P_{s,a}^0}^{\lambda,\omega} - \mu_{\widehat{P}_{s,a}^0}^{\lambda,\omega})|.$$

$$\tag{153}$$

In the first equality, we add the term $\mu_{P_{s,a}^0}^{\lambda,\omega}$ to retrieve the previous concentration problem, fixing $P_{s,a}^0$ and optimizing $\lambda, \omega$. In the second, we extend the max using triangular inequality. The first term in the last equality is exactly the term we have controlled previously, while the second one needs more attention. We decouple the data's dependency, then control the difference between the $\mu$. Then using the characterization of the optimal $\mu$ from equation (47):

$$\left(P_{s,a}^0 - \widehat{P}_{s,a}^0\right) (\mu_{P_{s,a}^0}^{\lambda,\omega} - \mu_{\widehat{P}_{s,a}^0}^{\lambda,\omega}) = \sum_{s'} \lambda \left(P_{s,a}^0(s') - \widehat{P}_{s,a}^0(s')\right) (\nabla\|P_{s,a}^0\| - \nabla\|\widehat{P}_{s,a}^0\|)$$

Here we assume that the subgradient is a gradient as we assume that the norm is $C^2$. The question that arises is whether the gradient of the norm is Lipschitz.

Note that we are considering the worst case as $(\mu_{P_{s,a}^0}^{\lambda,\omega} - \mu_{\widehat{P}_{s,a}^0}^{\lambda,\omega})$ can be zero in the case where $\mu$ the Lagrangian variable is equal to zero. Finally, note that we can also control this term when one of

the two terms $\mu_{P^0_{s,a}}^{\lambda,\omega}$ or $\mu_{\hat{P}^0_{s,a}}^{\lambda,\omega}$ is equal to zero as $\mu_{\hat{P}^0_{s,a}}^{\lambda,\omega}$ and $\mu_{P^0_{s,a}}^{\lambda,\omega}$ smaller that $V$ because $V - \mu$ need to be positive in equation (43). In this case, classical control using Bernstein's inequality without uniform concentration can be applied, giving the same result. In the worst case where all terms in $(\mu_{P^0_{s,a}}^{\lambda,\omega} - \mu_{\hat{P}^0_{s,a}}^{\lambda,\omega})$ are non zero, assuming that the norm is $C^2$, using mean value theorem, we know that

$$\left\| \left( \nabla \| P^0_{s,a} \| - \nabla \left\| \hat{P}^0_{s,a} \right\| \right) \right\|_2 \leq \sup_{x \in \Delta(S)} \left\| \nabla^2 \| x \| \right\|_2 \left\| \left( P^0_{s,a} - \hat{P}^0_{s,a} \right) \right\|_2.$$

As the norm is $C^2$, is continuous and as the simplex is bounded, this quantity exists according to the Extreme value theorem. It is possible to compute this contact depending on $S$ for explicit norms such as $L_p$. Indeed, for $L_2$:

$$\nabla^2 \| x \|_2 = \frac{\left( I - \frac{x \otimes x}{\|x\|_2^2} \right)}{\|x\|_2} \leq \frac{1}{\|x\|_2} I \leq \frac{1}{\min_{x \in \Delta(S)} \|x\|_2} I = \sqrt{S}$$

where $\otimes$ is the Kronecker product. So we have an upper bound independent of $x$. For $L_p = \|x\|_p$ norms, $p \geq 2$, we have simple taking derivative twice:

$$\nabla^2 \| x \|_p = \frac{p-1}{L_p} \left( \mathcal{A}^{p-2} - g_p g_p^T \right)$$

with

$$\mathcal{A} = \mathrm{Diag} \left( \frac{\mathrm{abs}(x)}{L_p} \right)$$

$$g_p = \mathcal{A}^{p-2} \left( \frac{x}{L_p} \right).$$

where $\mathrm{Diag}$ is the diagonal matrix. However, as $x \leq L_p$, $\mathcal{A} \leq I$, we get

$$H \leq \frac{p-1}{\|x\|_p} \leq (p-1)S^{1/q} = C_S \tag{154}$$

where the $1/L_p$ is minimized for the uniform distribution. Then using Cauchy Swartz inequality, it holds

$$\left( P^0_{s,a} - \hat{P}^0_{s,a} \right) \left( \mu_{P^0_{s,a}}^{\lambda,\omega} - \mu_{\hat{P}^0_{s,a}}^{\lambda,\omega} \right) \leq \lambda \left\| \left( P^0_{s,a} - \hat{P}^0_{s,a} \right) \right\|_2^2. \tag{155}$$

Then the question is how to bound the quantity $\left\| \left( P^0_{s,a} - \hat{P}^0_{s,a} \right) \right\|_2^2$. To do so, we will use McDiarmid inequality.

**Definition 3.** *Bounded difference property*

*A function $f : \mathcal{X}_1 \times \ldots \mathcal{X}_n \to \mathbb{R}$ satisfies the bounded difference property if for each $i = 1, \ldots, n$ the change of coordinate from $s_i$ to $s'_i$ may change the value of the function at most on $c_i$*

$$\forall i \in [n] : \sup_{x'_i \in \mathcal{X}_i} |f(x_1, \ldots, x_i, \ldots, x_n) - f(x_1, \ldots, x'_i, \ldots, x_n)| \leq c_i$$

In our case, we consider $f(X_1, \ldots, X_n) = \| \sum_{k=1}^n X_k \|_2$. Then we can notice that by triangle inequality for any $x_1, \ldots, x_n$ and $x'_k$ with $X_{i,s'} = P^0_{i,s,a}(s') - P^0_{s,a}(s')$ ( index $i$ holds for index of sample generated from the generative model) that

$$f(x_1, \ldots, x_k, \ldots, x_n) = \|x_1 + \ldots + x_n\|_2 \leq \|x_1 + \ldots + x_n - x_k + x'_k\|_2 + \|x_k - x'_k\|_2$$
$$\leq f(x_1, \ldots, x'_k, \ldots, x_n) + 2$$

**Theorem 5.** *(McDiarmid's inequality). McDiarmid et al. [1989] Let $f : \mathcal{X}_1 \times \ldots \mathcal{X}_n \to \mathbb{R}$ be a function satisfying the bounded difference property with bounds $c_1, \ldots, c_n$. Consider independent random variables $X_1, \ldots, X_n, X_i \in \mathcal{X}_i$ for all $i$. Then for any $t > 0$*

$$\mathbb{P}\left[ f\left(X_1, \ldots, X_n\right) - \mathbb{E}\left[ f\left(X_1, \ldots, X_n\right)\right] \geq t \right] \leq \exp\left( -\frac{2t^2}{\sum_{i=1}^n c_i^2} \right)$$

Using McDiarmid's inequality and union bound, we can bound the term here

$$\left( \left\| \left( P_{s,a}^0 - \widehat{P}_{s,a}^0 \right) \right\|_2 - \mathbb{E}\left[ \left\| \left( P_{s,a}^0 - \widehat{P}_{s,a}^0 \right) \right\|_2 \right] \right)^2 \leq \frac{2N \log(|S||A|/\delta))}{N^2}$$

with probability $1 - \delta/(|S||A|)$. Moreover, the additional term can be bounded as follows:

$$\mathbb{E}\left[ \left\| \left( P_{s,a}^0 - \widehat{P}_{s,a}^0 \right) \right\|_2^2 \right] = \mathbb{E}\left[ \sum_{s'} (P_{s,a}^0(s') - P_{s,a}^0(s'))^2 \right] = \mathbb{E}\left[ \sum_{s'} (\frac{1}{N} \sum_i^N X_{i,s'})^2 \right]$$

with $X_{i,s'} = P_{i,s,a}^0(s') - P_{s,a}^0(s')$ is one sample sampled from the generative model. Then

$$\mathbb{E}\left[ \left\| \left( P_{s,a}^0 - \widehat{P}_{s,a}^0 \right) \right\|_2^2 \right] = \frac{1}{N^2} \sum_{s'} \mathsf{Var}(\sum_i^N X_{i,s}) \overset{a}{=} \frac{1}{N^2} \sum_i^N \sum_{s'} \mathsf{Var}(X_{i,s}) \tag{156}$$

$$= \frac{1}{N^2} \sum_i^N \mathbb{E}(\sum_{s'} X_{i,s}^2) \leq \frac{4}{N} \tag{157}$$

where (a) the last equality comes from the independence of the random variables, and where the last inequality comes from the fact the maximum of two elements in the simplex is bounded by 2.

Moreover, we know that,

$$\mathbb{E}\left[ \left\| \left( P_{s,a}^0 - \widehat{P}_{s,a}^0 \right) \right\|_2 \right]^2 \leq \mathbb{E}\left[ \left\| \left( P_{s,a}^0 - \widehat{P}_{s,a}^0 \right) \right\|_2^2 \right] \tag{158}$$

due to Jensen's inequality. Finally, regrouping the two terms, we obtain with probability $1 - \delta/(|S||A|)$:

$$\left\| \left( P_{s,a}^0 - \widehat{P}_{s,a}^0 \right) \right\|_2^2 = \left( \left\| \left( P_{s,a}^0 - \widehat{P}_{s,a}^0 \right) \right\|_2 - \mathbb{E}\left[ \left\| \left( P_{s,a}^0 - \widehat{P}_{s,a}^0 \right) \right\|_2 \right] \right)^2 + \left( \mathbb{E}\left[ \left\| \left( P_{s,a}^0 - \widehat{P}_{s,a}^0 \right) \right\|_2 \right] \right)^2$$
$$+ 2\mathbb{E}\left[ \left\| \left( P_{s,a}^0 - \widehat{P}_{s,a}^0 \right) \right\|_2 \right] \left( \left\| \left( P_{s,a}^0 - \widehat{P}_{s,a}^0 \right) \right\|_2 - \mathbb{E}\left[ \left\| \left( P_{s,a}^0 - \widehat{P}_{s,a}^0 \right) \right\|_2 \right] \right)$$
$$\overset{a}{\leq} \frac{2N \log(|S||A|/(\delta))}{N^2} + \frac{4}{N} + \frac{\sqrt{\frac{4}{N}} \sqrt{2N \log(|S||A|/(\delta))}}{N}$$
$$\leq \frac{10 \log(|S||A|/(\delta))}{N} = \frac{L'}{N}$$

where in first inequality use $(a+b)^2 = a^2 + b^2 + 2ab$ and where in (a) we combine equation (158) and (157) and (156).

with $L' = 10 \log(|S||A|/(\delta))$. Finally, plugging the previous equation in (155):

$$\max_{\mu \in \mu_{\widehat{P}_{s,a}^0}^\lambda} \left( P_{s,a}^0 - \widehat{P}_{s,a}^0 \right) (\mu_{P_{s,a}^0}^\lambda - \mu)| \leq \max_\lambda \left\| \left( P_{s,a}^0 - \widehat{P}_{s,a}^0 \right) \right\|_2^2 C_S \lambda.$$

This term can be easily controlled by taking the supremum over $\lambda$, which is a 1 dimensional parameter. Then we can bound $\lambda \in [0, H \left\| 1 \right\|_*]$. Indeed,

$$\lambda^* = \left\| V - \mu^* - \eta \right\|_* \leq \left\| V \right\|_* \leq H \left\| 1 \right\|_*.$$

Finally, we obtain:

$$\max_\lambda \left\| \left(P^0_{s,a} - \widehat{P}^0_{s,a}\right) \right\|^2_2 C_S \lambda \leq \frac{L' C_S \|1\|_*}{N(1-\gamma)}.$$

Regrouping all terms:

$$g_{s,a}(\alpha^\lambda_{\widehat{P}}, V) \leq | \max_{\mu^\lambda_{P^0_{s,a}} \in \mathcal{M}^\lambda_{P^0_{s,a}}} \left(P^0_{s,a} - \widehat{P}^0_{s,a}\right)(V - \mu^\lambda_{P^0_{s,a}}) + \max_{\mu^\lambda_{\widehat{P}^0_{s,a}} \in \mathcal{M}^\lambda_{\widehat{P}^0_{s,a}}} \left(P^0_{s,a} - \widehat{P}^0_{s,a}\right)(\mu^\lambda_{P^0_{s,a}} - \mu^\lambda_{\widehat{P}^0_{s,a}})|$$

$$\leq 2\sqrt{\frac{L}{N}}\sqrt{\mathrm{Var}_{P^0_{s,a}}(V)} + \frac{L' C_S \|1\|_*}{N(1-\gamma)} + \frac{L}{N(1-\gamma)}$$

$$\leq 2\sqrt{\frac{L}{N}}\sqrt{\mathrm{Var}_{P^0_{s,a}}(V)} + \frac{3L C_S \|1\|_*}{N(1-\gamma)} \tag{159}$$

$$\tag{160}$$

We can recognize that the second term is a second-order term as long as $N \geq (C_S \|1\|_*)^2$, we can regroup the two terms. Finally, as $g_{s,a}(\alpha^\lambda_{\widehat{P}}, V) \geq g_{s,a}(\alpha^\lambda_P, V)$, we obtain

$$\left| P^{\pi,V}_{s,a} V - \widehat{P}^{\pi,V}_{s,a} V \right| \leq 2\sqrt{\frac{L}{N}}\sqrt{\mathrm{Var}_{P^0_{s,a}}(V)} + \frac{3L C_S \|1\|_*}{N(1-\gamma)} \tag{161}$$

It is important to note that the geometry of the norm is present in the second order term $\frac{3L C_S \|1\|}{N(1-\gamma)}$ but this term is negligible as it is proportional to $1/N$ with regard to the variance term in $1/\sqrt{N}$. Moreover, note that the quantity $C_S \|1\|_* = S$ for $L_2$ norms.

For the specific case of $TV$ which is not $C^2$ smooth, this lemma still holds as in (144), we only need to control one term without the dependency on data in the supremum as $\alpha^\lambda_P$ reduces to a scalar $\alpha$ which does not depend on $P$. Then extra decomposition using smoothness of the norm is not needed, as the only remaining term in the max in (144) is the left-hand side term.

For the $s$-rectangular case, the first equation can be rewritten simply by factorizing by $\pi(a|s)$ using lemma 4.

$$\left| P^{\pi,V}_{s,a} V - \widehat{P}^{\pi,V}_{s,a} V \right| = \left| \sum_a \pi(a|s) \max_{\mu^\lambda_{P^0_{s,a}} \in \mathcal{M}^\lambda_{P^0_{s,a}}} \left\{ P^0_{s,a}(V - \mu) - \sigma\left(\mathrm{sp}((V-\mu))_*\right)\right\}\right.$$

$$\left. - \max_{\mu^\lambda_{\widehat{P}^0_{s,a}} \in \mathcal{M}^\lambda_{\widehat{P}^0_{s,a}}} \left\{ \widehat{P}^0_{s,a}(V - \mu^\lambda_{\widehat{P}^0}) - \sigma\left(\mathrm{sp}((V-\mu^\lambda_{\widehat{P}^0_{s,a}})_*)\right)\right\} \right| \tag{162}$$

$$\leq \sum_a \pi(a|s) \left( 2\sqrt{\frac{L}{N}}\sqrt{\mathrm{Var}_{P^0_{s,a}}(V)} + \frac{L C_S \|1\|_*}{N(1-\gamma)}\right) \tag{163}$$

$$= 2\sqrt{\frac{L}{N}}\sqrt{\mathrm{Var}_{P^0_{s,a}}(V)} + \frac{3L C_S \|1\|_*}{N(1-\gamma)} \tag{164}$$

using $sa$-rectangular results, which gives the result for $s$-rectangular case.

Combining this lemma with a matrix notation using union bound, one has with probability $1 - \delta$:

$$\left| \widehat{\underline{P}}^{\pi^*,V} V^{\pi^*,\sigma} - \underline{P}^{\pi^*,V} V^{\pi^*,\sigma} \right| \leq 2\sqrt{\frac{L}{N}}\sqrt{\mathrm{Var}_{P^*}(V^{\star,\sigma})} + \frac{3L C_S \|1\|_*}{N(1-\gamma)} 1 \tag{165}$$

$$\tag{166}$$

$\square$

### D.3.4    Proof of Lemma 9

Using the same argument as in (216), it holds that for any $\alpha^*$ solution of (53)

$$\left(I - \gamma \widehat{\underline{P}}^{\pi^\star,V}\right)^{-1} \sqrt{\mathrm{Var}_{\widehat{\underline{P}}^{\pi^\star,V}}(V^{\star,\sigma})} = \sqrt{\frac{1}{1-\gamma}} \sqrt{\sum_{t=0}^{\infty} \gamma^t \left(\widehat{\underline{P}}^{\pi^\star,V}\right)^t \mathrm{Var}_{\widehat{\underline{P}}^{\pi^\star,V}}(V^{\star,\sigma})}. \quad (167)$$

Then we can control $\mathrm{Var}_{\widehat{\underline{P}}^{\pi^\star,V}}(V^{\star,\sigma})$. Defining $V' := V^{\star,\sigma} - \eta\mathbf{1}, \eta \in \mathbb{R}$, we use Bellman's equation in (32)) which lead to

$$V' = V^{\star,\sigma} - \eta\mathbf{1} \le V^{\star,\sigma} - \eta\mathbf{1} = r_{\pi^\star} + \gamma \underline{P}^{\pi^\star,V} V^{\star,\sigma} - \eta\mathbf{1} \quad (168)$$

$$= r_{\pi^\star} + \gamma P^{\pi^\star,V} V^{\star,\sigma} - \gamma\sigma\mathrm{sp}(V^{\star,\sigma})_* - \eta\mathbf{1} \quad (169)$$

$$= r'_{\pi^\star} + \gamma \widehat{\underline{P}}^{\pi^\star,V} V' + \gamma\left(P^{\pi^\star,V} - \widehat{\underline{P}}^{\pi^\star,V}\right) V^{\star,\sigma} - \gamma\sigma\mathrm{sp}(V^{\star,\sigma})_* \quad (170)$$

$$= r'_{\pi^\star} + \gamma \widehat{\underline{P}}^{\pi^\star,V} V' + \gamma\left(\underline{P}^{\pi^\star,V} - \widehat{\underline{P}}^{\pi^\star,V}\right) V^{\star,\sigma} \quad (171)$$

$$\le r'_{\pi^\star} + \gamma \widehat{\underline{P}}^{\pi^\star,V} V' + \gamma\left(\underline{P}^{\pi^\star,V} - \widehat{\underline{P}}^{\pi^\star,V}\right) V^{\star,\sigma} \quad (172)$$

where in the second line we use Lemma 3. and we define $r'_{\pi^\star} = r_{\pi^\star} - (1-\gamma)\eta < r_{\pi^\star} < 1$. We obtain the same result in $s$-rectangular case using lemma 4 instead. Then

$$\mathrm{Var}_{\widehat{\underline{P}}^{\pi^\star,V}}(V^{\star,\sigma}) \stackrel{(a)}{=} \mathrm{Var}_{\widehat{\underline{P}}^{\pi^\star,V}}(V') = \widehat{\underline{P}}^{\pi^\star,V}(V' \circ V') - \left(\widehat{\underline{P}}^{\pi^\star,V} V'\right) \circ \left(\widehat{\underline{P}}^{\pi^\star,V} V'\right)$$

$$= \widehat{\underline{P}}^{\pi^\star,V}(V' \circ V') - \left(\widehat{\underline{P}}^{\pi^\star,V} V'\right) \circ \left(\widehat{\underline{P}}^{\pi^\star,V} V'\right)$$

$$\stackrel{(b)}{\le} \widehat{\underline{P}}^{\pi^\star,V}(V' \circ V') - \frac{1}{\gamma^2}\left(V' - r'_{\pi^\star} - \gamma\left(\underline{P}^{\pi^\star,V} - \widehat{\underline{P}}^{\pi^\star,V}\right) V^{\star,\sigma}\right)^{\circ 2}$$

$$= \widehat{\underline{P}}^{\pi^\star,V}(V' \circ V') - \frac{1}{\gamma^2} V' \circ V' + \frac{2}{\gamma^2} V' \circ \left(r'_{\pi^\star} + \gamma\left(\underline{P}^{\pi^\star,V} - \widehat{\underline{P}}^{\pi^\star,V}\right) V^{\star,\sigma}\right)$$

$$\qquad - \frac{1}{\gamma^2}\left(r'_{\pi^\star} + \gamma\left(\underline{P}^{\pi^\star,V} - \widehat{\underline{P}}^{\pi^\star,V}\right) V^{\star,\sigma}\right)^{\circ 2}$$

$$\stackrel{(c)}{\le} \widehat{\underline{P}}^{\pi^\star,V}(V' \circ V') - \frac{1}{\gamma} V' \circ V' + \frac{2}{\gamma^2}\|V'\|_\infty\mathbf{1} \quad (173)$$

$$+ \frac{2}{\gamma}\|V'\|_\infty\left|\left(\underline{P}^{\pi^\star,V} - \widehat{\underline{P}}^{\pi^\star,V}\right) V^{\star,\sigma}\right| \quad (174)$$

$$\le \widehat{\underline{P}}^{\pi^\star,V}(V' \circ V') - \frac{1}{\gamma} V' \circ V' + \frac{2}{\gamma^2}\|V'\|_\infty\mathbf{1} \quad (175)$$

$$+ \frac{2}{\gamma}\|V'\|_\infty\left(2\sqrt{\frac{L}{(1-\gamma)^2 N}} + \frac{3C_S\|\mathbf{1}\|_* L}{N(1-\gamma)}\right)\mathbf{1}, \quad (176)$$

where (a) holds by the fact that $\mathrm{Var}_{P_\pi}(V - \eta\mathbf{1}) = \mathrm{Var}_{P_\pi}(V)$ for any scalar $\eta$, (b) follows from (172), moreover (c) comes from $\frac{1}{\gamma^2} V' \circ V' \ge \frac{1}{\gamma} V' \circ V'$ and $-1 \le r_{\pi^\star} - (1-\gamma)V_{\min}\mathbf{1} = r'_{\pi^\star} \le r_{\pi^\star} \le 1$.

Finally, the inequality is due to Lemma 8. Plugging (176) into (167) gives,

$$\left(I - \gamma \underline{\widehat{P}}^{\pi^\star, V}\right)^{-1} \sqrt{\mathrm{Var}_{\widehat{P}^{\pi^\star, V}}(V^{\star, \sigma})} \tag{177}$$

$$\leq \sqrt{\frac{1}{1-\gamma}} \Big( \sum_{t=0}^{\infty} \gamma^t \left(\underline{\widehat{P}}^{\pi^\star, V}\right)^t \Big( \widehat{\underline{P}}^{\pi^\star, V} (V' \circ V') - \frac{1}{\gamma} V' \circ V' + \frac{2}{\gamma^2} \|V'\|_\infty 1 \tag{178}$$

$$+ \frac{2}{\gamma} \|V'\|_\infty \Big( 2 \sqrt{\frac{L}{(1-\gamma)^2 N}} + \frac{3 C_S \|1\|_* L}{N(1-\gamma)} \Big) 1 \Big) \Big)^{1/2}$$

$$\overset{(i)}{\leq} \sqrt{\frac{1}{1-\gamma}} \sqrt{\left| \sum_{t=0}^{\infty} \gamma^t \left(\widehat{\underline{P}}^{\pi^\star, V}\right)^t \Big( \widehat{\underline{P}}^{\pi^\star, V} (V' \circ V') - \frac{1}{\gamma} V' \circ V' \Big) \right|}$$

$$+ \sqrt{\frac{1}{1-\gamma}} \sqrt{\sum_{t=0}^{\infty} \gamma^t \left(\widehat{\underline{P}}^{\pi^\star, V}\right)^t \Big( \frac{2}{\gamma^2} \|V'\|_\infty 1 + \frac{2}{\gamma} \|V'\|_\infty \Big( 2 \sqrt{\frac{L}{(1-\gamma)^2 N}} + \frac{3 C_S \|1\|_* L}{N(1-\gamma)} \Big) 1 \Big)}$$

$$\leq \sqrt{\frac{1}{1-\gamma}} \sqrt{\left| \sum_{t=0}^{\infty} \gamma^t \left(\widehat{\underline{P}}^{\pi^\star, V}\right)^t \Big[ \widehat{\underline{P}}^{\pi^\star, V} (V' \circ V') - \frac{1}{\gamma} V' \circ V' \Big] \right|} \tag{179}$$

$$+ \sqrt{\frac{\Big( 2 + 2 \Big( 2 \sqrt{\frac{L}{(1-\gamma)^2 N}} + \frac{3 C_S \|1\|_* L}{N(1-\gamma)} \Big) \Big) \|V'\|_\infty}{(1-\gamma)^2 \gamma^2}} 1, \tag{180}$$

using in (i) the triangle inequality. The final part of the proof focuses on the first term, which follows

$$\left| \sum_{t=0}^{\infty} \gamma^t \left(\widehat{\underline{P}}^{\pi^\star, V}\right)^t \Big( \widehat{\underline{P}}^{\pi^\star, V} (V' \circ V') - \frac{1}{\gamma} V' \circ V' \Big) \right|$$

$$= \left| \Big( \sum_{t=0}^{\infty} \gamma^t \left(\widehat{\underline{P}}^{\pi^\star, V}\right)^{t+1} - \sum_{t=0}^{\infty} \gamma^{t-1} \left(\widehat{\underline{P}}^{\pi^\star, V}\right)^t \Big) (V' \circ V') \right| \leq \frac{1}{\gamma} \|V'\|_\infty^2 1 \tag{181}$$

using recursion between the two sums. Then, using (181) back to (180) leads to

$$\left(I - \gamma \underline{\widehat{P}}^{\pi^\star, V}\right)^{-1} \sqrt{\mathrm{Var}_{\widehat{P}^{\pi^\star, V}}(V^{\star, \sigma})}$$

$$\leq \sqrt{\frac{\|V\|_\infty^2}{\gamma(1-\gamma)}} 1 + 3 \sqrt{\frac{\Big( 1 + \Big( \sqrt{\frac{L}{(1-\gamma)^2 N}} + \frac{C_S \|1\|_* L}{N(1-\gamma)} \Big) \Big) \|V'\|_\infty}{(1-\gamma)^2 \gamma^2}} 1$$

$$\leq 4 \sqrt{\frac{\Big( 1 + \Big( \sqrt{\frac{L}{(1-\gamma)^2 N}} + \frac{C_S \|1\|_* L}{N(1-\gamma)} \Big) \Big) \|V'\|_\infty}{(1-\gamma)^2 \gamma^2}} 1 \tag{182}$$

$$\leq 4 \sqrt{\frac{\Big( 1 + \Big( 1 \sqrt{\frac{L}{(1-\gamma)^2 N}} + \frac{C_S \|1\|_* L}{N(1-\gamma)} \Big) \Big) \|V'\|_*}{(1-\gamma)^2 \gamma^2}} 1 \tag{183}$$

Taking the infimum over $\eta$ in the right-hand side, recall $V' := V^{\star,\sigma} - \eta 1$, we obtain the definition of the span semi norm.

$$\left(I - \gamma \widehat{\underline{P}}^{\pi^\star,V}\right)^{-1}\sqrt{\mathrm{Var}_{\widehat{\underline{P}}^{\pi^\star,V}}(V^{\star,\sigma})} \le 4\sqrt{\frac{\left(1 + \left(\sqrt{\frac{L}{(1-\gamma)^2 N}} + \frac{C_S\|1\|_* L}{N(1-\gamma)}\right)\right)\mathrm{sp}(V^{\star,\sigma})_*}{(1-\gamma)^2\gamma^2}}1$$

$$\le 4\sqrt{\frac{\left(1 + \left(\sqrt{\frac{L}{(1-\gamma)^2 N}} + \frac{C_S\|1\|_* L}{N(1-\gamma)}\right)\right)}{\gamma^3(1-\gamma)^2\max\{1-\gamma, C_g\sigma\}}}1 \qquad (184)$$

$$\le 4\sqrt{\frac{\left(1 + \left(\sqrt{\frac{L}{(1-\gamma)^2 N}} + \frac{C_S\|1\|_* L}{N(1-\gamma)}\right)\right)}{\gamma^3(1-\gamma)^3}}1, \qquad (185)$$

where the penultimate inequality follows from applying Lemma 5 with $P = P^0$ and $\pi = \pi^\star$:

$$\mathrm{sp}(V^{\star,\sigma})_* \le \frac{1}{\gamma\max\{1-\gamma, C_g\sigma\}}$$

or with an extra factor for s rectangular assumptions.

$$\mathrm{sp}(V^{\star,\sigma})_* \le \frac{1}{\gamma\max\{1-\gamma, \min_s\|\pi_s\|_* \tilde{\sigma}Cg\}}.$$

### D.3.5 Proof of Lemma 10

In this proof, we will $sa$-rectangular notations, for any $(s,a) \in \mathcal{S} \times \mathcal{A}$, using the results in (144). In the $sa$-rectangular case:

$$\left|\widehat{P}_{s,a}^{\widehat{\pi},\widehat{V}} \widehat{V}^{\widehat{\pi},\sigma} - P_{s,a}^{\widehat{\pi},\widehat{V}} \widehat{V}^{\widehat{\pi},\sigma}\right| \le \max\left\{\left|\left(P_{s,a}^0 - \widehat{P}_{s,a}^0\right)[\widehat{V}^{\widehat{\pi},\sigma}]_{\alpha_{P_{s,a}}^{\lambda,\omega*}}\right|, \left|\left(P_{s,a}^0 - \widehat{P}_{s,a}^0\right)[\widehat{V}^{\widehat{\pi},\sigma}]_{\alpha_{\widehat{P}_{s,a}}^{\lambda,\omega*}}\right|\right\} \qquad (186)$$

The first term in this max can be bounded using:

$$\left|\left(P_{s,a}^0 - \widehat{P}_{s,a}^0\right)[\widehat{V}^{\widehat{\pi},\sigma}]_{\alpha_{P_{sa}}^{\lambda,\omega*}}\right| \qquad (187)$$

$$\overset{(a)}{\le} \left(\left|\left(P_{s,a}^0 - \widehat{P}_{s,a}^0\right)[\widehat{V}^{\star,\sigma}]_{\alpha_{P_{sa}}^{\lambda,\omega*}}\right| + \left|\left(P_{s,a}^0 - \widehat{P}_{s,a}^0\right)\left([\widehat{V}^{\widehat{\pi},\sigma}]_{\alpha_{P_{sa}}^{\lambda,\omega*}} - [\widehat{V}^{\star,\sigma}]_{\alpha_{P_{sa}}^{\lambda,\omega*}}\right)\right|\right)$$

$$\le \left(\left|\left(P_{s,a}^0 - \widehat{P}_{s,a}^0\right)[\widehat{V}^{\star,\sigma}]_{\alpha_{P_{sa}}^{\lambda,\omega*}}\right| + \left\|P_{s,a}^0 - \widehat{P}_{s,a}^0\right\|_1 \left\|[\widehat{V}^{\widehat{\pi},\sigma}]_{\alpha_{P_{sa}}^{\lambda,\omega*}} - [\widehat{V}^{\star,\sigma}]_{\alpha_{P_{sa}}^{\lambda,\omega*}}\right\|_\infty\right)$$

$$\overset{(b)}{\le} \left|\left(P_{s,a}^0 - \widehat{P}_{s,a}^0\right)[\widehat{V}^{\star,\sigma}]_{\alpha_{P_{sa}}^{\lambda,\omega*}}\right| + 2\left\|\widehat{V}^{\widehat{\pi},\sigma} - \widehat{V}^{\star,\sigma}\right\|_\infty$$

$$\overset{(c)}{\le} \left|\left(P_{s,a}^0 - \widehat{P}_{s,a}^0\right)[\widehat{V}^{\star,\sigma}]_{\alpha_{P_{sa}}^{\lambda,\omega*}}\right| + 2\varepsilon_{\mathsf{opt}} \qquad (188)$$

where (a) comes from the triangle inequality, and (b) comes from $\left\|P_{s,a}^0 - \widehat{P}_{s,a}^0\right\|_1 \le 2$ and $\left\|[\widehat{V}^{\widehat{\pi},\sigma}]_{\alpha_{P_{sa}}^{\lambda,\omega*}} - [\widehat{V}^{\star,\sigma}]_{\alpha_{P_{sa}}^{\lambda,\omega*}}\right\|_\infty \le \left\|\widehat{V}^{\widehat{\pi},\sigma} - \widehat{V}^{\star,\sigma}\right\|_\infty$, and (c) follows from the definition of the optimization error in (55). The second term of the max can be controlled in the same manner, i.e.:

$$\left|\left(P_{s,a}^0 - \widehat{P}_{s,a}^0\right)[\widehat{V}^{\widehat{\pi},\sigma}]_{\alpha_{\widehat{P}_{sa}}^{\lambda,\omega*}}\right| \le \left|\left(P_{s,a}^0 - \widehat{P}_{s,a}^0\right)[\widehat{V}^{\star,\sigma}]_{\alpha_{\widehat{P}_{sa}}^{\lambda,\omega*}}\right| + 2\varepsilon_{\mathsf{opt}} \qquad (189)$$

$$\le \left|\max_{\mu_{P_{s,a}^0}^\lambda \in \mathcal{M}_{P_{s,a}^0}^\lambda}\left(P_{s,a}^0 - \widehat{P}_{s,a}^0\right)(\widehat{V}^{\star,\sigma} - \mu_{P_{s,a}^0}^\lambda) + \max_{\mu_{\widehat{P}_{s,a}^0}^\lambda \in \mathcal{M}_{\widehat{P}_{s,a}^0}^\lambda}\left(P_{s,a}^0 - \widehat{P}_{s,a}^0\right)(\mu_{P_{s,a}^0}^\lambda - \mu_{\widehat{P}_{s,a}^0}^\lambda)\right|$$

$$\qquad (190)$$

$$+ 2\varepsilon_{\mathsf{opt}} \qquad (191)$$

where the last inequality follow the decomposition of (150). Finally, to control the remaining term

$$\max_{\mu_{P_{s,a}^0}^\lambda \in \mathcal{M}_{P_{s,a}^0}^\lambda} \left( P_{s,a}^0 - \widehat{P}_{s,a}^0 \right) (\widehat{V}^{\star,\sigma} - \mu_{P_{s,a}^0}^\lambda) = \max_{\alpha_P^\lambda \in \mathbb{A}_P^\lambda} \left\{ (P_{s,a}^0 - \widehat{P}_{s,a}^0) \, [V]_{\alpha_P^\lambda} \right\} \tag{192}$$

(191) for any given $\alpha \in [0, \alpha_{P_{sa}}^{\lambda,\omega*}[ \subset \left[0, \frac{1}{1-\gamma}\right]^S$ in the variational family with one parameter $\lambda$, with the dependency between $\widehat{V}^{\star,\sigma}$ and $\widehat{P}^0$, we resort to the following leave-one-out argument or absorbing MDPs used in [Agarwal et al., 2020, Li et al., 2022b, Shi and Chi, 2022, Clavier et al., 2023]. To begin, we create a collection of auxiliary RMDPs that exhibit the intended statistical independence between robust value functions and the estimated nominal transition kernel. These auxiliary RMDPs are designed to be minimally distinct from the initial RMDPs, subsequently, we manage to control the relevant term within these auxiliary RMDPs and demonstrate that its value closely approximates the target quantity for the desired RMDP. Recall that the empirical infinite-horizon robust MDP $\widehat{\mathcal{M}}_{\mathrm{rob}}$ is defined using the nominal transition kernel $\widehat{P}^0$. Inspired by Agarwal et al. [2020], we can construct an auxiliary absorbing robust MDP $\widehat{\mathcal{M}}_{\mathrm{rob}}^{s,u}$ for each state $s$ and any non-negative scalar $u \geq 0$, so that it is the same as $\widehat{\mathcal{M}}_{\mathrm{rob}}$ except for the transition properties in state $s$. These auxiliary MDPS are called absorbing MDPs are have been used for the first time in the context of RMDPS in Clavier et al. [2023]. Defining the reward function and nominal transition kernel of $\widehat{\mathcal{M}}_{\mathrm{rob}}^{s,u}$ as $P^{s,u}$ and $r^{s,u}$, which are expressed as follows using the same notation as Shi et al. [2023]:

$$\begin{cases} r^{s,u}(s,a) = u & \forall a \in \mathcal{A}, \\ r^{s,u}(\widetilde{s},a) = r(\widetilde{s},a) & \forall (\widetilde{s},a) \in \mathcal{S} \times \mathcal{A} \text{ and } \widetilde{s} \neq s. \end{cases} \tag{193}$$

$$\begin{cases} P^{s,u}(s' \,|\, s,a) = \mathbb{1}(s' = s) & \forall (s',a) \in \mathcal{S} \times \mathcal{A}, \\ P^{s,u}(\cdot \,|\, \widetilde{s},a) = \widehat{P}^0(\cdot \,|\, \widetilde{s},a) & \forall (\widetilde{s},a) \in \mathcal{S} \times \mathcal{A} \text{ and } \widetilde{s} \neq s, \end{cases} \tag{194}$$

Nominal transition probability at state $s$ of the auxiliary $\widehat{\mathcal{M}}_{\mathrm{rob}}^{s,u}$ never leaves state $s$ once entered, which gives the name absorbing to these auxiliary RMPDs. Finally, we define the robust Bellman operator $\widehat{\mathcal{T}}_{s,u}^\sigma(\cdot)$ associated $\widehat{\mathcal{M}}_{\mathrm{rob}}^{s,u}$ as

$$\widehat{\mathcal{T}}_{s,u}^\sigma(Q)(\widetilde{s},a) = r^{s,u}(\widetilde{s},a) + \gamma \inf_{\mathcal{P} \in \mathcal{U}^{\mathrm{sa},\sigma}(P_{\widetilde{s},a}^{s,u})} \mathcal{P}V, \qquad \text{with } V(\widetilde{s}) = \max_a Q(\widetilde{s},a). \tag{195}$$

in $sa$-rectangular case and with stochastic policy in $s$-rectangular case. Using these auxiliary RMDPs we can remark equivalence between $\widehat{\mathcal{M}}_{\mathrm{rob}}$ and the auxiliary RMDP $\widehat{\mathcal{M}}_{\mathrm{rob}}^{s,u}$ fixed-point. First, $\widehat{Q}^{\star,\sigma}$ is the unique-fixed point of $\widehat{\mathcal{T}}^\sigma(\cdot)$ with associated value $\widehat{V}^{\star,\sigma}$. We will show that the robust value function $\widehat{V}_{s,u^\star}^{\star,\sigma}$ obtained from the fixed point of $\widehat{\mathcal{T}}_{s,u}^\sigma(\cdot)$ is the same as the the robust value function $\widehat{V}^{\star,\sigma}$ derived from $\widehat{\mathcal{T}}^\sigma(\cdot)$, as long as we choose $u$ as

$$u^\star := u^\star(s) = \widehat{V}^{\star,\sigma}(s) - \gamma \inf_{\mathcal{P} \in \mathcal{U}^{\mathrm{sa},\sigma}(e_s)} \mathcal{P}\widehat{V}^{\star,\sigma}. \tag{196}$$

with $e_s$ is the $s$-th standard basis vector in $\mathbb{R}^S$. This assertion is verified as:

- **First for state** $s' \neq s$, **for all** $a \in \mathcal{A}$: it holds

$$r^{s,u^\star}(s',a) + \gamma \inf_{\mathcal{P} \in \mathcal{U}^{\mathrm{sa},\sigma}(P_{s',a}^{s,u^\star})} \mathcal{P}\widehat{V}^{\star,\sigma} = r(s',a) + \gamma \inf_{\mathcal{P} \in \mathcal{U}^{\mathrm{sa},\sigma}(\widehat{P}_{s',a}^0)} \mathcal{P}\widehat{V}^{\star,\sigma}$$

$$= \widehat{\mathcal{T}}^\sigma(\widehat{Q}^{\star,\sigma})(s',a) = \widehat{Q}^{\star,\sigma}(s',a), \tag{197}$$

  where the first equality holds because of (193) and (194), and the last inequality comes from that $\widehat{Q}^{\star,\sigma}$ is the fixed point of $\widehat{\mathcal{T}}^\sigma(\cdot)$ (see Lemma C.3) and the definition of the robust Bellman operator in (13).

- **Then for state** $s$, **for any** $a \in \mathcal{A}$:

$$r^{s,u^\star}(s,a) + \gamma \inf_{\mathcal{P} \in \mathcal{U}^\sigma(P_{s,a}^{s,u^\star})} \mathcal{P}\widehat{V}^{\star,\sigma} = u^\star + \gamma \inf_{\mathcal{P} \in \mathcal{U}^{\mathrm{sa},\sigma}(e_s)} \mathcal{P}\widehat{V}^{\star,\sigma}$$

$$= \widehat{V}^{\star,\sigma}(s) - \gamma \inf_{\mathcal{P} \in \mathcal{U}^{\mathrm{sa},\sigma}(e_s)} \mathcal{P}\widehat{V}^{\star,\sigma} + \gamma \inf_{\mathcal{P} \in \mathcal{U}^{\mathrm{sa},\sigma}(e_s)} \mathcal{P}\widehat{V}^{\star,\sigma} = \widehat{V}^{\star,\sigma}(s), \tag{198}$$

using in the first equality is the definition of $P_{s,a}^{s,u^\star}$ in (194) and where we use the definition of $u^\star$ in (196) in the second one.

Finally, we have proved that there exists a fixed point $\widehat{Q}_{s,u^\star}^{\star,\sigma}$ of the operator $\widehat{\mathcal{T}}_{s,u^\star}^{\sigma}(\cdot)$ by taking

$$\begin{cases} \widehat{Q}_{s,u^\star}^{\star,\sigma}(s,a) = \widehat{V}^{\star,\sigma}(s) & \forall a \in \mathcal{A}, \\ \widehat{Q}_{s,u^\star}^{\star,\sigma}(s',a) = \widehat{Q}^{\star,\sigma}(s',a) & \forall s' \neq s \text{ and } a \in \mathcal{A}. \end{cases} \tag{199}$$

we have confirmed the existence of a fixed point of the operator $\widehat{\mathcal{T}}_{s,u^\star}^{\sigma}(\cdot)$ with corresponding value function $\widehat{V}_{s,u^\star}^{\star,\sigma}$ that coincide with $\widehat{V}^{\star,\sigma}$. Note that the corresponding properties between $\widehat{\mathcal{M}}_{\text{rob}}$ and $\widehat{\mathcal{M}}_{\text{rob}}^{s,u}$ in Step 1 and Step 2 hold in fact for any uncertainty set and $s$- or $sa$-rectangular assumptions. Equipped with these fixed point equalities, we can use concentration inequalities to show this lemma.

**Concentration inequality using an $\varepsilon$-net for all reward values $u$.** First we can verify that

$$0 \leq u^\star \leq [\widehat{V}^{\star,\sigma}(s)]_{\alpha_{P_{s,a}}^{\lambda,\omega*}} \leq \widehat{V}^{\star,\sigma}(s) \leq \frac{1}{1-\gamma}. \tag{200}$$

Then, we define a $N_{\varepsilon_2}$-net over the interval $\left[0, 1/(1-\gamma)\right]$, where $|N_{\varepsilon_2}|$ the size of the net can be controlled by $|N_{\varepsilon_2}| \leq \frac{3}{\varepsilon_2(1-\gamma)}$ [Vershynin, 2018]. The only parameter that varies is $\lambda$ in the variation family, $\alpha_{P_{sa}}^{\lambda}$ so we have 1-dimensional control and not a vector in $\mathbb{R}^S$. Then similarly to Lemma C.3, it holds that for each $u \in N_{\varepsilon_2}$, there exists a unique fixed point $\widehat{Q}_{s,u}^{\star,\sigma}$ of the operator $\widehat{\mathcal{T}}_{s,u}^{\sigma}(\cdot)$, which satisfies $0 \leq \widehat{Q}_{s,u}^{\star,\sigma} \leq \frac{1}{1-\gamma} \cdot 1$. Consequently, the corresponding robust value function can be upper bounded by $\left\|\widehat{V}_{s,u}^{\star,\sigma}\right\|_{\infty} \leq \frac{1}{1-\gamma}$. Using (194) and (193) by construction for all $u \in N_{\varepsilon_2}$, $\widehat{\mathcal{M}}_{\text{rob}}^{s,u}$ is statistically independent of $\widehat{P}_{s,a}^0$. This independence indicates that $[\widehat{V}_{s,u}^{\star,\sigma}]_\alpha$ and $\widehat{P}_{s,a}^0$ are independent for a fixed $\alpha$. Using (148) and (149) and taking the union bound over all $(s,a,\alpha) \in \mathcal{S} \times \mathcal{A} \times N_{\varepsilon_1}$, $u \in N_{\varepsilon_2}$ gives that, with probability at least $1 - \delta$, it holds for all $(s,a,u) \in \mathcal{S} \times \mathcal{A} \times N_{\varepsilon_2}$ that

$$\max_{\alpha_{P_{sa}}^{\lambda,\omega} \in \mathrm{A}_{P_{sa}}^{\lambda,\omega}} \left| \left(P_{s,a}^0 - \widehat{P}_{s,a}^0\right) \left[\widehat{V}_{s,u}^{\star,\sigma}\right]_{\alpha_{P_{sa}}^{\lambda,\omega*}} \right| \leq 2\sqrt{\frac{2\log\left(\frac{18\|1\|_* SAN|N_{\varepsilon_2}|}{\delta}\right)}{N}}\sqrt{\mathrm{Var}_{P_{s,a}^0}(\widehat{V}_{s,u}^{\star,\sigma})} \tag{201}$$

$$+ \varepsilon_2$$

$$\leq 2\sqrt{\frac{2\log\left(\frac{18\|1\|_* SAN|N_{\varepsilon_2}|}{\delta}\right)}{(1-\gamma)^2 N}} + \varepsilon_2, \tag{202}$$

Finally, we use **uniform concentration** to obtain the lemma. Recalling that $u^\star \in \left[0, \frac{1}{1-\gamma}\right]$ (see (200)), we can always find some $\overline{u} \in N_{\varepsilon_2}$ such that $|\overline{u} - u^\star| \leq \varepsilon_2$. Consequently, plugging in the operator $\widehat{\mathcal{T}}_{s,u}^{\sigma}(\cdot)$ in (195) yields

$$\forall Q \in \mathbb{R}^{SA} : \quad \left\|\widehat{\mathcal{T}}_{s,\overline{u}}^{\sigma}(Q) - \widehat{\mathcal{T}}_{s,u^\star}^{\sigma}(Q)\right\|_{\infty} = |\overline{u} - u^\star| \leq \varepsilon_2$$

We can then remark that the fixed points of $\widehat{\mathcal{T}}_{s,\overline{u}}^{\sigma}(\cdot)$ and $\widehat{\mathcal{T}}_{s,u^\star}^{\sigma}(\cdot)$ obey

$$\left\|\widehat{Q}_{s,\overline{u}}^{\star,\sigma} - \widehat{Q}_{s,u^\star}^{\star,\sigma}\right\|_{\infty} = \left\|\widehat{\mathcal{T}}_{s,\overline{u}}^{\sigma}(\widehat{Q}_{s,\overline{u}}^{\star,\sigma}) - \widehat{\mathcal{T}}_{s,u^\star}^{\sigma}(\widehat{Q}_{s,u^\star}^{\star,\sigma})\right\|_{\infty}$$

$$\leq \left\|\widehat{\mathcal{T}}_{s,\overline{u}}^{\sigma}(\widehat{Q}_{s,\overline{u}}^{\star,\sigma}) - \widehat{\mathcal{T}}_{s,\overline{u}}^{\sigma}(\widehat{Q}_{s,u^\star}^{\star,\sigma})\right\|_{\infty} + \left\|\widehat{\mathcal{T}}_{s,\overline{u}}^{\sigma}(\widehat{Q}_{s,u^\star}^{\star,\sigma}) - \widehat{\mathcal{T}}_{s,u^\star}^{\sigma}(\widehat{Q}_{s,u^\star}^{\star,\sigma})\right\|_{\infty}$$

$$\leq \gamma \left\|\widehat{Q}_{s,\overline{u}}^{\star,\sigma} - \widehat{Q}_{s,u^\star}^{\star,\sigma}\right\|_{\infty} + \varepsilon_2,$$

where we use that the operator $\widehat{\mathcal{T}}_{s,u}^{\sigma}(\cdot)$ is a $\gamma$-contraction. It gives that:

$$\left\|\widehat{Q}_{s,\overline{u}}^{\star,\sigma} - \widehat{Q}_{s,u^\star}^{\star,\sigma}\right\|_{\infty} \leq \frac{\varepsilon_2}{(1-\gamma)} \quad \text{and} \quad \left\|\widehat{V}_{s,\overline{u}}^{\star,\sigma} - \widehat{V}_{s,u^\star}^{\star,\sigma}\right\|_{\infty} \leq \left\|\widehat{Q}_{s,\overline{u}}^{\star,\sigma} - \widehat{Q}_{s,u^\star}^{\star,\sigma}\right\|_{\infty} \leq \frac{\varepsilon_2}{(1-\gamma)}. \tag{203}$$

Finally to control the first term in (191), using the identity $\widehat{V}^{\star,\sigma} = \widehat{V}^{\star,\sigma}_{s,u^\star}$ or fixed point relation between the two RMPDS, established in previous step of the proof gives that: for all $(s,a) \in \mathcal{S} \times \mathcal{A}$,

$$\max_{\alpha^{\lambda,\omega}_{P_{s,a}} \in A^{\lambda,\omega}_{P_{s,a}}} \left| \left( P^0_{s,a} - \widehat{P}^0_{s,a} \right) [\widehat{V}^{\star,\sigma}]_{\alpha^{\lambda,\omega}_{P_{s,a}}} \right|$$

$$\leq \max_{\alpha^{\lambda,\omega}_{P_{s,a}} \in A^{\lambda,\omega}_{P_{s,a}}} \left| \left( P^0_{s,a} - \widehat{P}^0_{s,a} \right) [\widehat{V}^{\star,\sigma}]_{\alpha^{\lambda,\omega}_{P_{s,a}}} \right|$$

$$\overset{(a)}{\leq} \max_{\alpha^{\lambda,\omega}_{P_{s,a}} \in A^{\lambda,\omega}_{P_{s,a}}} \left\{ \left| \left( P^0_{s,a} - \widehat{P}^0_{s,a} \right) [\widehat{V}^{\star,\sigma}_{s,\overline{u}}]_{\alpha^{\lambda,\omega}_{P_{s,a}}} \right| + \left| \left( P^0_{s,a} - \widehat{P}^0_{s,a} \right) \left( [\widehat{V}^{\star,\sigma}_{s,\overline{u}}]_{\alpha^{\lambda,\omega}_{P_{s,a}}} - [\widehat{V}^{\star,\sigma}_{s,u^\star}]_{\alpha^{\lambda}_{P_{s,a}}} \right) \right| \right\}$$

$$\overset{(b)}{\leq} \max_{\alpha^{\lambda,\omega}_{P_{s,a}} \in A^{\lambda,\omega}_{P_{s,a}}} \left| \left( P^0_{s,a} - \widehat{P}^0_{s,a} \right) [\widehat{V}^{\star,\sigma}_{s,\overline{u}}]_{\alpha^{\lambda}_{P_{s,a}}} \right| + \frac{2\varepsilon_2}{(1-\gamma)}$$

$$\overset{(c)}{\leq} \frac{2\varepsilon_2}{(1-\gamma)} + \varepsilon_2 + 2\sqrt{\frac{2\log(\frac{18\|1\|_* SAN |N_{\varepsilon_2}|}{\delta})}{N}} \sqrt{\mathrm{Var}_{P^0_{s,a}}(\widehat{V}^{\star,\sigma}_{s,u})} + \frac{4\log(\frac{18\|1\|_* SAN |N_{\varepsilon_2}|}{\delta})}{3N(1-\gamma)}$$

$$\leq \frac{3\varepsilon_2}{(1-\gamma)} + 2\sqrt{\frac{2\log(\frac{18\|1\|_* SAN |N_{\varepsilon_2}|}{\delta})}{N}} \sqrt{\mathrm{Var}_{P^0_{s,a}}(\widehat{V}^{\star,\sigma})} + \frac{4\log(\frac{18\|1\|_* SAN |N_{\varepsilon_2}|}{\delta})}{3N(1-\gamma)}$$

$$+ 2\sqrt{\frac{2\log(\frac{18\|1\|_* SAN |N_{\varepsilon_2}|}{\delta})}{N}} \sqrt{\left| \mathrm{Var}_{P^0_{s,a}}(\widehat{V}^{\star,\sigma}) - \mathrm{Var}_{P^0_{s,a}}(\widehat{V}^{\star,\sigma}_{s,\overline{u}}) \right|}$$

$$\overset{(d)}{\leq} \frac{3\varepsilon_2}{(1-\gamma)} + 2\sqrt{2\frac{\log(\frac{18\|1\|_* SAN |N_{\varepsilon_2}|}{\delta})}{N}} \sqrt{\mathrm{Var}_{P^0_{s,a}}(\widehat{V}^{\star,\sigma})} + 2\sqrt{\frac{4\varepsilon_2 \log(\frac{18\|1\|_* SAN |N_{\varepsilon_2}|}{\delta})}{N(1-\gamma)^2}}$$

$$\tag{204}$$

$$\leq 2\sqrt{\frac{L''}{N}} \sqrt{\mathrm{Var}_{P^0_{s,a}}(\widehat{V}^{\star,\sigma})} + \frac{14\log(\frac{54\|1\|_* SAN |N_{\varepsilon_2}|}{\delta})}{N(1-\gamma)} \tag{205}$$

$$\leq 16\sqrt{\frac{L''}{(1-\gamma)^2 N}}, \tag{206}$$

with $L'' = \log\left(\frac{54\|1\|_* SAN^2}{(1-\gamma)\delta}\right)$ where (a) comes from triangular inequality, (b) is due (203), for any $\alpha \in \mathbb{R}^S$

$$\left| \left( P^0_{s,a} - \widehat{P}^0_{s,a} \right) \left( [\widehat{V}^{\star,\sigma}_{s,\overline{u}}]_\alpha - [\widehat{V}^{\star,\sigma}_{s,u^\star}]_\alpha \right) \right| \leq \left\| P^0_{s,a} - \widehat{P}^0_{s,a} \right\|_1 \left\| [\widehat{V}^{\star,\sigma}_{s,\overline{u}}]_\alpha - [\widehat{V}^{\star,\sigma}_{s,u^\star}]_\alpha \right\|_\infty$$

$$\leq 2 \left\| \widehat{V}^{\star,\sigma}_{s,\overline{u}} - \widehat{V}^{\star,\sigma}_{s,u^\star} \right\|_\infty \leq \frac{2\varepsilon_2}{(1-\gamma)}, \tag{207}$$

(c) follows from (201), (d) holds using Lemma 1 with (203). Here, the two last inequalities hold by letting $\varepsilon_2 = \frac{2\log(\frac{18\|1\|_* SAN |N_{\varepsilon_2}|}{\delta})}{N}$, which gives $|N_{\varepsilon_2}| \leq \frac{3}{\varepsilon_2(1-\gamma)} \leq \frac{3N}{1-\gamma}$, and the last inequality holds by the fact $\mathrm{Var}_{P^0_{s,a}}(\widehat{V}^{\star,\sigma}) \leq \|\widehat{V}^{\star,\sigma}\|_\infty \leq \frac{1}{1-\gamma}$ and letting $N \geq 2\log\left(\frac{54\|1\|_* SAN^2}{(1-\gamma)\delta}\right) = L''$.

Rewriting (186), the first term of the max is controlled.

$$\max\left\{ \left| \left( P^0_{s,a} - \widehat{P}^0_{s,a} \right) [\widehat{V}^{\widehat{\pi},\sigma}]_{\alpha^{\lambda*}_{P_{s,a}}} \right|, \left| \left( P^0_{s,a} - \widehat{P}^0_{s,a} \right) [\widehat{V}^{\widehat{\pi},\sigma}]_{\alpha^{\lambda*}_{\widehat{P}_{s,a}}} \right| \right\}$$

The second term can be controlled by the same term as the first one plus an additional term with

$$\left| \left( P^0_{s,a} - \widehat{P}^0_{s,a} \right) [\widehat{V}^{\widehat{\pi},\sigma}]_{\alpha^{\lambda*}_{\widehat{P}_{s,a}}} \right| \leq$$

$$\left| \max_{\mu^\lambda_{P^0_{s,a}} \in \mathcal{M}^\lambda_{P^0_{s,a}}} \left( P^0_{s,a} - \widehat{P}^0_{s,a} \right) (\widehat{V}^{\star,\sigma} - \mu^\lambda_{P^0_{s,a}}) + \max_{\mu^\lambda_{\widehat{P}^0_{s,a}} \in \mathcal{M}^\lambda_{\widehat{P}^0_{s,a}}} \left( P^0_{s,a} - \widehat{P}^0_{s,a} \right) (\mu^\lambda_{P^0_{s,a}} - \mu^\lambda_{\widehat{P}^0_{s,a}}) \right|$$

and similarly to previous lemma in (159), the residual or term in the right in the previous equation can be controlled with $\frac{L'C_S\|1\|_*}{N(1-\gamma)}$ Finally, putting (205) and (206) back into Equation (191) and using Eq. (206) with probability at least $1-\delta$ we obtain

$$
\begin{aligned}
\left|\widehat{P}_{s,a}^{\widehat{\pi},\widehat{V}}\widehat{V}^{\widehat{\pi},\sigma} - P_{s,a}^{\widehat{\pi},\widehat{V}}\widehat{V}^{\widehat{\pi},\sigma}\right| &\leq \max_{\alpha_{P_{s,a}}^{\lambda,\omega} \in A_{P_{s,a}}^{\lambda,\omega}} \left|\left(P_{s,a}^0 - \widehat{P}_{s,a}^0\right)[\widehat{V}^{\star,\sigma}]_{\alpha_{P_{s,a}}^{\lambda,\omega}}\right| + 2\varepsilon_{\mathsf{opt}} \\
&\leq 2\sqrt{\frac{L'}{N}}\sqrt{\mathrm{Var}_{P_{s,a}^0}(\widehat{V}^{\star,\sigma})} + 2\varepsilon_{\mathsf{opt}} + \frac{14L''C_S\|1\|_*}{N(1-\gamma)} \\
&\leq 2\sqrt{\frac{L''}{(1-\gamma)^2 N}} + 2\varepsilon_{\mathsf{opt}} + \frac{14L''C_S\|1\|_*}{N(1-\gamma)},
\end{aligned}
\tag{208}
$$

$\forall(s,a) \in \mathcal{S} \times \mathcal{A}$. Using matrix form we obtain finally:

$$
\begin{aligned}
\left|\widehat{\underline{P}}^{\widehat{\pi},\widehat{V}}\widehat{V}^{\widehat{\pi},\sigma} - \underline{P}^{\widehat{\pi},\widehat{V}}\widehat{V}^{\widehat{\pi},\sigma}\right| &\leq 2\sqrt{\frac{L''}{N}}\sqrt{\mathrm{Var}_{P_{s,a}^0}(\widehat{V}^{\star,\sigma})}1 + 2\varepsilon_{\mathsf{opt}}1 \\
&\leq 2\sqrt{\frac{L''}{(1-\gamma)^2 N}}1 + 2\varepsilon_{\mathsf{opt}}1. + \frac{14L''C_S\|1\|_*}{N(1-\gamma)}1
\end{aligned}
\tag{209}
$$

The proof is similar in the $s$-rectangular case, factorising by $\pi(a|s)$, like in in 8. Moreover, the proof is similar for $TV$ without the geometric term depending on $C_S$.

### D.3.6 Proof of Lemma 11

We always use the same manner as in Appendix D.3.4. Similarly to (167), it holds:

$$
\left(I - \gamma\underline{P}^{\widehat{\pi},\widehat{V}}\right)^{-1}\sqrt{\mathrm{Var}_{\underline{P}^{\widehat{\pi},\widehat{V}}}(\widehat{V}^{\widehat{\pi},\sigma})} \leq \sqrt{\frac{1}{1-\gamma}}\sqrt{\sum_{t=0}^{\infty}\gamma^t\left(\underline{P}^{\widehat{\pi},\widehat{V}}\right)^t\mathrm{Var}_{\underline{P}^{\widehat{\pi},\widehat{V}}}(\widehat{V}^{\widehat{\pi},\sigma})}.
\tag{210}
$$

In order to upper bound $\mathrm{Var}_{\underline{P}^{\widehat{\pi},\widehat{V}}}(\widehat{V}^{\widehat{\pi},\sigma})$, we define $V' := \widehat{V}^{\widehat{\pi},\sigma} - \eta 1$ with $\eta \in \mathbb{R}$. Using as (174), it holds

$$
\mathrm{Var}_{\underline{P}^{\widehat{\pi},\widehat{V}}}(\widehat{V}^{\widehat{\pi},\sigma}) \leq \underline{P}^{\widehat{\pi},\widehat{V}}(V' \circ V') - \frac{1}{\gamma}V' \circ V' + \frac{2}{\gamma^2}\|V'\|_\infty 1 + \frac{2}{\gamma}\|V'\|_\infty\left|\left(\widehat{\underline{P}}^{\widehat{\pi},\widehat{V}} - \underline{P}^{\widehat{\pi},\widehat{V}}\right)\widehat{V}^{\widehat{\pi},\sigma}\right|
$$

$$
\leq \underline{P}^{\widehat{\pi},\widehat{V}}(V' \circ V') - \frac{1}{\gamma}V' \circ V' +
\tag{211}
$$

$$
\frac{2}{\gamma^2}\|V'\|_\infty 1 + \frac{2}{\gamma}\|V'\|_\infty\left(2\sqrt{\frac{L''}{(1-\gamma)^2 N}} + 2\varepsilon_{\mathsf{opt}} + \frac{14L''C_S\|1\|_*}{N(1-\gamma)}\right)1,
\tag{212}
$$

where the last inequality makes use of Lemma 10. Plugging (212) back into (210) leads to

$$\left(I - \gamma \underline{P}^{\widehat{\pi}, \widehat{V}}\right)^{-1} \sqrt{\mathrm{Var}_{\underline{P}^{\widehat{\pi}, \widehat{v}}}(\widehat{V}^{\widehat{\pi}, \sigma})} \overset{(a)}{\leq} \sqrt{\frac{1}{1 - \gamma}} \sqrt{\left| \left| \sum_{t=0}^{\infty} \gamma^t \left(\underline{P}^{\widehat{\pi}, \widehat{V}}\right)^t \left(\underline{P}^{\widehat{\pi}, \widehat{V}} \left(V' \circ V'\right) - \frac{1}{\gamma} V' \circ V'\right) \right| \right|}$$

$$+ \sqrt{\frac{1}{(1 - \gamma)^2 \gamma^2} \left(2 \sqrt{\frac{L''}{(1 - \gamma)^2 N}} + 2\varepsilon_{\mathsf{opt}} + \frac{14 L'' C_S \|1\|_*}{N(1 - \gamma)}\right) \|V'\|_\infty 1}$$

$$\overset{(b)}{\leq} \sqrt{\frac{\|V'\|_\infty^2}{\gamma(1 - \gamma)} 1} + \sqrt{\frac{\left(2 \sqrt{\frac{L''}{(1 - \gamma)^2 N}} + 2\varepsilon_{\mathsf{opt}} + \frac{14 L'' C_S \|1\|_*}{N(1 - \gamma)}\right) \|V'\|_\infty}{(1 - \gamma)^2 \gamma^2} 1}$$

$$\overset{(c)}{\leq} \sqrt{\frac{\|V'\|_\infty^2}{\gamma(1 - \gamma)} 1} + 5 \sqrt{\left(1 + \varepsilon_{\mathsf{opt}} + \frac{L'' C_S \|1\|_*}{N(1 - \gamma)}\right) \frac{\|V'\|_\infty}{(1 - \gamma)^2 \gamma^2} 1} \tag{213}$$

$$\leq 6 \sqrt{\left(1 + \varepsilon_{\mathsf{opt}} + \frac{L'' C_S \|1\|_*}{N(1 - \gamma)}\right) \frac{\|V'\|_\infty}{(1 - \gamma)^2 \gamma^2} 1}, \tag{214}$$

where (a) is the same as (180), (b) holds by repeating the argument of (181), (c) follows by taking $N \geq \frac{L''}{(1 - \gamma)^2}$ and then the last inequality holds by $\|V'\|_\infty \leq \|V^{\star, \sigma}\|_\infty \leq \frac{1}{1 - \gamma}$. Then taking the infimum over $\eta$ in the right-hand side of the equation in the definition of $V'$ and using $\mathrm{sp}(.)_\infty \leq \|.\|_*$ gives

$$\left(I - \gamma \underline{P}^{\widehat{\pi}, \widehat{V}}\right)^{-1} \sqrt{\mathrm{Var}_{\underline{P}^{\widehat{\pi}, \widehat{v}}}(\widehat{V}^{\widehat{\pi}, \sigma})} \leq 6 \sqrt{\left(1 + \varepsilon_{\mathsf{opt}} + \frac{L'' C_S \|1\|_*}{N(1 - \gamma)}\right) \frac{\mathrm{sp}(V)_\infty}{(1 - \gamma)^2 \gamma^2} 1}$$

Finally, applying Lemma 5 with $P = \widehat{P}^0$ and $\pi = \widehat{\pi}$ yields

$$\mathrm{sp}(\widehat{V}^{\widehat{\pi}, \sigma})_* \leq \frac{1}{\gamma \max\{1 - \gamma, \gamma C_g \sigma\}}, \tag{215}$$

for $sa$-rectangular or

$$\mathrm{sp}(\widehat{V}^{\widehat{\pi}, \sigma})_* \leq \frac{1}{\gamma \max\{1 - \gamma, \min_s \|\widehat{\pi}\|_* \tilde{\sigma}\}}$$

in the $s$-rectangular case, which can be inserted into (214) and gives in $sa$-rectangular case:

$$\left(I - \gamma \underline{P}^{\widehat{\pi}, \widehat{V}}\right)^{-1} \sqrt{\mathrm{Var}_{\underline{P}^{\widehat{\pi}, \widehat{v}}}(\widehat{V}^{\widehat{\pi}, \sigma})} \leq 6 \sqrt{\frac{\left(1 + \varepsilon_{\mathsf{opt}} + \frac{L'' C_S \|1\|_*}{N(1 - \gamma)}\right)}{\gamma^3 (1 - \gamma)^2 \max\{1 - \gamma, \sigma\}} 1}$$

$$\leq 6 \sqrt{\frac{\left(1 + \varepsilon_{\mathsf{opt}} + \frac{L'' C_S \|1\|_*}{N(1 - \gamma)}\right)}{(1 - \gamma)^3 \gamma^3} 1}$$

where first inequalities comes from that we can bound it Eq. left-hand side of equation (214) by $\|V'\|_\infty \leq \|V^{\star, \sigma}\|_\infty \leq \frac{1}{1 - \gamma}$. Proof for $s$-rectangular is similar, but requires adding an extra factor depending on the norm of the current policy and we have:

$$\left(I - \gamma \underline{P}^{\widehat{\pi}, \widehat{V}}\right)^{-1} \sqrt{\mathrm{Var}_{\underline{P}^{\widehat{\pi}, \widehat{v}}}(\widehat{V}^{\widehat{\pi}, \sigma})} \leq 6 \sqrt{\frac{\left(1 + \varepsilon_{\mathsf{opt}} + \frac{L'' C_S \|1\|_*}{N(1 - \gamma)}\right)}{\gamma^3 (1 - \gamma)^2 \max\{1 - \gamma, C_g \tilde{\sigma} \min_s \|\widehat{\pi}_s\|_\infty\}} 1}$$

$$\leq 6 \sqrt{\frac{\left(1 + \varepsilon_{\mathsf{opt}} + \frac{L'' C_S \|1\|_*}{N(1 - \gamma)}\right)}{(1 - \gamma)^3 \gamma^2} 1}.$$

### D.3.7 Proof of Lemma 7

First, if each row of $P_\pi$ belongs to the simplex $\Delta(S)$, it lead that the row of $(1-\gamma)\,(I-\gamma P_\pi)^{-1}$ falls into $\Delta(S)$. Then,

$$
(I-\gamma P_\pi)^{-1}\sqrt{\mathrm{Var}_{P_\pi}(V^{\pi,P})} = \frac{1}{1-\gamma}(1-\gamma)\,(I-\gamma P_\pi)^{-1}\sqrt{\mathrm{Var}_{P_\pi}(V^{\pi,P})}
$$

$$
\overset{(a)}{\le} \frac{1}{1-\gamma}\sqrt{(1-\gamma)\,(I-\gamma P_\pi)^{-1}\mathrm{Var}_{P_\pi}(V^{\pi,P})}
$$

$$
= \sqrt{\frac{1}{1-\gamma}}\sqrt{\sum_{t=0}^{\infty}\gamma^t\,(P_\pi)^t\,\mathrm{Var}_{P_\pi}(V^{\pi,P})}, \tag{216}
$$

where (a) is due to Jensen's inequality. Then for any $\eta \in \mathbb{R}^+$, $V' := V^{\pi,P}-\eta 1$, we can upper bound $\mathrm{Var}_{P_\pi}(V^{\pi,P})$ :

$$
\mathrm{Var}_{P_\pi}(V^{\pi,P}) \overset{(i)}{=} \mathrm{Var}_{P_\pi}(V') = P_\pi\,(V'\circ V') - (P_\pi V')\circ(P_\pi V')
$$

$$
\overset{(ii)}{\le} P_\pi\,(V'\circ V') - \frac{1}{\gamma^2}(V'-r_\pi+(1-\gamma)\eta 1)\circ(V'-r_\pi+(1-\gamma)\eta 1)
$$

$$
= P_\pi\,(V'\circ V') - \frac{1}{\gamma^2}V'\circ V' + \frac{2}{\gamma^2}V'\circ(r_\pi-(1-\gamma)\eta 1) \tag{217}
$$

$$
- \frac{1}{\gamma^2}(r_\pi-(1-\gamma)\eta 1)\circ(r_\pi-(1-\gamma)\eta 1)
$$

$$
\le P_\pi\,(V'\circ V') - \frac{1}{\gamma}V'\circ V' + \frac{2}{\gamma^2}\|V'\|_\infty 1 \le P_\pi\,(V'\circ V') - \frac{1}{\gamma}V'\circ V' + \frac{2}{\gamma^2}\|V'\|_\infty 1, \tag{218}
$$

where (i) holds by the fact that $\mathrm{Var}_{P_\pi}(V^{\pi,P}-b1) = \mathrm{Var}_{P_\pi}(V^{\pi,P})$ for any scalar $b$ and $V^{\pi,P}\in\mathbb{R}^S$, (ii) follows from $V' \le r_\pi+\gamma P_\pi V^{\pi,P}-\eta 1 = r_\pi-(1-\gamma)\eta 1+\gamma P_\pi V'$, and the last line arises from $\frac{1}{\gamma^2}V'\circ V' \ge \frac{1}{\gamma}V'\circ V'$ and $\|r_\pi-(1-\gamma)\eta 1\|_\infty \le 1$. for $\eta \in [0, 1/(1-\gamma)[$. Plugging (218) back to (216) leads to

$$
(I-\gamma P_\pi)^{-1}\sqrt{\mathrm{Var}_{P_\pi}(V^{\pi,P})} \le \sqrt{\frac{1}{1-\gamma}}\sqrt{\sum_{t=0}^{\infty}\gamma^t\,(P_\pi)^t\left(P_\pi\,(V'\circ V') - \frac{1}{\gamma}V'\circ V' + \frac{2}{\gamma^2}\|V'\|_\infty 1\right)}
$$

$$
\overset{(i)}{\le} \sqrt{\frac{1}{1-\gamma}}\sqrt{\left|\sum_{t=0}^{\infty}\gamma^t\,(P_\pi)^t\left(P_\pi\,(V'\circ V') - \frac{1}{\gamma}V'\circ V'\right)\right|} + \sqrt{\frac{1}{1-\gamma}}\sqrt{\sum_{t=0}^{\infty}\gamma^t\,(P_\pi)^t\frac{2}{\gamma^2}\|V'\|_\infty 1}
$$

$$
\le \sqrt{\frac{1}{1-\gamma}}\sqrt{\left|\left(\sum_{t=0}^{\infty}\gamma^t\,(P_\pi)^{t+1} - \sum_{t=0}^{\infty}\gamma^{t-1}\,(P_\pi)^t\right)(V'\circ V')\right|} + \sqrt{\frac{2\|V'\|_\infty 1}{\gamma^2(1-\gamma)^2}}
$$

$$
\overset{(ii)}{\le} \sqrt{\frac{\|V'\|_\infty^2 1}{\gamma(1-\gamma)}} + \sqrt{\frac{2\|V'\|_\infty 1}{\gamma^2(1-\gamma)^2}}
$$

$$
\le \sqrt{\frac{8\|V'\|_\infty 1}{\gamma^2(1-\gamma)^2}}, \tag{219}
$$

$$
\tag{220}
$$

where (i) holds due to, (ii) holds by following recursion between the two sums, and the last inequality holds because $\|V'\|_\infty \le \frac{1}{1-\gamma}$. Then taking the minimum over $\eta$ in the right-hand side of the equation gives the result.

$$
(I-\gamma P_\pi)^{-1}\sqrt{\mathrm{Var}_{P_\pi}(V^{\pi,P})} \le \sqrt{\frac{8\mathrm{sp}(V^{\pi,P})_\infty}{\gamma^2(1-\gamma)^2}}
$$

However, we also $\|V'\|_\infty \le \|V^{\pi,P}\|_\infty \le \frac{1}{1-\gamma}$ in (219). So finally, the result is

$$(I - \gamma P_\pi)^{-1} \sqrt{\mathrm{Var}_{P_\pi}(V^{\pi,P})} \le \sqrt{\frac{8}{\gamma^2(1-\gamma)^2} \mathrm{sp}(V^{\pi,P})_\infty} 1.$$

# E  Proof of Theorem 2

In this section, we focus on the scenarios in the uncertainty sets are constructed with $(s, a)$-rectangularity condition with some general norms. Towards this, we firstly observe that for the two limiting cases $\ell_1$ norm and $\ell_\infty$ norm, one has $\|p_1 - p_2\|_1 \le 2$ and $\|p_1 - p_2\|_\infty \le 1$ for any two probability distribution $p_1, p_2 \in \mathbb{R}^S$. Namely, the accessible ranges of the uncertainty level $\sigma$ for $\ell_1$ norm and $\ell_\infty$ norm are $(0, 2]$ and $(0, 1]$, respectively. In addition, we have

$$\forall p_1, p_2 \in \mathbb{R}^S: \quad \|p_1 - p_2\|_\infty \le \|p_1 - p_2\| \le \|p_1 - p_2\|_1 \tag{221}$$

for any norm $\|\cdot\|$. It indicates that the accessible range of the uncertainty level $\sigma_{\|\cdot\|}$ for any given norm $\|\cdot\|$ is between $(0, \sigma_{\|\cdot\|}^{\max}]$, where $1 \le \sigma_{\|\cdot\|}^{\max} \le 2$.

To continue, we specify the definition of the uncertainty set with $sa$-rectangularity condition with some given general norm $\|\cdot\|$ as below: for any nominal transition kernel $P \in \mathbb{R}^{SA \times S}$,

$$\mathcal{U}_{\|\cdot\|}^\sigma(P) \coloneqq \mathcal{U}_{\|\cdot\|}^\sigma(P) = \otimes\, \mathcal{U}_p^\sigma(P_{s,a}), \qquad \mathcal{U}_{\|\cdot\|}^\sigma(P_{s,a}) \coloneqq \left\{ P_{s,a}' \in \Delta(\mathcal{S}) : \left\| P_{s,a}' - P_{s,a} \right\| \le \sigma_{\|\cdot\|} \right\}. \tag{222}$$

Then, we recall the assumption of the uncertainty radius $\sigma_{\|\cdot\|} \in \left(0, \sigma_{\|\cdot\|}^{\max}(1 - c_0)\right]$ with $0 < c_0 < 1$.

Then, resorting to the same class of hard MDPs in [Shi et al., 2023, Section C.1], we can complete the proof by directly following the same proof pipeline of Shi et al. [2023, Section C] by replacing $\sigma$ with $\sigma_{\|\cdot\|}^{\max} \sigma_{\|\cdot\|}$.

# F  Proof of Theorem 4

Developing the lower bound for the cases with $s$-rectangular uncertainty set involves several new challenges compared to that of $(s, a)$-rectangular cases. Specifically, the first challenge is that the optimal policy can be stochastic and hard to be characterized with a closed form for the RMDPs with a $s$-rectangular uncertainty set, rather than deterministic polices in $(s, a)$-rectangular cases. Such richer and smoother class of optimal policies makes slightly changing the transition kernel generally could only leads to a smoothly changed stochastic optimal policy instead of a completely different one. Such reduced changing of optimal policy further gives smaller performance gap, thus challenges of a tighter lower bound. Second, most of the hard instances in the literature are constructed as $SA$ states with a constant number of action spaces without loss of generality. While when it comes to $s$-rectangular uncertainty set, the action space size becomes important and can't be assumed as a constant anymore. So a new class of instances are required.

To address these challenges, in this section, we construct a new set of hard RMDP instances for two limiting cases: $\ell_1$ norm and $\ell_\infty$ norm.

### F.1  Construction of the hard problem instances

Before proceeding, we introduce two useful sets related to the state space and action space as below:

$$\mathcal{S} = \{0, 1, \ldots, S\}, \qquad \text{and} \qquad \mathcal{A} = \{0, 1, \cdots, A - 1\}.$$

In this section, we construct a set of RMDPs termed as $\mathcal{M}_{\ell_\infty}$, which consists of $S(A-1)$ components including $S(A-1)$ components, each associates with some different state-action pair. Specifically, it is defined as

$$\mathcal{M}_{\ell_\infty} \coloneqq \left\{ \mathcal{M}_\theta = \left(\mathcal{S}, \mathcal{A}, \mathcal{U}^\sigma(P^\theta), r, \gamma\right) \mid \theta \in \Theta = \left\{(i, j) : (i, j) \in \mathcal{S} \times \mathcal{A} \setminus \{0\}\right\}\right\}. \tag{223}$$

We introduce the detailed definition of $\mathcal{M}_{\ell_\infty}$ by introducing several key components of it sequentially. In particular, for any RMDP $\mathcal{M}_\theta \in \mathcal{M}_{\ell_\infty}$, the state space is of size $2S$, which includes two classes

of states $\mathcal{X} = \{x_0, x_1, \cdots, x_{S-1}\}$ and $\mathcal{Y} = \{y_0, y_1, \cdots, y_{S-1}\}$. The action space for each state is $\mathcal{A}$ of $A$ possible actions. So we have totally $2S$ states and there is in total $2SA$ state-action pairs.

Armed with the above definitions, we can first introduce the following nominal transition kernel: for all $(s, a) \in \mathcal{X} \cup \mathcal{Y} \times \mathcal{A}$

$$P^{(0,0)}(s' \mid s, a) = \begin{cases} p\mathbb{1}(s' = y_i) + (1-p)\mathbb{1}(s' = x_i) & \text{if} \quad s = x_i, a = 0, \quad \forall i \in \mathcal{S} \\ q\mathbb{1}(s' = y_i) + (1-q)\mathbb{1}(s' = x_i) & \text{if} \quad s = x_i, a \neq 0, \quad \forall i \in \mathcal{S} \\ \mathbb{1}(s' = s) & \text{if} \quad s \in \mathcal{Y} \end{cases} \tag{224}$$

Here, $p$ and $q$ are set according to

$$0 \leq p \leq 1 \quad \text{and} \quad 0 \leq q = p - \Delta \tag{225}$$

for some $p$ and $\Delta > 0$ that will be introduced momentarily.

Then we introduce the $S(A-1)$ components inside $\mathcal{M}_\infty$. Namely, for any $(i, j) \in \mathcal{S} \times \mathcal{A} \setminus \{0\}$, the nominal transition kernel of $\mathcal{M}_{(i,j)}$ is specified as

$$P^{(i,j)}(s' \mid s, a) = \begin{cases} p\mathbb{1}(s' = y_i) + (1-p)\mathbb{1}(s' = x_i) & \text{if } s = x_i, a = j \\ q\mathbb{1}(s' = y_i) + (1-q)\mathbb{1}(s' = x_i) & \text{if } s = x_i \in \mathcal{X}, a = 0 \\ P^{(0,0)}(s' \mid s, a) & \text{otherwise} \end{cases} \tag{226}$$

In words, the nominal transition kernel of each variant $\mathcal{M}_{(i,j)}$ only differs slightly from that of the basic nominal transition kernel $P^{(0,0)}$ when $s = x_i$ and $a = \{0, j\}$, which makes all the components inside $\mathcal{M}_{\ell_\infty}$ closed to each other.

In addition, the reward function is defined as

$$\forall a \in \mathcal{A}: \quad r(s, a) = \begin{cases} 1 & \text{if } s \in \mathcal{Y} \\ 0 & \text{otherwise.} \end{cases} \tag{227}$$

**Uncertainty set of the transition kernels.** Recall the following useful notation for any transition probability $P$, i.e., the transition vector associated with some state $s$ is denoted as:

$$P_s := P(\cdot, \cdot \mid s) \in \mathbb{R}^{1 \times SA}, \quad P_s^0 := P^0(\cdot, \cdot \mid s) \in \mathbb{R}^{1 \times SA}. \tag{228}$$

With this in hand, the uncertainty set (definition in (5)) with $\ell_\infty$ norm for any $P^\theta$ with $\theta \in \Theta$ can be represented as:

$$\mathcal{U}_\infty^{\mathsf{s},\tilde{\sigma}}(P_s^\theta) := \mathcal{U}_{\|\cdot\|}^{\mathsf{s},\tilde{\sigma}}(P_s^\theta) = \left\{ P_s' \in \Delta(\mathcal{S})^{\mathcal{A}} : \left\| P_s' - P_s^\theta \right\| \leq \tilde{\sigma} = \sigma \|1\|_\infty = \sigma \right\}. \tag{229}$$

So without loss of generality, we set the radius $\sigma \in (0, (1-c_0)]$ with $0 < c_0 < 1$. Before proceeding, we observe that as the uncertainty set above is defined with respect to $\ell_\infty$, it directly implies that for each $(s, a) \in \mathcal{S} \times \mathcal{A}$, the uncertainty set is independent and can be decomposed as

$$\mathcal{U}_\infty^{\mathsf{s},\tilde{\sigma}}(P_s^\theta) = \otimes \mathcal{U}_{\|\cdot\|}^{\mathsf{s},\tilde{\sigma}}(P_{s,a}^\theta) = \left\{ P_{s,a}' \in \Delta(\mathcal{S}) : \left\| P_{s,a}' - P_{s,a}^\theta \right\| \leq \sigma \right\}. \tag{230}$$

Notably, this indicates that using $s$-rectangular uncertainty set with $\ell_\infty$ norm as the divergence function is analogous to the case of using $(s, a)$-rectangular uncertainty set with $\ell_\infty$ norm. As a result, we follow the pipeline of the prior art Shi et al. [2023, Section C] which established the minimax-optimal lower bound for $(s, a)$-rectangular RMDPs with TV distance, which is analogous to the $\ell_\infty$ case. Towards this, we set $p, q, \Delta$ as the same as the ones in Shi et al. [2023, Section C.1], where we recall the expressions of $p, q, \Delta$ for self-contained as below: taking $c_1 := \frac{c_0}{2}$,

$$p = (1 + c_1) \max\{1 - \gamma, \sigma\} \quad \text{and} \quad \Delta \leq c_1 \max\{1 - \gamma, \sigma\}, \tag{231}$$

which ensure several facts:

$$0 \leq p \leq 1 \quad \text{and} \quad p \geq q \geq \max\{1 - \gamma, \sigma\}. \tag{232}$$

**Value functions and optimal policies.** For each RMDP instance $\mathcal{M}_\theta \in \mathcal{M}_{\ell_\infty}$, with some abuse of notation, we denote $\pi_\theta^\star$ as the optimal policy. In addition, let $V_\theta^{\pi,\sigma}$ (resp. $V_\theta^{\star,\sigma}$) represent the corresponding robust value function of any policy $\pi$ (resp. $\pi_\theta^\star$) with uncertainty level $\sigma$. Armed with these notations, the following lemma shows some essential properties concerning the value functions and optimal policies; the proof is postponed to Appendix F.3.

**Lemma 12.** *Consider any $\mathcal{M}_\theta \in \mathcal{M}_{\ell_\infty}$ and any policy $\pi$, one has*

$$\forall (i,j) \in \Theta: \quad V_{(i,j)}^{\pi,\sigma}(x_i) \leq \frac{\gamma\left(z_{(i,j)}^\pi - \sigma\right)}{(1-\gamma)\left(1 + \frac{\gamma\left(z_{(i,j)}^\pi - \sigma\right)}{1-\gamma(1-\sigma)}\right)(1 - \gamma(1-\sigma))}, \tag{233}$$

*where $z_{(i,j)}^\pi$ is defined as*

$$\forall (i,j) \in \Theta: \quad z_{(i,j)}^\pi := p\pi(j \mid x_i) + q\left[1 - \pi(j \mid x_i)\right]. \tag{234}$$

*In addition, the robust optimal value functions and the robust optimal policies satisfy*

$$\forall (i,j) \in \Theta, s \in \mathcal{X}: \quad V_{(i,j)}^{\star,\sigma}(s) = \frac{\gamma(p-\sigma)}{(1-\gamma)\left(1 + \frac{\gamma(p-\sigma)}{1-\gamma(1-\sigma)}\right)(1 - \gamma(1-\sigma))} \tag{235}$$

*and*

$$\pi_{(i,j)}^\star(j \mid x_i) = 1 \quad and \quad \pi_{(i,j)}^\star(0 \mid s) = 1 \quad \forall s \in \mathcal{X} \setminus \{x_i\}. \tag{236}$$

In words, this lemma shows that for any RMDP $\mathcal{M}_{(i,j)}$, the optimal policy on state $x_i$ satisfies $\pi_{(i,j)}^\star(j \mid x_i) = 1$ and will focus on $a = 0$ for all other states $s \in \mathcal{X} \setminus \{x_i\}$.

### F.2 Establishing the minimax lower bound

**Step 1: converting the goal to estimate $(i,j)$.** Now we are in position to derive the lower bound. Recall the goal is to control the following quantity associated with any policy estimator $\widehat{\pi}$ based on the dataset with in total $N_{\mathsf{all}}$ samples:

$$\max_{(i,j) \in \Theta} \mathbb{P}_{(i,j)}\left\{\max_{s \in \mathcal{X} \cup \mathcal{Y}}\left(V_{(i,j)}^{\star,\sigma}(s) - V_{(i,j)}^{\widehat{\pi},\sigma}(s)\right)\right\} \geq \max_{(i,j) \in \Theta} \mathbb{P}_{(i,j)}\left\{\max_{s \in \mathcal{X}}\left(V_{(i,j)}^{\star,\sigma}(s) - V_{(i,j)}^{\widehat{\pi},\sigma}(s)\right)\right\}. \tag{237}$$

To do so, we can invoke a key claim in Shi et al. [2023] here since our problem setting can be reduced to the same one in Shi et al. [2023]: With $\varepsilon \leq \frac{c_1}{32(1-\gamma)}$, letting

$$\Delta = 32(1-\gamma)\max\{1-\gamma, \sigma\}\varepsilon \leq c_1 \max\{1-\gamma, \sigma\} \tag{238}$$

which satisfies (231), it leads to that for any policy $\widehat{\pi}$ and all $(i,j) \in \Theta$,

$$V_{(i,j)}^{\star,\sigma}(x_i) - V_{(i,j)}^{\widehat{\pi},\sigma}(x_i) \geq 2\varepsilon\left(1 - \widehat{\pi}(j \mid x_i)\right),$$

$$\forall s \in \mathcal{X} \setminus \{x_i\}: \quad V_{(i,j)}^{\star,\sigma}(s) - V_{(i,j)}^{\widehat{\pi},\sigma}(s) \geq 2\varepsilon\left(1 - \widehat{\pi}(0 \mid s)\right). \tag{239}$$

Before continuing, we introduce a useful notation for the subset of $\Theta$ excluding the cases with state $i$ is selected:

$$\forall i \in \mathcal{S}: \quad \Theta_{-i} = \Theta \setminus \{(i',j) : i' = i, j \in \mathcal{A} \setminus \{0\}\}. \tag{240}$$

Armed with the above facts and notations, we first suppose there exists a policy $\widehat{\pi}$ such that for some $(i,j) \in \Theta$,

$$\mathbb{P}_{(i,j)}\left\{V_{(i,j)}^{\star,\sigma}(x_i) - V_{(i,j)}^{\widehat{\pi},\sigma}(x_i) \leq \varepsilon\right\} \geq \frac{3}{4}. \tag{241}$$

which in view of (239) indicates that we necessarily have $\widehat{\pi}(j \mid x_i) \geq \frac{1}{4}$ with probability at least $\frac{3}{4}$.

As a result, taking

$$j' = \arg\max_{a \in \mathcal{A}} \widehat{\pi}(a \mid x_i), \tag{242}$$

we are motivated to construct the following estimate of $\theta$:

$$\widehat{\theta} \begin{cases} = (i, j') & \text{if } j' > 0 \\ \in \mathcal{G}_{-w} & \text{if } j' = 0, \end{cases} \tag{243}$$

which satisfies

$$\mathbb{P}_{(i,j)}\{\widehat{\theta} = (i,j)\} \geq \mathbb{P}_{(i,j)}\{j' = j\} \geq \mathbb{P}_{(i,j)}\{\widehat{\pi}(j \mid x_i) > \frac{1}{A}\} \geq \frac{3}{4}. \tag{244}$$

**Step 2: developing the probability of error in testing multiple hypotheses.** Before proceeding, we discuss the dataset consisting of in total $N_{\text{all}}$ independent samples. Observing that each RMDP inside the set $\mathcal{M}_{\ell_\infty}$ are constructed symmetrically associated with one pair of states $(x_i, y_i)$ for all $i \in \mathcal{S}$ and another action $j \in \mathcal{A} \times \{0\}$, respectively. Therefore, it is obvious that the dataset is supposed to be generated uniformly on each $(x_i, y_i, j)$ to maximize the information gain, leading to $\frac{N_{\text{all}}}{S(A-1)}$ samples for any states-action $(x_i, y_i, j)$ with $i \in \mathcal{S}, j \in \mathcal{A} \setminus \{0\}$.

Then we are ready to turn to the hypothesis testing problem over $(i, j) \in \Theta$. Towards this, we consider the minimax probability of error defined as follows:

$$p_{\text{e}} := \inf_{\phi} \max_{(i,j) \in \Theta} \{\mathbb{P}_{(i,j)}(\phi \neq (i,j))\}, \tag{245}$$

where the infimum is taken over all possible tests $\phi$ constructed from the dataset introduced above.

To continue, armed with the above dataset with $N_{\text{all}}$ independent samples, we denote $\mu^{i,j}$ (resp. $\mu^{i,j}(s, a)$) as the distribution vector (resp. distribution) of each sample tuple $(s, a, s')$ under the nominal transition kernel $P^{(i,j)}$ associated with $\mathcal{M}_{(i,j)}$. With this in mind, combined with Fano's inequality from Tsybakov [2009, Theorem 2.2] and the additivity of the KL divergence (cf. Tsybakov [2009, Page 85]), we obtain

$$p_{\text{e}} \geq 1 - N_{\text{all}} \frac{\max\limits_{(i,j),(i',j') \in \Theta, (i,j) \neq (i',j')} \mathsf{KL}(\mu^{i,j} \mid \mu^{i',j'}) + \log 2}{\log |\Theta|}$$

$$\overset{(i)}{\geq} 1 - N_{\text{all}} \max_{(i,j),(i',j') \in \Theta, (i,j) \neq (i',j')} \mathsf{KL}(\mu^{i,j} \mid \mu^{i',j'}) - \frac{1}{2}$$

$$= \frac{1}{2} - N_{\text{all}} \max_{(i,j),(i',j') \in \Theta, (i,j) \neq (i',j')} \mathsf{KL}(\mu^{i,j} \mid \mu^{i',j'}) \tag{246}$$

where (i) holds by $\log |\Theta| \geq 2 \log 2$ as long as $S(A-1)$ are large enough. Then following the same proof pipeline of Shi et al. [2023, Section C.2], we can arrive at

$$p_{\text{e}} \geq \frac{1}{2} - \frac{N_{\text{all}}}{S(A-1)} \frac{4096}{c_1} (1-\gamma)^2 \max\{1-\gamma, \sigma\} \varepsilon^2 \geq \frac{1}{4}, \tag{247}$$

if the sample size is selected as

$$N_{\text{all}} \leq \frac{c_1 S(A-1)}{16396(1-\gamma)^2 \max\{1-\gamma, \sigma\} \varepsilon^2}. \tag{248}$$

**Step 3: summing up the results together.** Finally, we suppose that there exists an estimator $\widehat{\pi}$ such that

$$\max_{(i,j) \in \Theta} \mathbb{P}_{(i,j)} \left[ \max_{s \in \mathcal{X} \cup \mathcal{Y}} \left( V_{(i,j)}^{\star,\sigma}(s) - V_{(i,j)}^{\widehat{\pi},\sigma}(s) \right) \geq \varepsilon \right] < \frac{1}{4}, \tag{249}$$

then according to (237), we necessarily have

$$\forall s \in \mathcal{X}: \quad \max_{(i,j) \in \Theta} \mathbb{P}_{(i,j)} \left[ V_{(i,j)}^{\star,\sigma}(s) - V_{(i,j)}^{\widehat{\pi},\sigma}(s) \geq \varepsilon \right] < \frac{1}{4}, \tag{250}$$

which indicates

$$\forall s \in \mathcal{X}: \quad \max_{(i,j)\in\Theta} \mathbb{P}_{(i,j)}\left[V_{(i,j)}^{\star,\sigma}(s) - V_{(i,j)}^{\hat{\pi},\sigma}(s) < \varepsilon\right] \geq \frac{3}{4}. \tag{251}$$

As a consequence, (244) shows we must have

$$\forall (i,j) \in \Theta: \quad \mathbb{P}_{(i,j)}\left[\hat{\theta} = (i,j)\right] \geq \frac{3}{4} \tag{252}$$

to achieve (249). However, this would contract with (247) if the sample size condition in (248) is satisfied. Thus, we complete the proof.

### F.3 Proof of Lemma 12

Without loss of generality, we first consider any $\mathcal{M}_{(i,j)}$ with $(i,j) \in \mathcal{S} \times \mathcal{A} \setminus \{0\}$. Following the same routine of Shi et al. [2023, Section C.3.1], we can verify that the order of the robust value function $V_{(i,j)}^{\pi,\sigma}$ over different states satisfies

$$\forall k \in \mathcal{S}: \quad V_{(i,j)}^{\pi,\sigma}(x_k) \leq V_{(i,j)}^{\pi,\sigma}(y_k), \tag{253}$$

which means the robust value function of the states inside $\mathcal{X}$ are always not larger than the corresponding states inside $\mathcal{Y}$.

Then we denote the minimum of the robust value function over states as below:

$$V_{(i,j),\min}^{\pi,\sigma} := \min_{s\in\mathcal{S}} V_{(i,j)}^{\pi,\sigma}(s). \tag{254}$$

In the following arguments, we first take a moment to assume $V_{(i,j),\min}^{\pi,\sigma} = V_{(i,j)}^{\pi,\sigma}(x_i)$. With this in mind, we arrive at

$$V_{(i,j)}^{\pi,\sigma}(y_i) = 1 + \gamma(1-\sigma)V_{(i,j)}^{\pi,\sigma}(y_i) + \gamma\sigma V_{(i,j),\min}^{\pi,\sigma} = \frac{1 + \gamma\sigma V_{(i,j)}^{\pi,\sigma}(x_i)}{1 - \gamma(1-\sigma)}. \tag{255}$$

Then, when we move on to the characterization of the robust value function at state $x_i$. To do so, we notice two important facts:

1) The nominal transition probability $P_{x_i,a}^{(i,j)}$ at state-action pair $(x_i, a)$ for any $a \in \mathcal{A}$ is a Bernoulli distribution (see (226) and (224)). The TV distance and the $\ell_\infty$ norm between two Bernoulli distribution are the same.

2) Invoking the definitions of the nominal transition probability in (226) and (224), we have

$$P_{x_i,j}^{(i,j)} = p\mathbb{1}(s' = y_i) + (1-p)\mathbb{1}(s' = x_i)$$
$$P_{x_i,a}^{(i,j)} = q\mathbb{1}(s' = y_i) + (1-q)\mathbb{1}(s' = x_i) \quad \forall a \in \mathcal{A} \setminus \{j\}. \tag{256}$$

With the above two facts in hand, our problem setting is reduced to the same one in Shi et al. [2023] and can reuse the results in Shi et al. [2023, Section C.3.1] to achieve

$$V_{(i,j)}^{\pi,\sigma}(x_i) \leq \frac{\frac{\gamma(z_{(i,j)}^{\pi}-\sigma)}{1-\gamma(1-\sigma)}}{(1-\gamma)\left(1 + \frac{\gamma(z_{(i,j)}^{\pi}-\sigma)}{1-\gamma(1-\sigma)}\right)}. \tag{257}$$

and

$$\pi_{(i,j)}^{\star}(j \mid x_i) = 1$$

$$V_{(i,j)}^{\star,\sigma}(x_i) = \frac{\frac{\gamma(z_{(i,j)}^{\pi^\star}-\sigma)}{1-\gamma(1-\sigma)}}{(1-\gamma)\left(1 + \frac{\gamma(z_{(i,j)}^{\pi^\star}-\sigma)}{1-\gamma(1-\sigma)}\right)} = \frac{\frac{\gamma(p-\sigma)}{1-\gamma(1-\sigma)}}{(1-\gamma)\left(1 + \frac{\gamma(p-\sigma)}{1-\gamma(1-\sigma)}\right)}. \tag{258}$$

Analogously, we can verify that for other $x_k \in \mathcal{X} \setminus \{x_i\}$,

$$\pi_{(i,j)}^{\star}(0 \mid x_k) = 1$$

$$V_{(i,j)}^{\star,\sigma}(x_k) = \frac{\frac{\gamma(p-\sigma)}{1-\gamma(1-\sigma)}}{(1-\gamma)\left(1 + \frac{\gamma(p-\sigma)}{1-\gamma(1-\sigma)}\right)}. \tag{259}$$

# G DRVI for $sa-$ rectangular algorithm for arbitrary norm

In order to compute the fixed point of $\widehat{\mathcal{T}}^\sigma$, distributionally robust value iteration (DRVI), is defined in Algorithm 1. For $sa$-rectangularity, starting from an initialization $\widehat{Q}_0 = 0$, the update rule at the $t$-th ($t \geq 1$) iteration is the following $\forall (s, a) \in \mathcal{S} \times \mathcal{A}$:

$$\widehat{Q}_t^\pi(s, a) = \widehat{\mathcal{T}}^\sigma \widehat{Q}_{t-1}^\pi(s, a) = r(s, a) + \gamma \inf_{\mathcal{P} \in \mathcal{U}_{\|.\|}^{\mathsf{sa},\sigma}(\widehat{P}_{s,a}^0)} \mathcal{P} \widehat{V}_{t-1}, \tag{260}$$

where $\widehat{V}_{t-1}(s) = \max_\pi \widehat{Q}_{t-1}^\pi(s, a)$ for all $s \in \mathcal{S}$.

Directly solving (260) is computationally expensive since it involves optimization over a $S$-dimensional probability simplex at each iteration, especially when the dimension of the state space $\mathcal{S}$ is large. Fortunately, given strong duality (260) can be equivalently solved using its dual problem, which concerns optimizing a two variable ($\lambda$ and $\omega$) and thus can be solved efficiently. The specific form of the dual problem depends on the choice of the norm $\|.\|$, which we shall discuss separately in Appendix C.3. To complete the description, we output the greedy policy of the final Q-estimate $\widehat{Q}_T$ as the final policy $\widehat{\pi}$, namely,

$$\forall s \in \mathcal{S}: \quad \widehat{\pi}(s) = \arg\max_a \widehat{Q}_T(s, a). \tag{261}$$

Encouragingly, the iterates $\left\{\widehat{Q}_t\right\}_{t \geq 0}$ of $DRVI$ converge linearly to the fixed point $\widehat{Q}^{\star,\sigma}$, owing to the appealing $\gamma$-contraction property of $\widehat{\mathcal{T}}^\sigma$.

---

**input:** empirical nominal transition kernel $\widehat{P}^0$; reward function $r$; uncertainty level $\sigma$; number of iterations $T$.
**initialization:** $\widehat{Q}_0(s, a) = 0$, $\widehat{V}_0(s) = 0$ for all $(s, a) \in \mathcal{S} \times \mathcal{A}$.
**for** $t = 1, 2, ..., T$ **do**
    **for** $s \in \mathcal{S}, a \in \mathcal{A}$ **do**
        Set $\widehat{Q}_t(s, a)$ according to (260);
    **end**
    **for** $s \in \mathcal{S}$ **do**
        Set $\widehat{V}_t(s) = \max_a \widehat{Q}_t(s, a)$;
    **end**
**end**
**output:** $\widehat{Q}_T$, $\widehat{V}_T$ and $\widehat{\pi}$ obeying $\widehat{\pi}(s) := \arg\max_a \widehat{Q}_T(s, a)$.
**Algorithm 1:** Distributionally robust value iteration ($DRVI$) for infinite-horizon RMDPs for $sa$-rectangular for arbitrary norm

---

Using Algorithm 1, it allows getting an $\epsilon_{opt}$ error in the empirical MDP in the $sa$-rectangular case. In the $s$-rectangular case, finding an algorithm to get $\epsilon_{opt}$ is more difficult to use, as the policy is not deterministic anymore and 1 cannot anymore be applied. For $L_p$ norms, Clavier et al. [2023] derived an algorithm but for arbitrary norm we need to consider a more general problem for arbitrary norm in Appendix G

