# OpenReview forum: "Near-Optimal Distributionally Robust Reinforcement Learning with General $L_p$ Norms"
_NeurIPS.cc/2024/Conference — NeurIPS 2024 poster_

### Official Review · Reviewer_QVRk · 2024-07-06

**Soundness:** 3
**Presentation:** 3
**Contribution:** 3
**Rating:** 7
**Confidence:** 3

**Summary:**

The paper investigates distributionally robust Markov decision processes (RMDPs) in the discounted infinite-horizon setting. The authors consider the case where the transition kernel can be arbitrarily chosen from a prescribed (given) uncertainty set centred around a reference kernel, where the uncertainty set is specified using a smooth norm--the authors consider L_p norms. Assuming access to a generative model (simulator) that can sample from the reference transition kernel, the authors analyse a model-based approach that first constructs an empirical approximation of the reference transition kernel and then applies distributional robust value iteration to it. Considering the robust value function--which measures the worst case performance over all the possible transition kernels in the uncertainty set--the authors show that their method convergences to the optimal robust policy under two commonly adopted conditions (sa-rectangularity and s-rectangularity) describing the decoupled nature of the uncertainty sets. The authors establish upper bounds and sample complexity results for the procedure they consider (or any procedure that achieves a certain optimization error), as well as algorithmic-independent lower bounds.

**Strengths:**

The paper makes several contributions. Firstly, the authors improve upon previous upper and lower bounds, demonstrating near minimax optimality for any choice of L_p norm. They show that the sample complexity for solving robust RL is at least the same as, and sometimes (when the uncertainty level is relatively large) smaller than, that of standard RL. This generalizes the findings of Shi et al., 2023, where minimax optimality and comparisons between robust RL and standard RL were only established for the total variation distance (p=1). Secondly, the results indicate that solving s-rectangular RMDPs is not more difficult than solving sa-rectangular RMDPs in terms of sample complexity.

On the technical side, the analysis seems non-trivial, utilizing a finite-sample approach established by the authors that leverages a novel dual formulation for RMDPs optimization, which could be of independent interest. The literature review is comprehensive, and the positioning and contributions of this work are clearly articulated.

**Weaknesses:**

Overall, I found the main paper to be well-written, though it does contain some minor typos, formatting inconsistencies, and punctuation errors within the main text.

Establishing the lower bounds in the s-rectangularity setting for any L_p norm, not just the L_{\infty} norm, would strengthen the paper, enabling it to discuss minimax optimality also in that setting.

**Questions:**

1- At the technical level, beyond considering a new dual formulation, what is the main difference compared to the proof scheme in Shi et al., 2023?

2- What challenges are involved in establishing lower bounds that hold for any L_p norm in the s-rectangularity case?

3- Is there a fundamental limitation to extending the results to the *entire* range of the uncertainty level?

4- On the motivation side, the authors write (page 2) that "in practice, numerous applications are based on optimizations or learning approaches that involve general norms beyond those that have already been studied". Could the authors provide relevant citations?

**Limitations:**

The paper includes numerous remarks that qualify the level of contribution and limitations of the results. The conclusion identifies the important task of establishing f-divergences as future work.

---

> ### Author Rebuttal · Authors · 2024-08-07
>
> **Answer to reviewer QVRk**
>
>
>
> We appreciate the reviewer for careful review and insightful feedback. It is rewarding to know that the reviewer recognizes the significance of our contributions. In what follows, we provide our response to the reviewer's comments.
>
>
>
> > Overall, I found the main paper to be well-written, though it does contain some minor typos, formatting inconsistencies, and punctuation errors within the main text.
>
> We will try to correct as much as possible all minor typos in the next version of the manuscript.
>
>
>
> >* 1- At the technical level, beyond considering a new dual formulation, what is the main difference compared to the proof scheme in Shi et al., 2023?
>
>
> The challenge in comparing to the work of  Shi et al., 2023 is the following. Is it possible to generalize the work of Shi et al., 2023  to general $L_p$ norms and to $s$-rectangular case ?
> One of the key of their proof was to bound the range of the value function by $1/\sigma$ for large radius $\sigma\geq 1-\gamma$. Is it possible to adapt it to $L_p$ RMDPs and to $s$-rectangular assumptions. Moreover is there a chance to concentrate more complex dual for $L_p$ RMDPs compare to TV case in Shi et al., 2023 ?
>
>
>
> From upper bound perspective:
>
> * **Adaptation  and generalisation of key lemma from Shi et al., 2023 to reduce sample complexity in the $L_p$ context.**
> Two new key lemmas 5 and 6 which are different from $TV$ case are derived,  introducing $C_g$ coefficient which is not present in Shi et al., 2023.
>
>
> * **Concentration of more complex dual for $L_p$ RMDPs compare to TV case**
> Concentration lemma (lemma 8) is very different from Shi et al. 2023 with additional term to control. Indeed as the dual form for $TV$ and $L_P$ , $p>1$  involve respectively a scalar optimization and a vector in the case of $L_p$, uniform concentration in lemma 8 is more challenging in the $L_p$ case with factor depending on the geometry of the norm called $C_s$.
>
> * **Adaptation and generalisation of the proof to $s$-rectangular case**
> Contrary to Shi et al., 2023 , our proof tackles the problem of $s$-rectangular.  We introduce quantities related to the stochasticity of the policy in the upper bound which is not present in Shi et al., 2023 to derive bound in the $s$-rectangular case.
>
>
> * Finally **from a lower bound perspective** : new lower bound using counter example with stocastic optimal policy  for $s$-rectangular case  are derived and result from $sa$ rectangular is extended to $L_p$ norms.
>
>
> > 2- What challenges are involved in establishing lower bounds that hold for any $L_p$ norm in the s-rectangularity case? and  Establishing the lower bounds in the s-rectangularity setting for any L_p norm, not just the L_\{linfty\} norm, would strengthen the paper, enabling it to discuss minimax optimality also in that setting.
>
>
> The lower bounds are for the case of $s$-rectangularity, which poses entirely new challenges compared to the case of $sa$-rectangularity: the
> optimal policies may be stochastic and difficult to characterise as closed forms, compared to the deterministic ones in $sa$ cases. When using different norms, the corresponding optimal policies might even not have closed forms, which is also the bottleneck to extend the $L_\infty$ case (Theorem 4) to more general $L_p$ arbitrary norms (as in Theorem 2).
>
>
>
> > Q3  Is there a fundamental limitation to extending the results to the entire range of the uncertainty level?
>
> Indeed, we consider the full range of the uncertainty level in our theorems.
> In Theorem 1 and 3, the entire possible range of uncertainty level $\sigma$ is considered (see Theorem 1 and 3 where we defined $\sigma_{max}$ and $\tilde{\sigma}_{max}$ in the $sa$- and $s$-rectangular case.
>
> > Q4- On the motivation side, the authors write (page 2) that "in practice, numerous applications are based on optimizations or learning approaches that involve general norms beyond those that have already been studied". Could the authors provide relevant citations?
>
>
> In the context of RMDPs the work of  [1] proposed to use weighted norm to define a RDMPs. Moreover, in other application without the scope of RDMPS, general norm are used in robust optmization such as in [2,3].
>
> give examples from other areas, such as supervised learning, adversarial learning.
>
> >[1] Reazul Hasan Russel, Bahram Behzadian, and Marek Petrik. "Optimizing norm-bounded weighted ambiguity
> sets for robust mdps"  arXiv preprint arXiv:1912.02696, 2019
>
> > [2] Dimitris Bertsimas, Dessislava Pachamanova, Melvyn Sim  "Robust linear optimization under general norms
>
> > [3]  J Rony, L Gustavo, R Sabourin, E Granger : Decoupling direction and norm for efficient gradient-based l2 adversarial attacks

---

> > ### Comment · Reviewer_QVRk · 2024-08-10
> >
> > I would like to thank the authors for their response. I continue to hold a positive outlook on this work, and I maintain my score.

---

> > > ### Author Response · Authors · 2024-08-12
> > >
> > > We are grateful for your feedback and will incorporate your suggestions into the final version of the manuscript.

---

### Official Review · Reviewer_Tr2g · 2024-07-12

**Soundness:** 3
**Presentation:** 3
**Contribution:** 2
**Rating:** 7
**Confidence:** 3

**Summary:**

This paper dives into the theoretical understanding of learning robust MDPs with a generative model. The robust set is modeled as a distribution ball induced by the general $L_p$-norm centered around the nominal model. The sample complexity is provided for both  $\mathcal{S}\times\mathcal{A}$-rectangular and $\mathcal{S}$-rectangular robust sets and for the former case the minimax optimality is established.

**Strengths:**

**Originality and Significance:**

1. The paper considers a generalized smooth version of $L_p$-norm with $p\geq 2$ (whose algorithm and analysis also easily extend to the TV case) for the robust set and provides the corresponding sample complexity for the first time.
2. The paper studies both $\mathcal{S}\times\mathcal{A}$-rectangular and $\mathcal{S}$-rectangular robust sets. For both cases, the corresponding sample complexity improves over the prior art for a special case (standard $L_p$ norm provided by [1]) when the robust set size is relatively large.
3. For the $\mathcal{S}\times\mathcal{A}$-rectangular case, the paper proves a matching lower bound showing its minimax optimality. For the $\mathcal{S}$-rectangular case, the paper provides a lower bound for the special case of $L_{\infty}$-norm robust set. The results also shows the learning $\mathcal{S}$-rectangular robust MDPs with general $L_p$-norm is no harder than learning $\mathcal{S}\times\mathcal{A}$-rectangular robust MDPs.

**Clarity and Quality:**

The paper is well written. All the theoretical results are sound and are well proved. Some typos exist but are minor. See my questions below.

**References:**

[1] Clavier, P., Pennec, E.L. and Geist, M., 2023. Towards minimax optimality of model-based robust reinforcement learning. *arXiv preprint arXiv:2302.05372*.

**Weaknesses:**

1. One of the key message is that learning general $L_p$-norm robust MDPs is easier than learning standard MDPs. However, this result is not superising given the prior arts on certain special cases such as the TV norm robust MDP [2]. So I think it is less discussed how the exact choice of the generalized $L_p$-norm would further affect the diffuculty of learning this type of robust MDPs.
2. Although theoretically sound, it is not well discussed when and why people need to consider modeling the inherent uncertainty of the MDP parameters using the generalized $L_p$-norm.

**References:**

[2] Shi L, Li G, Wei Y, Chen Y, Geist M, Chi Y., 2023. The curious price of distributional robustness in reinforcement learning with a generative model. Advances in Neural Information Processing Systems.

**Questions:**

1. Regarding Section 8.3:
   - (Line 687-689) It is claimed that the dual form is derived for *arbitrary* norms, which seems like an overclaim since the proofs still depend on the assumptions in Definition 1 (e.g., Line 718). I think this is a misleading claim and needs revision.
   - It seems that the proof of duality in Section 8.3 resembles that in [1] without mentioning that work. Could the authors highlight a bit more the difference between the proofs here the and the proofs for duality in the previous work [1]?
2. Regarding Theorems 3 and 4 for the $\mathcal{S}$-rectangular case. The upper bound in Theorem 3 (Equation (21)) involves a minimization over two terms (this is different from the $\mathcal{S}\times\mathcal{A}$-case where only the first term appears, which I think comes from a different upper bound on the span of the value function), but the lower bound in Theorem (Equation (23)) for a special case only involves the first term in the minimization in the upper bound, which seems like a contradiction. Could the authors explain more on that?
3. Regarding Theorem 4. For the $\mathcal{S}$-rectangular robust set case, the lower bound is only for the standard $L_{\infty}$-norm case. Could the authors elaborate more on the difficulty in proving a lower bound for the general $L_p$-norm case?
4. Some typos and grammatical mistake that I found:
   - Line 137 to 138.
   - Line 144: simple -> simplex.
   - Line 155: typo in defining the domain of the norm.
   - Inconsistent usage of label (sometimes Theorem x but sometimes Theorem (x)), e.g., Lines 62 and 71, Lines 218 and 222.

**Limitations:**

Please refer to the weakness section and the question section above.

---

> ### Author Rebuttal · Authors · 2024-08-07
>
> **Aswer to revierwer Tr2g**
>
> We appreciate the reviewer's comprehensive feedback and recognition of the significance of our contributions.
>
> > Q1. One of the key message is that learning general $L_p$-norm robust MDPs is easier than learning standard MDPs. So I think it is less discussed how the exact choice of the generalized $L_p$-norm would further affect the difficulty of learning this type of robust MDPs.
>
> Thank you for the clarification question. How different $L_p$ norm metric compared to TV affects the sample complexity or difficulty of learning RMDPs is present in two coefficients $C_g$ and $C_s$. For instance, in $sa$-rectangular cases, we recall that our sample complexity results is in the order of $O(\frac{ S A}{(1-\gamma)^2 \max \{ 1-\gamma, C_g \sigma\} \varepsilon^2} + \frac{ SA C_S \left\|1_S\right\|  }{(1-\gamma)^2 \epsilon})$.
>
> - **The coefficient $C_g>0$ is related to the geometry of the norm.** For classical non-weighted $L_p$ norms, this coefficient is bigger than $1$, leading to sample complexity smaller than in classical TV case. However, for arbitrary weighed norms, this coefficient can be smaller than 1, which imply a bigger sample complexity compared to TV cases.
> - **The coefficient $C_s>0$ represent how smooth is the gradient of the norm.** This coefficient does not affect the sample complexity for small error $\epsilon \leq    \left(\max \{1-\gamma, C_g \sigma\}\right) /\left(C_S\left\|1_S\right\|\right)$, with $\sigma$ the radius of the uncertainty ball, and $\left\| . \right\|$ the considered norm, as it is a second order term in terms of $\epsilon$ or burn in term. The smoother the gradient of the norm is, the smaller is $C_s$ which lead to smaller burn-in term.
>
> > Q2. Although theoretically sound, it is not well discussed when and why people need to consider modeling the inherent uncertainty of the MDP parameters using the generalized $L_p$-norm.
>
> Thank you for the insightful suggestion. We consider general $L_p$ norm since its soundness in both theory and practice. [4] use TV for RL learning of Online 3D Bin or [6] for Offline policy optimization and [5] use $L_\infty$ for learning S-rectangular Robust MDPs. Moreover, the general $L_p$ problem formulation has unique interesting optimization properties, namely, relaxing slightly the problems will lead to a closed-form dual problems ([1,2] developed respectively Value Iteration and policy-based methods to solve RMDPs with $L_p$ norms).
>
> >[1] Navdeep Kumar, Esther Derman, Matthieu Geist, Kfir Levy, Shie Mannor "Policy Gradient for Rectangular Robust Markov Decision Processes"
>
> >[2] Navdeep Kumar, Kfir Levy, Kaixin Wang, Shie Mannor "Efficient Policy Iteration for Robust Markov Decision Processes via Regularization"
>
> >[4] Adjustable Robust Reinforcement Learning for Online 3D Bin Packing
> Yuxin Pan, Yize Chen, Fangzhen Lin
>
> >[5] B Behzadian, M Petrik, CP Ho  Fast Algorithms for $ L_\infty $-constrained S-rectangular Robust MDPs, Advances in Neural Information Processing Systems, 2021
>
> >[6] Lee, J., Jeon, W., Lee, B., Pineau, J., and Kim, K.-E. (2021). Optidice: Offline policy optimization via
> Stationary distribution correction estimation.
>
> > Q3 (Line 687-689) It is claimed that the dual form is derived for arbitrary norms, which seems like an overclaim since the proofs still depend on the assumptions in Definition 1 (e.g., Line 718). I think this is a misleading claim and needs revision.
>
> Yes it is true, we will replace it, this is a typo.
>
> >  Q4 It seems that the proof of duality in Section 8.3 resembles that in [1] without mentioning that work. Could the authors highlight a bit more the difference between the proofs here the and the proofs for duality in the previous work [1]?
>
> There are many differences between our work and Clavier et al. [2023], such as our key lemma 5 and 6 to improve the sample complexity, the Variance decomposition (equation 64 of our work) which is central in our paper and differ from Clavier et al. [2023]. Please refer to the General Response for all technical contributions and new challenge addressed in our paper upon Clavier et al. [2023].
>
> > Q4. Regarding Theorems 3 and 4 for the $\mathcal{S}$-rectangular case. The upper bound in Theorem 3 (Equation (21)) involves a minimization over two terms (this is different from the $\mathcal{S} \times \mathcal{A}$-case where only the first term appears, which I think comes from a different upper bound on the span of the value function), but the lower bound in Theorem (Equation (23)) for a special case only involves the first term in the minimization in the upper bound, which seems like a contradiction. Could the authors explain more on that?
>
> Thank you for the insightful question. In equation (20) of Theorem 3, using the result for the  $L_\infty$ gives that the upper bound is exatly proportional to $\frac{SA}{\epsilon^2(1-\gamma)^2\max\{1-\gamma,\sigma \}  }$. The other cases of the upper bound in equation (20) involve the quantity
>
> $\min_{s \in \mathcal{S}} (  \lVert \pi_s^*\ \rVert_*   \lVert 1_A \rVert ,  \lVert \hat{\pi_s} \rVert_*   \lVert 1_A \rVert )=1$
> in the case of $L_\infty$ RMDPs because.
>
> $\min_{s \in \mathcal{S}} (  \lVert \pi_s^*\ \rVert_*   \lVert 1_A \rVert ,  \lVert \hat{\pi_s} \rVert_*   \lVert 1_A \rVert )=\min_{s \in \mathcal{S}} (  \lVert \pi_s^*\ \rVert_1   \lVert 1_A \rVert_\infty ,  \lVert \hat{\pi_s} \rVert_1   \lVert 1_A \rVert_\infty )=1$
> So there is no contradiction with our lower bound which exactly match the upper bound.

---

> > ### Comment · Reviewer_Tr2g · 2024-08-09
> >
> > Thank you very much for your detailed response! I agree with your clarifications on several points that I am concerned with. But I think the proof of the duality result in the paper still ought to mention the existence of the results in Clavier et al. [2023]. I currently have no further questions. Given the answer and the contributions of the work, I am willing to increase my score.

---

> > > ### Author Response · Authors · 2024-08-12
> > >
> > > Thank you for your response, we will revise the final version of the manuscript, following your recommendations.

---

> ### Author Response · Authors · 2024-08-07
> **Folllowing of the Aswer to revierwer Tr2g**
>
> > Q5 Regarding Theorem 4. For the $\mathcal{S}$-rectangular robust set case, the lower bound is only for the standard $L_{\infty}$-norm case. Could the authors elaborate more on the difficulty in proving a lower bound for the general $L_p$-norm case?
>
>
> The lower bounds are for the case of $s$-rectangularity, which poses entirely new challenges compared to the case of $sa$-rectangularity: the
> optimal policies can be stochastic and difficult to characterize as closed forms, compared to the deterministic ones in $sa$ cases. When using different norms, the corresponding optimal policies might even not have closed forms, which is also the bottleneck to extend the $L_\infty$ case (Theorem 4) to more general $L_p$/arbitrary norms (as in Theorem 2).
>
> > Q6. Some typos and grammatical mistake that I found:
>
> Many thanks for pointing out the typos. We have revised them and check throughout the entire paper again.

---

### Official Review · Reviewer_5t5a · 2024-07-12

**Soundness:** 3
**Presentation:** 2
**Contribution:** 2
**Rating:** 6
**Confidence:** 3

**Summary:**

The paper presents an analysis of the sample complexity of solving distributionally robust Markov decision processes (RMDPs) with general Lp norms as the distance metric for the uncertainty set. The authors consider both the sa-rectangular and s-rectangular settings and provide near-optimal upper and lower bounds on the sample complexity.

**Strengths:**

For sa-rectangular RMDPs, the authors derive a near-minimax optimal sample complexity upper bound that matches the lower bound, showing their result is tight for almost the full range of the uncertainty level. This improves upon prior work which only achieved minimax optimality for the specific case of TV distance.

For s-rectangular RMDPs, the authors provide the first sample complexity upper and lower bounds, showing that solving s-rectangular RMDPs is not harder than solving sa-rectangular RMDPs in terms of sample requirement. This is an interesting and non-trivial result, as s-rectangular RMDPs have a more complicated optimization formulation.

The authors show that solving robust MDPs can be at least as sample-efficient as, and sometimes more sample-efficient than, solving standard MDPs. This provides important motivation for the study and use of distributionally robust RL.


This work develops new technical tools such as new concentration lemmas to obtain tighter sample complexity bounds.

**Weaknesses:**

The primary issue is that this paper closely aligns with Clavier et al. [2023]. To clarify the unique contributions, the authors should include more in-depth discussions comparing their work to Clavier et al. [2023]. For instance, the proof in Section 8.3 appears to follow Clavier et al. [2023] without proper citation.

A potential limitation of the paper is its focus on the tabular setting. It would be valuable to explore whether these insights can be applied to the function approximation setting as well. Nevertheless, this does not diminish the importance of the contributions presented in this work.

**Questions:**

na

**Limitations:**

The authors have discussed limitations in the paper.

---

> ### Author Rebuttal · Authors · 2024-08-07
>
> **Answer to reviewer 5t5a**
>
> We appreciate the reviewer's careful review and insightful feedback. It is rewarding to know that the reviewer recognizes the significance of our contributions. In what follows, we provide our response to the reviewer's comments.
>
> > Q1) The primary issue is that this paper closely aligns with Clavier et al. [2023]. To clarify the unique contributions, the authors should include more in-depth discussions comparing their work to Clavier et al. [2023]. For instance, the proof in Section 8.3 appears to follow Clavier et al. [2023] without proper citation.
>
>
> There are many differences between our work and Clavier et al. [2023], e.g., our key lemma 5 and 6 to improve the sample complexity and the Variance decomposition (equation 64 of our work) which is central in our paper and differ from Clavier et al. [2023]. Please refer to the General Response for all technical contributions and new challenge addressed in our paper upon Clavier et al. [2023].
>
>
> Section 8.3 of this work about optimization duality is similar to Clavier et al. [2023], while our proof is slightly more general as it works for weighted $L_p$ norms. We will refer to it in our last version of the proof of this manuscript. Moreover, the difference in optimization part is that we are not relaxing the dual of RMDPs problem like in Lemma C2 of Clavier et al. [2023].
>
>
> > Q2) A potential limitation of the paper is its focus on the tabular setting. It would be valuable to explore whether these insights can be applied to the function approximation setting as well. Nevertheless, this does not diminish the importance of the contributions presented in this work.
>
> Thanks for proposing this interesting direction, investigating robust RL with linear function approximation or other settings such as online settings, or model-free algorithms are definitely interesting directions in the future work. Robust RL with linear function has inspired some recent works such as [1,2,3]. We believe the findings of current results in tabular cases about norms lays a solid foundation to carry out to cases with linear function approximation, e.g., the finding of using any $L_p$ norm may lead to less sample size requirement. While the entire pipeline in this work will need to adapt or change a lot since this setting involves additional challenges from problem definition (how to define a reasonable uncertainty set is still relatively open) towards algorithm design.
>
> > [1] Blanchet, J., Lu, M., Zhang, T., and Zhong, H. (2024). Double pessimism is provably efficient for distributionally robust offline reinforcement learning: Generic algorithm and robust partial coverage. Advances in
> Neural Information Processing Systems, 36.
>
> > [2] H Wang, L Shi, Y Chi  Sample complexity of offline distributionally robust linear markov decision processes arXiv preprint arXiv:2403.12946, 2024
>
> > [3] Liu, Z. and Xu, P. (2024a). Distributionally robust off-dynamics reinforcement learning: Provable efficiency
> with linear function approximation. arXiv preprint arXiv:2402.15399.

---

> > ### Comment · Reviewer_5t5a · 2024-08-13
> >
> > Thank you for your response. The clarification provided by the authors addresses my main concerns, and I would like to raise the score.

---

### Official Review · Reviewer_6HyR · 2024-07-13

**Soundness:** 3
**Presentation:** 3
**Contribution:** 3
**Rating:** 6
**Confidence:** 3

**Summary:**

This paper proposes tighter than prior art sample complexity bounds for Robust Markov Decision Processes. In sa-rectangularity and s-rectangularity conditions with non-zero uncertainty measured using $L_p$  around a nominal transition kernel, the upper bound is $\frac{SA}{(1-\gamma)^3\\epsilon^2}$. The setup assumes access to a simulator to collect $N \times S \times A$ samples to first learn a transition kernel and determines optimal policy using distributional robust value iteration. The goal is utilize few samples there after to identify the optimal policy

**Strengths:**

The paper is generally well-written, and results improve existing bounds. However, I didn't thoroughly read the proofs.

**Weaknesses:**

It would be better has NSA been not used in Eq. 17, 18, 19, 20, 21 . Although the intention of the authors is clear, but that would improved readability.

In Table 1, the bounds are used in order sense, it would improve readability to have written in $\mathcal{O}(.)$, especially since the lower and upper bounds are of same order

**Questions:**

It would great improve the draft with some empirical evaluation to back the theoretical bounds, even a toy example.
Especially important since the bounds are not significant improvement from the prior art

**Limitations:**

Please add experimental work to back the theoretical bounds

---

> ### Author Rebuttal · Authors · 2024-08-07
>
> **Answer to Reviewer 6HyR**
>
> We appreciate the reviewer for recognizing our contributions, and for providing constructive suggestions. Here some comments on the different questions raised :
>
> > Q1: NSA been not use in Eq. 17, 18,19,20,21. Although the intention of the authors is clear, but that would improved readability.
>
> Thanks for the suggestions. We add a new notation $N_{\mathsf{all}} = NSA$ to represent the total number of samples and use $N_{\mathsf{all}}$ in Eq. 17, 18,19,20,21 or other equations to make it clearer for the readers.
>
> > Q2: In Table 1, the bounds are used in order sense, it would improve readability to have written in $\mathcal{O}(.)$, especially since the lower and upper bounds are of same order.
>
> Thank you for the clarification question. Throughout the paper (including Table 1), we use the same metric to measure the sample complexity --- the order that defined as:
> - "Let $\mathcal{X} := \big( S, A, \frac{1}{1-\gamma}, \sigma, \frac{1}{\varepsilon}, \frac{1}{\delta} \big)$.  The notation $f(\mathcal{X}) = O(g(\mathcal{X}))$ or $f(\mathcal{X}) \lesssim g(\mathcal{X})$ indicates that there exists a universal constant $C_1>0$ such that $f\leq C_1 g$,  the notation $f(\mathcal{X}) \gtrsim g(\mathcal{X})$ indicates that $g(\mathcal{X})=O(f(\mathcal{X}))$, and the notation $f(\mathcal{X})\asymp g(\mathcal{X})$ indicates that  $f(\mathcal{X}) \lesssim g(\mathcal{X})$ and  $f(\mathcal{X}) \gtrsim g(\mathcal{X})$ hold simultaneously. Additionally, the notation $\widetilde{O}(\cdot)$ is defined in the same way as ${O}(\cdot)$ except that it hides logarithmic factors."
>
> We added the footnote inside Table 1 to introduce the definition of order as above and clarify that we omit $\widetilde{O}(\cdot)$ in the caption, as the reviewer suggested.
>
>
> > Q3: It would great improve the draft with some empirical evaluation to back the theoretical bounds, even a toy example.
>
> Thank you for the constructive suggestion. We will conduct an experiment with simple RMDPs defined with $L_p$ balls on a toy example to show the dependency in the radius of the ball $\sigma$ on the sample complexity.
>
>
> > Q4: Does our sample complexity improvement significant improve prior art?
>
>
> For RL problems, especially in theory, the sample complexity dependency on the salient parameter $\frac{1}{1-\gamma}$ is one of the significant terms that dominate the sample efficiency. A line of work has committed to the endeavor of gradually improve this dependency towards optimal in various RL problems including but not limited to standard RL [1,2], and robust RL [3,4,5] that this work focus on.
>
>  In $sa$-rectangular cases. We focus on improving the sample complexity when $\sigma >1-\gamma$, where the prior art is still far from optimal. In $sa$-rectangular cases, we improve upon the prior art [5] sample complexity $\widetilde{O}(\frac{S A}{(1-\gamma)^4  \varepsilon^2}$) to $\widetilde{O}(\frac{S A}{(1-\gamma)^2 \sigma \varepsilon^2})$ by at least a factor of $\frac{1}{1-\gamma}$ and goes to $\frac{1}{(1-\gamma)^2}$ when $\sigma \geq O(1)$. Notably, this results is minimax-optimal in all salients parameters that match our lower bound for general $L_p$ norms, as long as the accuracy level $\epsilon$ is in a reasonable range. This is a significant improvement and near-optimal result. The same improvement is done for the $s$-rectangular case, by at least a factor of $1/(1-\gamma)$ compared to [5].
>
>
>
>
> >[1] Azar, Mohammad and Munos, Rémi and Kappen Hilbert , Minimax PAC bounds on the sample complexity of reinforcement learning with a generative model
>
> >[2]  Gheshlaghi Azar, R Munos, HJ Kappen - Minimax PAC bounds on the sample complexity of reinforcement learning with a generative model
>
> >[3]  Wenhao Yang, Liangyu Zhang, and Zhihua Zhang. Toward theoretical understandings of robust
> Markov decision processes: Sample complexity and asymptotics
>
> >[4] Laixi Shi, Gen Li, Yuting Wei, Yuxin Chen, Matthieu Geist, Yuejie Chi The Curious Price of Distributional Robustness in Reinforcement Learning with a Generative Model
>
> >[5] Pierre Clavier, Erwan Le Pennec, Matthieu Geist Towards minimax optimality of model-based robust reinforcement learning

---

> > ### Comment · Reviewer_6HyR · 2024-08-12
> >
> > Thank you very much for your response.
> > Updated score

---

### Author Rebuttal · Authors · 2024-08-07

# General response:
First, We would like thank the reviewers for their careful reading of the paper and their insightful and valuable feedback.


### Highlight of our new challenges and technical contributions

We would like to highlight the technical challenge to answer reviewers 6HyR and 5t5a and explain the comparison to prior work Clavier et al. [2023]. Our main contributions focus on statistical perspective.



Compared to Clavier et al. [2023], the motivation was to improve the sample complexity when the radius of the uncertainty set  $\sigma>1-\gamma$. The bottleneck of Clavier et al. [2023] is that when $\sigma>1-\gamma$, the sample complexity is proportional to $1/(1-\gamma)^4$, which is far from optimal. Towards optimal and more broader results, our technical contributions is summarized as:

1) **Use a tighter error decomposition.** Clavier et al. [2023] use a  decomposition of Q functions related to the work of [1] (Lemma C1 in their paper). However, this decomposition is not fine enough for large radius $\sigma$ to obtain tighter bounds. The Variance decomposition in our paper allow tighter control of errors and is completely different (equation 64 of our work) compared to their work. Denoting $V^{\star, \sigma}$ the robust value function and the different kernel , and  $P^\pi$, the projection  $\pi$ projection  of kernel $P$ and   $\underline{P}^V$ the worst kernel such as  $\underline{P}^V =\arg \min_{P} PV$,
 we use :

$ \sqrt{Var_{P^{\pi^{\star}}}\left(V^{\star, \sigma}\right)}=\left(\sqrt{Var_{P^{\pi^{\star}}}\left(V^{\star, \sigma}\right)}-\sqrt{Var_{\widehat{P}^{\pi^{\star}}}\left(V^{\star, \sigma}\right)}\right)+\sqrt{Var_{\widehat{P}^{\pi^{\star}}}\left(V^{\star, \sigma}\right)} $

$
 \leq\left(\sqrt{Var_{P^{\pi^{\star}}}\left(V^{\star, \sigma}\right)}-\sqrt{Var_{\widehat{P^{\pi^{\star}}}}\left(V^{\star, \sigma}\right)}\right)
 +\sqrt{\left|Var_{\widehat{P}^{\pi^{\star}}}\left(V^{\star, \sigma}\right)-Var_{\widehat{\underline{P}}^{\pi^{\star}, V}}\left(V^{\star, \sigma}\right)\right|}+\sqrt{Var_{\widehat{\underline{P}}^{\pi^{\star}, V}}\left(V^{\star, \sigma}\right)} .$

>[1] Azar, Mohammad and Munos, Rémi and Kappen Hilbert , Minimax PAC bounds on the sample complexity of reinforcement learning with a generative model

2) **Use exact dual of RMDPs problem and not a relaxion.** Clavier et al. [2023] have an extra term in their upper bound proportional to $\frac{2 \gamma \beta|S|^{1 / q}}{1-\gamma}$ contrary to our work. To control error in their second theorem (Th.2),  they use a relaxation of the dual form or RMDPs in their Lemma C2, which lead to non-minimax bound for large radius $\sigma$ contrary to our proof. On contrary, we use the exact form of the dual in our proof in Theorem 1 and 3.


3) **Use a key lemma about the range of the value function.** Clavier et al. [2023] cannot achieve sample complexity lower than $1/(1-\gamma)^3$ as they did not use fundamental idea than the range of value function in RMPDS is constrained for large radius. The key idea that allows us to get smaller sample complexity, namely lemma 5 and 6 for $sa$- and $s$-rectangular for radius $\sigma>1-\gamma$ for $L_p$ norms, is that the range of the value function is bounded for $L_p$ RMDPs.



**From a lower bound perspective**, we derive the first  minimax lower bound for general $L_p$ in the $sa$ rectangular cases. In addition, we develop the first lower bound $s$-rectangular cases using divergence function for $L_\infty$.
The main technical contributions focus on the lower bounds are for $s$-rectangularity, which poses entirely new challenges compared to the case of $sa$-rectangularity: the optimal policies can be stochastic and difficult to characterize as closed forms, compared to the deterministic ones in $sa$ cases. When using different $L_p$ norms, the corresponding optimal policies might even not have closed forms, which is also the bottleneck to extend the $L_\infty$ case (Theorem 4) to more general $L_p$ arbitrary norms (as in Theorem 2).

---

### Decision · Program_Chairs · 2024-09-25

**Decision:**

Accept (poster)

**Comment:**

The reviewers appreciated the optimal sample complexity bound in the Lp RMDP settings. I concur and believe are of great interest to the distributionally robust RL community. A clear accept.